# Profile Entropy: A Fundamental Measure for the Learnability and Compressibility of Distributions

**Yi Hao** , **Alon Orlitsky**
Department of Electrical and Computer Engineering
University of California, San Diego
{yih179, aorlitsky}@eng.ucsd.edu

## Abstract

The profile of a sample is the multiset of its symbol frequencies. We show that for samples of discrete distributions, profile entropy is a fundamental measure unifying the concepts of estimation, inference, and compression. Specifically, profile entropy: a) determines the speed of estimating the distribution relative to the best natural estimator; b) characterizes the rate of inferring all symmetric properties compared with the best estimator over any label-invariant distribution collection; c) serves as the limit of profile compression, for which we derive optimal near-linear-time block and sequential algorithms. To further our understanding of profile entropy, we investigate its attributes, provide algorithms for approximating its value, and determine its magnitude for numerous structural distribution families.

## 1 Introduction

Recent research in statistical machine learning, ranging from neural-network training and online learning, to density estimation and property testing, has advanced evaluation criteria beyond worst-case analysis. New performance measures apply more refined metrics relating the algorithm's accuracy and efficiency to the problem's inherent structure.

Consider for example learning an unknown discrete distribution from its i.i.d. samples (see also Section 2.2). The classical worst-case analysis states that in the worst case, the number of samples required to estimate a distribution to a given KL-divergence grows linearly in the alphabet size.

However, this formulation is pessimistic, since distributions are rarely the worst possible, and many practical distributions can be estimated with significantly smaller samples. Furthermore, once the sample is drawn, it reveals the distribution's complexity and hence the hardness of the learning task.

Going beyond worst-case analysis, one can design an *adaptive* learning algorithm whose theoretical guarantees vary according to the problem's simplicity. For example, Orlitsky and Suresh [2015] recently proposed an estimator that instance-by-instance achieves nearly the same performance as a genie algorithm designed with prior knowledge of the underlying distribution.

We introduce *profile entropy*, a fundamental measure for the complexity of discrete distributions, and show that it connects three vital scientific tasks: estimation, inference, and compression. The resulting algorithms have guarantees directly relating to the sample profile entropy, hence also adapt to the intrinsic simplicity of the tasks at hand.

The next subsection formalizes relevant concepts and useful notation.

**Sample Profiles and Their Entropy**

Consider an arbitrary sequence $x^n$ over a finite or countably infinite alphabet $\mathcal{X}$. The *multiplicity* $\mu_y(x^n)$ of a symbol $y \in \mathcal{X}$ is the number of times $y$ appears in $x^n$. The *prevalence* of an integer $\mu$ is the number $\varphi_\mu(x^n)$ of symbols in $x^n$ with multiplicity $\mu$. The *profile* of $x^n$ is the multiset $\varphi(x^n)$ of multiplicities of the symbols in $x^n$. We refer to it as a profile of *length* $n$. For example, consider the sequence $x^7 = bananas$, in which $a$ appears thrice, $n$ appears twice, and $b$ and $s$ each appears once. Then, the profile of the sequence is multiset $\varphi(x^7) = \{3, 2, 1, 1\}$.

The number $\mathcal{D}(S)$ of distinct elements in a multiset $S$ is its *dimension*. For convenience, we also write $\mathcal{D}(x^n)$ for profile dimension. In the above example, we have $\mathcal{D}(x^7) = \mathcal{D}(\varphi(x^7)) = 3$, corresponding to values $1, 2,$ and $3$. The dimension of a length-$n$ profile over $\mathcal{X}$ is at most $\min\{\sqrt{2n}, |\mathcal{X}|\}$. In general, the profile entropy $\mathcal{H}_n(p)$ is no more than $3\sqrt{n}$.

Let $\Delta$ be the collection of all discrete distributions, and $\Delta_\mathcal{X}$ be the collection of those over $\mathcal{X}$. Draw a size-$n$ sample $X^n$ from an arbitrary distribution in $p \in \Delta$. Then, the profile $\Phi^n$ of $X^n$ is a random multiset whose distribution depends on only $p$ and $n$. We therefore write $\Phi^n \sim p$, and call $\mathcal{H}_n(p) := \mathrm{H}(\Phi^n)$ the *profile entropy* with respect to $(p, n)$. For example, if we draw a sample of size $n = 3$ from $p = (\frac{1}{2}, \frac{1}{2})$, then profiles $\{1, 1, 1\}$, $\{2, 1\}$, and $\{3\}$ appear with probabilities $0, \frac{3}{4}$, and $\frac{1}{4}$, respectively. And the profile entropy is thus $\mathcal{H}_3(\frac{1}{2}, \frac{1}{2}) = H(0, \frac{3}{4}, \frac{1}{4}) \approx 0.56$.

Analogously, we call $\mathcal{D}_n := \mathcal{D}(\Phi^n)$, the *profile dimension* associated with $(p, n)$, and write $\mathcal{D}_n \sim p$.

For notational simplicity, we will assume that $\mathcal{H}_n(p) \geq 1$ throughout the paper, and respectively write $a \simeq b$, $a \gtrsim b$, and $a \lesssim b$ instead of $a = \tilde{\Theta}(b)$, $a = \tilde{\Omega}(b)$, and $a = \tilde{\mathcal{O}}(b)$, where the asymptotic notation hides logarithmic factors of $n$.

**Applications of Sample Profiles**

Sample profiles have essential applications in numerous aspects of scientific research, ranging from property inference to the study of degree distributions of networks/graphs.

*Property inference*    As Section 2.3 shows, profiles are sufficient for inferring all symmetric properties, such as entropy, Rényi entropy, and support size, not only in the sense of sufficient statistics, but also in the sense of Theorem 3, stating that profile-based estimators are as good as any others.

*Distribution learning*    The entropy of a sample profile, equaling its dimension in order with high probability (Theorem 1), directly characterizes how well we can estimate a distribution and approach the performance of the best human-designed estimator (Theorem 2), for every distribution.

*Theory of long tail*    The notable long tail theory in economics [Anderson, 2006] describes the strategy of selling a large number of different items that each sells in relatively small quantities. The profile of the product selling data, and the induced (PML) probability multiset estimate (Section 2.3), accurately characterize the tail shape of the data, and that of the underlying distribution, respectively.

*Password frequency lists*    In the research of password defense, it is vital to understand the distribution of passwords. Due to security concerns, organizations typically do not publish the complete data displaying each password and its frequency. Instead, they reveal the anonymized list of password frequencies, with each password hashed or replaced by some dummy string, which is equivalent to showing the password data's profile.

*Degree distributions of networks*    Degree distribution is one of the most widely studied attributes of networks (and graphs) that describes the fractions of nodes with different degrees. As the degree distribution ignores symbol labeling and focuses only on the frequency of each degree, it is equivalent to the profile of the node degree data.

## 2    Main Results

This paper aims to provide a thorough theory of profile entropy. Most of the results either are the first of their kind or significantly improve the state-of-the-art.

Specifically, Section 2.1 presents the fundamental equivalence relation between profile dimension and entropy (Thm. 1). Building on the equivalence, we respectively establish essential connections

between profile entropy and the estimation of discrete distributions (Section 2.2; Thm. 2), inference of their properties (Section 2.3; Thm. 4), and compression of sample profiles (Section 2.4; Thm. 5). These results characterize how well one can compete with an instance-optimal algorithm for each task, over *every single distribution*. For a real sense of how profile entropy behaves, Section 2.5 ultimately determines its magnitude for three prominent structural distribution families, log-concave (Thm. 6), power-law (Thm. 7), and histogram (Thm. 8). Going even further, Section 3 presents several additional applications and extensions of our theory and results, including robust learning under domain symbol permutations, profile entropy for mixture models, competitive property estimation, adaptive testing and classification, and connection to the method of types.

*For space considerations, we relegate detailed reviews on related work and most technical proofs to the supplementary material. For numerical analysis, we present two sets of experiments in Section B.5 and C.4 of the supplementary material, demonstrating the adaptiveness of the proposed methods in distribution estimation and inference of arbitrary property.*

## 2.1 Dimension-Entropy Equivalence of Profiles

The following theorem shows that for every distribution and sampling parameter $n$, the induced profile entropy and dimension are of the same order, with high probability.

**Theorem 1** (Entropy-dimension equivalence). *For any distribution $p \in \Delta$ and $\mathcal{D}_n \sim p$,*

$$\Pr(\mathcal{D}_n \simeq \mathcal{H}_n(p)) \geq 1 - \frac{1}{\sqrt{n}}.$$

We briefly comment on Theorem 1.

First, the theorem reveals a novel and fundamental relation between profile dimension and entropy. The relation also yields an intrinsic method to approximate the entropy of the sample's profile, a fairly involved functional, by only counting its dimension. In general, the number of possible length-$n$ profiles of a distribution could be as large the number of partitions of integer $n$, and grows with $n$ at a sub-exponential speed. Hence, even if $p$ is known, computing the exact value of $\mathcal{H}_n(p)$ could be hard. On the other hand, if one applies our theorem to approximate $\mathcal{H}_n(p)$, we only need to draw a sample $X^n \sim p$, and find its profile dimension, which is computable in linear time through counting. Section A.4 of the supplementary further illustrates how to estimate $\mathcal{H}_n$ with $m \ll n$ observations.

Second, the theorem serves as an essential building block for the subsequent results on distribution estimation, property inference, and profile compression, and enables us to establish their optimality. For example, in the process of deriving the optimal profile compression scheme and proving Theorem 5, we reason with $\mathcal{D}_n$ to bound the space of storing the profile, and utilize $\mathcal{H}_n(p)$ as an essential lower bound for lossless compression.

Third, despite the simple form of the theorem, the proof of this result is highly nontrivial, and relies on a recent breakthrough in solving the Shepp-Olkin monotonicity conjecture [Hillion et al., 2019], which asserts that the entropy of a Poisson-binomial random variable is monotone in the defining success probabilities, over a hypercube near the origin.

## 2.2 Competitive (Instance-Optimal) Distribution Estimation

Estimating distributions from their samples is a statistical-inference cornerstone, and has numerous applications, ranging from biological studies [Armañanzas et al., 2008] to language modeling [Chen and Goodman, 1999]. A learning algorithm $\hat{p}$ in this setting is called a *distribution estimator*, which associates with every sequence $x^n$ a distribution $\hat{p}(x^n) \in \Delta$. Given a sample $X^n \sim p$, we measure the performance of $\hat{p}$ in estimating distribution $p$ by the Kullback-Leibler (KL) divergence $\mathrm{D}(p \,\|\, \hat{p}(X^n))$.

Let $r_n(p, \hat{p}) := \min\{r : \Pr(\mathrm{D}(p \,\|\, \hat{p}(X^n)) \leq r) \geq 9/10\}$ be the *minimal KL error* $\hat{p}$ could achieve with probability at least $9/10$. Then, the *worst-case error* of estimator $\hat{p}$ over $\mathrm{P} \subseteq \Delta$ is $r_n(\mathrm{P}, \hat{p}) := \max_{p \in \mathrm{P}} r_n(p, \hat{p})$, and the lowest worst-case error for $\mathrm{P}$, achieved by the optimal estimator, is the *minimax error* $r_n(\mathrm{P}) := \min_{\hat{p}'} r_n(\mathrm{P}, \hat{p}')$. The most widely studied distribution set $\mathrm{P}$ is simply $\Delta_{\mathcal{X}}$. With $\mathcal{X}$ being finite, it has become a classical result that $r_n(\Delta_{\mathcal{X}}) = \Theta(|\mathcal{X}|/n)$, which is achievable, up to constant factors, by an add-constant estimator [Braess and Sauer, 2004, Kamath et al., 2015].

**Beyond minimax** Despite being minimax optimal, the $|\mathcal{X}|/n$-result and the algorithm, are not satisfiable from a practical point of view. The reason is that the formulation puts much of its emphasis on the worst-case performance, and ignores the intrinsic simplicity of $p$ in a pessimistic fashion. Hence, the desire to design more efficient estimators for practical distributions, like power-law, or Poisson, has led to algorithms that possess adaptive estimation guarantees.

Concretely, the minimax formulation has two modifiable components – the collection P and the error function $D$. A common approach to specifying P is adding structural assumptions, such as monotonicity, $m$-modality, and log-concavity, which, in many cases, makes algorithm refinement possible by leveraging structural simplicity. An orthogonal approach to encouraging adaptability without imposing structures is to replace absolute error by relative error, which we illustrate below.

**Competitive estimation** Without strong prior knowledge on the underlying distribution, a reasonable estimator should *naturally* assign the same probability to symbols appearing an equal number of times. *Competitive estimation* calls for finding a universally near-optimal estimator that learns *every* distribution as well as the best natural estimator that knows the true distribution.

Denote by $\mathcal{N}$ the collection of all natural estimators. For any distribution $p \in \Delta$ and sample $X^n \sim p$, a given estimator $\hat{p}$ incurs, with respect to the best natural estimator knowing $p$, an instance-by-instance *relative KL error* of

$$\mathrm{D}_{\mathrm{nat}}(p \,\|\, \hat{p}(X^n)) := \mathrm{D}(p \,\|\, \hat{p}(X^n)) - \min_{\hat{q} \in \mathcal{N}} \mathrm{D}(p \,\|\, \hat{q}(X^n)).$$

Analogous to the minimax formulation, we denote by $r_n^{\mathrm{nat}}(p, \hat{p}) := \min\{r : \mathrm{Pr}(\mathrm{D}_{\mathrm{nat}}(p \,\|\, \hat{p}(X^n)) \leq r) \geq 9/10\}$ the *minimal relative error* $\hat{p}$ achieves with probability at least $9/10$, by $r_n^{\mathrm{nat}}(\mathrm{P}, \hat{p})$ the *worst-case relative error* of $\hat{p}$ over $\mathrm{P} \subseteq \Delta$, and by $r_n^{\mathrm{nat}}(\mathrm{P})$ the *minimax relative error*.

**Old and new results** Initiating the competitive formulation, Orlitsky and Suresh [2015] show that a simple variant of the well-known Good-Turing estimator achieves $r_n^{\mathrm{nat}}(\Delta) \lesssim 1/n^{1/3}$, and a more involved estimator in Acharya et al. [2013] attains the optimal $r_n^{\mathrm{nat}}(\Delta) \simeq 1/\sqrt{n}$. For a fully adaptive guarantee, Hao and Orlitsky [2019b] further refine the bound and design an estimator $\hat{p}^\star$ achieving $r_n^{\mathrm{nat}}(p, \hat{p}^\star) \lesssim \mathbb{E}_{\mathcal{D}_n \sim p}[\mathcal{D}_n/n] \lesssim r_n^{\mathrm{nat}}(\Delta)$, for every $p \in \Delta$, but provide no lower bounds.

In this work, we completely characterize $r_n^{\mathrm{nat}}(p, \cdot)$ with essentially matching lower and upper bounds. Surprisingly, we show that for nearly every sample size $n$, the quantity behaves like $\mathcal{H}_n(p)/n$.

**Theorem 2** (Optimal competitive error). *There is a near-linear-time computable estimator $\hat{p}^\star$, such that for any distribution $p$ and $n$,*

$$r_n^{\mathrm{nat}}(p, \hat{p}^\star) \lesssim \frac{\mathcal{H}_n(p)}{n},$$

*where $\hat{p}^\star$ is the near linear-time computable estimator in Hao and Orlitsky [2019b] mentioned above. On the other hand, for any $H \in [0, \sqrt{n})$,*

$$\min_{\hat{p}} \max_{p:\mathcal{H}_n(p) \lesssim H} r_n^{\mathrm{nat}}(p, \hat{p}) \gtrsim \frac{H}{n}.$$

First, we comment on the lower bound. Due to the classical minimax formulation, one might expect a lower bound in one of the following two forms – for every $\hat{p}$, $r_n^{\mathrm{nat}}(p, \hat{p}) \gtrsim \mathcal{H}_n(p)/n$ for 1) some $p$ or 2) every $p$. Form 1) turns out to be weak under the competitive formulation. Specifically, let $p$ be a *trivial distribution* that assigns probability 1 to some symbol. Then, both the profile entropy and the error of the best natural estimator are zero, and the inequality trivially holds for every $\hat{p}$. Form 2), on the other hand, is purely impossible. Specifically, for every distribution $p$, one can set $\hat{p}$ to be best natural estimator, which leads to a relative error of zero, greater than $\mathcal{H}_n(p)/n$ unless $p$ is trivial.

Second, we illustrate the significance of the result. The notable work of Hardy and Ramanujan [1918] shows that the number of integer partitions of $n$, which equals the number of length-$n$ profiles, is at most $\exp(3\sqrt{n})$, implying that $\mathcal{H}_n(p) \leq 3\sqrt{n}$ for any $p \in \Delta$. Therefore, the $\mathcal{H}_n(p)/n$ upper and lower bounds in the theorem yields $r_n^{\mathrm{nat}}(\Delta) \simeq 1/\sqrt{n}$, recovering the main result of Orlitsky and Suresh [2015]. Besides set $\Delta$, the theorem and its proof also imply nearly tight minimax relative-error bounds on numerous distribution sets P. Below, we present two results that fall into this category. In both cases, the minimax relative error is much lower than $1/\sqrt{n}$ if the parameter involved is $o(\sqrt{n})$.

The first example addresses the set $\Delta_H$ of distributions whose $n$-sample profile entropy is $H$.

**Corollary 1.** *For any $H \gtrsim 1$, the minimax relative error over $\Delta_H$ is $r_n^{nat}(\Delta_H) \simeq H/n$.*

For a more concrete example, denote by $\mathcal{L}_\sigma$ the collection of log-concave distributions over $\mathbb{Z}$ whose variance is $\sigma^2$. Then, Theorem 2 and the profile entropy bounds in Theorem 6 imply

**Corollary 2.** *For any $1 \lesssim \sigma \leq \sqrt{n}$, the minimax relative error over $\mathcal{L}_\sigma$ is $r_n^{nat}(\mathcal{L}_\sigma) \simeq \sigma/n$.*

### 2.3 Competitive-Optimal Property Inference

Numerous practical applications call for inferring *property values* of an unknown distribution from its samples, including entropy for graphical modeling [Koller and Friedman, 2009], Rényi entropy for sequential decoding [Arikan, 1996], and support size for species richness estimation [Magurran, 2013]. Therefore, *property inference* has attracted considerable attention over the past few decades. For interested readers, please refer to Section B.3 in the supplementary for a two-page review of prior works and discussions about relevant methods.

**Property inference**    Formally, a *distribution property* over some collection $P \subseteq \Delta$ is a functional $f : P \to \mathbb{R}$ that associates with each distribution a real value. Given a sample $X^n$ from an unknown distribution $p \in P$, the problem of interest is to infer the value of $f(p)$. For this purpose, we employ another functional $\hat{f} : \mathcal{X}^* \to \mathbb{R}$, an *estimator* mapping every sample to a real value. We measure the statistical efficiency of $\hat{f}$ in approximating $f$ over P by its *absolute error* $|\hat{f}(X^n) - f(p)|$.

Given $X^n \sim p \in P$, the *minimal absolute error rate*, or simply *error*, that $\hat{f}$ achieves with probability at least $9/10$ is $r_n(p, \hat{f}) := \min\{r : \Pr(|\hat{f}(X^n) - f(p)| \leq r) \geq 9/10\}$, where the dependence on $f$ is *implicit*. While $p$ is often unknown, the *worst-case error* of an estimator $\hat{f}$ over all distributions in P is $r_n(P, \hat{f}) := \max_{p \in P} r_n(p, \hat{f})$, and the lowest worst-case error for P, achieved by the optimal estimator, is the *minimax error* $r_n(P) := \min_{\hat{f}'} r_n(P, \hat{f}')$.

**Profile maximum likelihood**    An important class of properties is the collection of symmetric ones, which encompasses numerous well-known distribution characteristics, such as Shannon entropy, Rényi entropy, support size, and $\ell_1$ distance to the uniform distribution. Symmetry connects the estimation of such property to the sample profile, a sufficient statistic for the task in hand. The general principle of maximum likelihood then provides an intuitive estimator, *profile maximum likelihood (PML)* [Orlitsky et al., 2004], that maximizes the probability of observing the profile.

Naturally and generally, we study symmetric property inference over a distribution collection $P \subseteq \Delta$ that is also *symmetric*, i.e., if $p \in P$, then P as well contains all the symbol-permuted versions of $p$. For every sample $x^n \in \mathcal{X}^n$ and symmetric P, the *PML estimator* over P maps $x^n$ to a distribution

$$\mathcal{P}_\varphi(x^n) := \arg\max_{p \in P} \Pr_{X^n \sim p} (\varphi(X^n) = \varphi(x^n)).$$

Given a sample $X^n \sim p \in P$ and a symmetric property $p$, the PML plug-in estimator uses $f \circ \mathcal{P}(X^n)$ to estimate $f(p)$. The PML estimator often behaves differently from the classical empirical distribution estimator. For example, if $P = \Delta$ and $\varphi = \{2, 1, 1\}$, the PML estimate turns out to be $\mathcal{P}_\varphi = (\frac{1}{5}, \frac{1}{5}, \frac{1}{5}, \frac{1}{5}, \frac{1}{5})$, deviating from the empirical distribution $(\frac{1}{2}, \frac{1}{4}, \frac{1}{4})$ by 0.8 in $L_1$ distance.

Recent researches [Acharya et al., 2017, Hao and Orlitsky, 2019a] show that for an extensive family of symmetric properties, including the previously mentioned four, the PML plug-in estimator *universally* achieves minimax error in the large-alphabet regime, up to constant factors.

The formulation of PML makes it part of two estimator classes, the maximum-likelihood and the *profile-based*, where the latter corresponds to estimators whose values depend on only the profile. The theorem below shows that profile-based estimators are sufficient for inferring symmetric properties.

**Theorem 3** (Sufficiency of profiles). *For any symmetric property $f$ and set $P \subseteq \Delta$, and estimator $\hat{f}$, we can construct an explicit estimator $\hat{F}$ over length-$n$ profiles satisfying*

$$r_n(p, \hat{f}) = r_n(P, \hat{F} \circ \varphi),$$

*where both estimators can have independent randomness.*

The next result shows that the PML estimator is adaptive to the simplicity of underlying distributions in inferring all symmetric properties, over any symmetric P. Specifically, the theorem states that the $n$-sample PML plug-in essentially performs as well as the optimal $n/\mathcal{H}_n(p)$-sample estimator, which approaches the performance of the optimal $n$-sample estimator if $p$ has a small $\mathcal{H}_n(p)$. Furthermore, for any property and estimator, there is a symmetric set P' for which this $1/\mathcal{H}_n(p)$ ratio is *optimal*.

**Theorem 4** (Competitiveness of PML). *For any symmetric property $f$ and set $P \subseteq \Delta$, and every distribution $p \in P$, the PML plug-in estimator satisfies*

$$r_n(p, f \circ \mathcal{P}_\varphi) \leq 2r_{n_p}(P),$$

*where $n_p :\simeq n/\mathcal{H}_n(p)$. On the other hand, for any estimator $\hat{f}$ and symmetric property $f$, there exists a symmetric set $P' \subseteq \Delta$ such that for some $p \in P'$,*

$$r_n(p, \hat{f}) \geq 2r_{n_p}(P').$$

We provide some brief comments here and more in Section 3. First, the above theorem holds for a polynomial-time PML approximation [Anari et al., 2020], and for any symmetric property, while nearly all previous works require the property to possess certain forms and be smooth. In particular, the algorithm in Anari et al. [2020] achieves the best-known guarantees for approximating PML, requires no additional assumptions on the distribution/property's structure, and works universally on all symmetric properties and adaptively on all profiles (hence distributions). Second, the result holds for any symmetric distribution set $P \subseteq \Delta$, which covers numerous domains of interest that appeared in the literature, such as the widely studied $\Delta_\mathcal{X}$, and its subset $\Delta_{1/|\mathcal{X}|}$ for the study of support size estimation, where each distribution's positive probabilities are at least $1/|\mathcal{X}|$. Third, the result trivially implies a weaker version in Acharya et al. [2017] where $\mathcal{H}_n(p)$ is replaced by $\sqrt{n}$, which, as we show in Section 2.5, can be significantly larger.

## 2.4 Optimal Compression of Profiles

None of the scientific applications in Section 1 is possible without first storing the sample profile. Hence, we focus on the task of lossless profile compression in this section. Besides the theoretical fundamentality and numerous applications, the task is essential as storing a sample's profile, compared with storing the entire sample sequence, often takes much less space. Specifically, Shannon entropy is the measure of limit of lossless compression, which, for sample $X^n \sim p \in \Delta$, is $nH(p)$, and for the sample's profile, is $\mathcal{H}_n(p)$. In particular, the sample entropy grows as $\Omega(n)$ whenever $p$ has an entropy of at least one, while the profile entropy is at most $3\sqrt{n}$ by our argument in Section 2.2.

While the $n$-to-$\sqrt{n}$ improvement is already significant, the compression schemes we propose under the standard block and sequential settings surely take profile compression to the next level. Specifically, for every distribution $p$ and sample size $n$, both schemes essentially compress the sample profile $\varphi(X^n)$ to its entropy $\mathcal{H}_n(p)$, the information-theoretic limit, in expectation. In other words, our algorithms are *instance-by-instance optimal* and essentially *unimprovable*. Furthermore, we achieve this instance optimality with *near-optimal time complexity* – both algorithms have a running time near-linear in the sample size $n$. Because of this instance optimality, we omit experimental evaluation.

**Block compression** We propose an intuitive and easy-to-implement block compression algorithm.

Recall that the profile of a sequence $x^n$ is the multiset $\varphi(x^n)$ of multiplicities associated with symbols in $x^n$. The ordering of elements in a multiset is not informative. Hence equivalently, we can compress $\varphi(x^n)$ into the set $\mathcal{C}(\varphi(x^n))$ of corresponding multiplicity-prevalence pairs, i.e.,

$$\mathcal{C}(\varphi(x^n)) := \{(\mu, \varphi_\mu(x^n)) : \mu \in \varphi(x^n)\}.$$

The number of pairs in $\mathcal{C}(\varphi(x^n))$ is equal to the profile dimension $\mathcal{D}(\varphi(x^n))$. Besides, both prevalence and its multiplicity are integers in $[0, n]$, and storing the pair takes $2\log n$ nats. Hence, it takes at most $2(\log n) \cdot \mathcal{D}(\varphi(x^n))$ nats to store the compressed profile. By Theorem 1, for any distribution $p \in \Delta$ and sample $X^n \sim p$,

$$\mathbb{E}[2(\log n) \cdot \mathcal{D}(X^n)] \simeq \mathcal{H}_n(p).$$

We have shown that storing a profile $\varphi$ as $\mathcal{C}(\varphi)$ is a near-optimal block compression scheme.

---

**Algorithm 1** Sequential Profile Compression

---
**input** sequence $(\mu_{x_t}(x^{t-1}))_{t=1}^n$, tree $\mathcal{T} = \varnothing$
**output** tree $\mathcal{T}$ that encodes the input sequence
  **for** t = 1 to n **do**
    **if** $\mu := \mu_{x_t}(x^{t-1}) \in \mathcal{T}$ **then**
      **if** $\mu + 1 \in \mathcal{T}$ **then**
        $\varphi_{\mu+1} := \mathcal{T}(\mu+1) \leftarrow \mathcal{T}(\mu+1) + 1$
      **else**
        add $(\mu + 1, 1)$ to $\mathcal{T}$
      **end if**
      **if** $\varphi_\mu = 1$ **then** delete $(\mu, \varphi_\mu)$ from $\mathcal{T}$
      **else** $\varphi_\mu := \mathcal{T}(\mu) \leftarrow \mathcal{T}(\mu) - 1$ **endif**
    **else**
      **if** $1 \notin \mathcal{T}$ **then** add $(1, 1)$ to $\mathcal{T}$
      **else** $\mathcal{T}(1) \leftarrow \mathcal{T}(1) + 1$ **endif**
    **end if**
  **end for**

---

**Sequential compression** For any sequence $x^n$, the setting for sequential profile compression is that at time step $t \in [n]$, the compression algorithm knows only $\varphi(x^t)$ and sequentially encodes the new information. This process is equivalent to providing the algorithm $\mu_{x_t}(x^{t-1})$ at time step $t$.

Suppress $x, x^t$ in the expressions for the ease of illustration. For efficient compression, we sequentially encode the profile $\varphi$ into a *self-balancing binary search tree* $\mathcal{T}$, with each node storing a multiplicity-prevalence pair $(\mu, \varphi_\mu)$ and $\mu$ being the search key. We present the compression scheme as Algorithm 1, and establish the following guarantee.

**Theorem 5.** *Algorithm 1 runs for exactly $n$ iterations, with an $\mathcal{O}(\log n)$ per-iteration time complexity. For an i.i.d. sample $X^n \sim p$, the expected space complexity is $\tilde{\Theta}(\mathcal{H}_n(p))$. On the other hand, any algorithm that compresses the profile losslessly has an expected space complexity of at least $\mathcal{H}_n(p)$.*

## 2.5 Optimal Characterization for Structured Families

In this section, we characterize the profile entropy of several important structured distribution families, including log-concave, power-law, histogram, and their mixtures. All the matching lower bounds are entirely new, and all the upper bounds, with the exception of that in Theorem 8, are much stronger than those induced by the prior work [Hao and Orlitsky, 2019b] via Theorem 1. For interested readers, see Section D of the supplementary for a detailed comparison.

**Log-concave** The log-concave family encompasses a broad range of discrete distributions, such as Poisson, hyper-Poisson, Poisson binomial, binomial, negative binomial, and geometric, and hyper-geometric, with broad applications to statistics [Saumard and Wellner, 2014], computer science [Lovász and Vempala, 2007], economics [An, 1997], and geometry [Stanley, 1989].

Formally, a distribution $p \in \Delta_{\mathbb{Z}}$ is *log-concave* if $p$ has a contiguous support and $p_x^2 \geq p_{x-1} \cdot p_{x+1}$ for all $x \in \mathbb{Z}$. The next result bounds the profile entropy of this family, and is *tight* up to logarithmic factors. For simplicity, henceforth we write $a \wedge b$ for $\min\{a, b\}$ (and $\vee$ for $\max$), and slightly abuse the notation and write $a \simeq b$ for $a + 1 = \tilde{\Theta}(b+1)$, which does not change the nature of the results.

**Theorem 6.** *Let $\mathcal{L}_\sigma \subseteq \Delta_{\mathbb{Z}}$ denote the collection of log-concave distributions with variance $\sigma^2$. Then,*

$$\max_{p \in L_\sigma} \mathcal{H}_n(p) \simeq \sigma \wedge \frac{n}{\sigma}.$$

*In particular, if we discretize a Gaussian variable $X \sim \mathcal{N}(\mu, \sigma^2)$ by rounding it to the nearest integer, the distribution of the resulting variable achieves the maximum, up to logarithmic factors. Moreover, such a discretization procedure preserves log-concavity for any continuous distribution over $\mathbb{R}$.*

**Power-law** Power-law is a ubiquitous structure appearing in many situations of scientific interest, ranging from natural phenomena such as the initial mass function of stars [Kroupa, 2001], species and

genera [Humphries et al., 2010], rainfall [Machado and Rossow, 1993], population dynamics [Taylor, 1961], and brain surface electric potential [Miller et al., 2009], to human-made circumstances such as the word frequencies in a text [Baayen, 2002], income rankings [Drăgulescu and Yakovenko, 2001], company sizes [Axtell, 2001], and internet topology [Faloutsos et al., 1999].

Formally, a discrete distribution $p \in \Delta_{\mathbb{Z}}$ is a *power-law with power* $\alpha \geq 0$ if $p$ has a support of $[k] := \{1, \ldots, k\}$ for some $k \in \mathbb{Z}^+ \cup \{\infty\}$ and $p_x \propto x^{-\alpha}$ for all $x \in [k]$. Note that if $\alpha \in [0, 1]$, the distribution is well-defined for only finite $k$. The next result fully characterizes the profile entropy of power-laws over all $\alpha, n$, and $k$ ranges, and significantly improves that in Hao and Orlitsky [2019b].

**Theorem 7.** *Let $p \in \Delta_{[k]}$ be a power-law distribution with power $\alpha$. Then,*

$$\mathcal{H}_n(p) \simeq \begin{cases} k & \text{if } \alpha > \frac{k^{1+\alpha}}{n} \vee 1 \text{ or } 1 \geq \alpha > \frac{k^2}{n}, \\ n^{\frac{1}{\alpha+1}} & \text{if } \frac{k^{1+\alpha}}{n} \geq \alpha > 1, \\ \left(\frac{n}{k^{1-\alpha}}\right)^{\frac{1}{1+\alpha}} & \text{if } \frac{k^2}{n} \wedge 1 \geq \alpha > \frac{k^{1-\alpha}}{n}, \\ \frac{n}{k^{1-\alpha}} - \frac{n}{k} & \text{if } \frac{k^{1-\alpha}}{n} \wedge 1 \geq \alpha \text{ and } \alpha \geq 2\log_k\left(7\sqrt{\frac{k}{n}} + 1\right), \\ k \wedge \sqrt{\frac{n}{k^{1-\alpha}}} & \text{if } \frac{k^{1-\alpha}}{n} \wedge 1 \geq \alpha \text{ and } 2\log_k\left(7\sqrt{\frac{k}{n}} + 1\right) > \alpha. \end{cases}$$

*In particular, as $\alpha \to 0$, the bound degenerates to $k \wedge \sqrt{\frac{n}{k}}$, which is at most $n^{\frac{1}{3}}$.*

Since a power-law sample profile is completely specified by $\alpha$, $k$, and $n$, the above theorem directly applies to model parameter estimation. Specifically, we first compute $\mathcal{D}_n \sim p$, which is a simple function of the symbol counts. By Theorem 1, we can then use it to approximate $\mathcal{H}_n(p)$. Finally, we utilize the characterization theorem and find the parameter relations (testing might be necessary).

**Histogram** While histogram is among the most widely studied representations, histogram distributions' importance also rises with the rapid growth of data sizes in modern scientific applications. For example, *subsampling*, a generic strategy to handle large datasets, naturally induces a histogram distribution over different categories of the data. This induced distribution often summarizes vital data statistics, leveraging which yields efficient and flexible inference procedures.

Formally, a discrete distribution $p \in \Delta_{\mathbb{Z}}$ is a *t-histogram* if we can partition its support into at most $t$ pieces such that $p$ takes the same probability value over each piece. The theorem below provides near-optimal bounds on the profile entropy of the $t$-histogram distributions.

**Theorem 8.** *Denote by $\mathcal{I}_t \subseteq \Delta_{\mathbb{Z}}$ the collection of t-histogram distributions. Then,*

$$\max_{p \in \mathcal{I}_t} \mathcal{H}_n(p) \simeq \min\left\{(nt^2)^{\frac{1}{3}}, \sqrt{n}\right\}.$$

In practical settings, the value of $t$ is often poly-logarithmic in $n$, and the bound reduces to $\tilde{\mathcal{O}}(n^{1/3})$. For the particular case of $t = 1$, distribution $p$ is uniform over some unknown contiguous support. This result overlaps with Theorem 7 with $\alpha = 0$, yielding the following bound.

**Corollary 3.** *For any uniform distribution $p$ with support size $k$, we have $\mathcal{H}_n(p) \simeq k \wedge \sqrt{\frac{n}{k}}$.*

## 3 Applications and Extensions

**Robust learning** The profile of any sequence is invariant to domain-symbol permutations. Since entropy is a symmetric property, the profile entropy of an i.i.d. sample is also permutation invariant. Consequently, a result in this paper that holds for a distribution will also hold for *any distributions possessing the same probability multiset*. For numerous practical applications, this *robustness to symbol permutation* is a desirable and novel notion of robustness that particularly resides in discrete domains, as samples often come as categorical data, while the alphabet ordering for the underlying distribution to exhibit certain structure is frequently unknown [Hao and Orlitsky, 2019b].

For example, the sample may consist of different fruits, not integers. But suppose there is a hidden mapping from the fruit domain to integers that makes the distribution log-concave over $\mathbb{Z}$. Then, all our results such as Theorem 2, 4, 5, and 6 are in effect. For another example, in natural language processing, we observe words and punctuation marks. Even we know that observations come from a

power-law distribution [Mitzenmacher, 2004], it is often unclear how to order the alphabet to realize such a condition. The robustness of our approach again enables us to achieve a variety of learning objectives, such as understanding the relation between different model parameters (Theorem 7).

**Mixture models**   The results in Section 2.5 provide optimal characterization for simple structured families. A standard extension to incorporate more complex structures in the model is spanning a distribution family by including (weighted) mixtures. A typical example is the Gaussian mixture model, which is among the most widely studied probabilistic models.

In the supplementary material, we present such results for all three families in Section 2.5, and for mixtures of discretized high-dimensional Gaussians. In fact, we obtain a simple and intuitive profile-entropy characterization for all distributions. Partition the unit interval into a sequence of ranges, $I_j := \left( (j-1)^2 \frac{\log n}{n}, \ j^2 \frac{\log n}{n} \right], 1 \le j \le \sqrt{\frac{n}{\log n}}$, and for any distribution $p$, denote by $p_{I_j}$ the number of probabilities in $I_j$. Then,

**Lemma 1.** *For any $n \in \mathbb{Z}^+$ and $p \in \Delta$, we have $\mathcal{H}_n(p) \simeq \sum_{j \ge 1} \min \left\{ p_{I_j}, j \cdot \log n \right\}$.*

**Competitive property estimation**   Theorem 2 on PML holds for every distribution, any symmetric property, and distribution collection, such as a finite-dimensional simplex, regardless of other parameters such as the alphabet size. To the best of our knowledge, this is one of the most general results in the field. Below we provide a basic example for its applications.

For an arbitrary $\beta > 0$, let $f$ be the order-$\beta$ Rényi entropy, and P be the set of distributions whose probability multisets correspond to power-laws with power $\alpha \ge 3$. The minimax error rate $r_m(\mathrm{P})$ is unknown for this problem as recent works (e.g., [Acharya et al., 2016]) mainly focused on the standard simplexes. On the other hand, Theorem 4, together with Theorem 7, shows that the $n$-sample PML plug-in estimator essentially performs as well as the best $n^{3/4}$-sample estimator. Note that while the guarantee of PML uniformly holds for all $\beta$, the best estimator can optimize its performance for every $\beta$. Following the same rationale, we can derive such nontrivial competitive estimation results for numerous properties and distribution families without having to analyze them in detail.

**Adaptive testing and classification**   Profile entropy also directly connects to adaptive testing and classification. Such a connection arises from computing the *profile probability* [Acharya et al., 2011, 2012], the probability of observing the sample's profile under the same sampling process.

Specifically, the first paper designs an algorithm that distinguishes two unknown distributions using near-optimal sample sizes whenever the optimal algorithm has an exponentially small error probability. In addition, the algorithm is simply a ratio test between the probabilities of two profiles. Given sample $X^n \sim p$ over a finite domain, we can compute its profile probability in $\exp(\Theta(\mathcal{H}_n(p)))$ operations. For example, if the underlying distribution is a 4-histogram, then by Theorem 8, the running time exponent is of order $n^{1/3}$. The result follows by the equivalence of the problem and computing the permanent of a rank-$\mathcal{D}_n$ matrix [Barvinok, 1996, Vontobel, 2012, 2014, Barvinok, 2016].

**Method of types**   We connect our approach to *the method of types*, an important technical tool in Shannon theory and many other fields [Csiszar and Körner, 2011, Wolfowitz, 2012]. In the notation of this paper, the *type* of a sequence $x^n$ over some finite domain $\mathcal{X}$ is the ordered list of multiplicities $\mu_y(x^n)$, which associates symbol $y$ with its number of appearances in $x^n$. For this multiplicity list, the method of types associates each $\mu_y(x^n)$ with the number of symbols having this multiplicity, which is precisely $\varphi_{\mu_y(x^n)}(x^n)$. Hence, the profile of a sequence is *the type of its type*.

Given the above arguments, understanding the deep connection between profile-based algorithms and the method of types is a meaningful future research direction to explore.

## Broader Impact

Classical information theory states that an i.i.d. sample contains $H(X^n \sim p) = nH(p)$ information, which provides little insight for statistical applications. We present a different view by decomposing the sample information into three parts: the labeling of the profile elements, ordering of them, and profile entropy. With no bias towards any symbols, the *profile entropy* rises as a fundamental measure unifying the concepts of estimation, inference, and compression. We believe this view could help researchers in information theory, statistical learning theory, and computer science communities better understand the information composition of i.i.d. samples over discrete domains.

The results established in this work are general and fundamental, and have numerous applications in privacy, economics, data storage, supervised learning, etc. A potential downside is that the theoretical guarantees of the associated algorithms rely on the assumption correctness, e.g., the domain should be discrete and the sampling process should be i.i.d. . In other words, it will be better if users can confirm these assumptions by prior knowledge, experiences, or statistical testing procedures. Taking a different perspective, we think a potential research direction following this work is to extend these results to Markovian models, making them more robust to model misspecification.

## Acknowledgments and Disclosure of Funding

We want to thank the anonymous reviewers and meta-reviewer for their insightful and valuable suggestions and feedback. Funding in direct support of this work: NSF grants CIF-1564355 and CIF-1619448. Additional revenues related to this work: Not applicable.

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
