[Supplementary Material]

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

**Appendix orgnization**  In the appendix, we order the results and proofs according to their logical priority. In other words, the proof of a theorem or lemma mainly relies on preceding results. For the ease of reference, the numbering of the theorems is consistent with that in the main paper.

# A    Entropy and Dimension of Sample Profiles

Consider an arbitrary sequence $x^n$ over a finite or countably infinite alphabet $\mathcal{X}$. The *multiplicity* $\mu_y(x^n)$ of a symbol $y \in \mathcal{X}$ is the number of times $y$ appears in $x^n$. The *prevalence* of an integer $\mu$ is the number $\varphi_\mu(x^n)$ of symbols in $x^n$ with multiplicity $\mu$. The *profile* of $x^n$ is the multiset $\varphi(x^n)$ of multiplicities of the symbols in $x^n$. We refer to it as a profile of *length* $n$.

The number $\mathcal{D}(S)$ of distinct elements in a multiset $S$ is its *dimension*. For convenience, we also write $\mathcal{D}(x^n)$ for profile dimension. The dimension of a length-$n$ profile over $\mathcal{X}$ is at most $\min\{\sqrt{2n}, |\mathcal{X}|\}$.

Let $\Delta$ be the collection of all discrete distributions, and $\Delta$ be the collection of those over $\mathcal{X}$. Draw a size-$n$ sample $X^n$ from an arbitrary distribution in $p \in \Delta$. Then, the profile $\Phi^n$ of $X^n$ is a random multiset whose distribution depends on only $p$ and $n$. We therefore write $\Phi^n \sim p$, and call $\mathcal{H}_n(p) := H_n(p)$ the *profile entropy* with respect to $(p, n)$. Analogously, we call $\mathcal{D}_n := \mathcal{D}_n$, the *profile dimension* associated with $(p, n)$, and write $\mathcal{D}_n \sim p$.

Consider an arbitrary sequence $x^n$ over a finite or countably infinite alphabet $\mathcal{X}$. The *multiplicity* $\mu_y(x^n)$ of a symbol $y \in \mathcal{X}$ is the frequency of $y$ in $x^n$. The *prevalence* of an integer $\mu$ is the number $\varphi_\mu(x^n)$ of symbols in $x^n$ with multiplicity $\mu$. The *profile* of $x^n$ is the multiset $\varphi(x^n)$ of multiplicities of the symbols in $x^n$, which we describe as a profile of *length* $n$.

## A.1    Concentration of Profile Dimension

First we express the dimension of a sample profile in terms of the symbol multiplicities. Denote by $\bigvee$ the logical OR operator. For any distribution $p$ and $X^n \sim p$,

$$\mathcal{D}_n = \sum_{\mu=1}^{n} \bigvee_{x \in \mathcal{X}} \mathbb{1}_{\mu_x(X^n)=\mu}.$$

The statistical dependency landscape of terms in the summation is rather complex, since $\mu_x(X^n)$ and $\mu_y(X^n)$ are dependent for every $(x, y)$ pair due to the fixed sample size; and so are $\mathbb{1}_{\mu_x(X^n)=\mu_1}$ and $\mathbb{1}_{\mu_x(X^n)=\mu_2}$ for every pair of distinct $\mu_1$ and $\mu_2$. To simplify the derivations, we relate this quantity to its variant under the *Poisson sampling scheme*, i.e., making the sample size an independent $N \sim \mathrm{Poi}(n)$. Specifically, define

$$\tilde{\mathcal{D}}_N := \tilde{\mathcal{D}}(X^N) := \sum_{U=1}^{n} \bigvee_{x \in \mathcal{X}} \mathbb{1}_{\mu_x(X^N)=U}.$$

Note that this is not the same as $\mathcal{D}_N$ since the summation index goes up only to $n$.

Denote the expected value of $\tilde{\mathcal{D}}_N$ by $E_n(p)$, which will frequently appear in the rest discussions. Our first result shows that the original $\mathcal{D}_n$ satisfies a Chernoff-Hoeffding type bound centered at $E_n(p)$.

**Theorem 9.** *Under the above conditions and for any $n \in \mathbb{Z}^+$, $p \in \Delta$, and $\gamma > 0$,*

$$\Pr\left(\frac{\mathcal{D}_n}{1+\gamma} \geq E_n(p)\right) \leq 3\sqrt{n}e^{-\min\{\gamma^2, \gamma\}E_n(p)/3},$$

*and for any $\gamma \in (0, 1)$,*

$$\Pr\left(\frac{\mathcal{D}_n}{1-\gamma} \leq E_n(p)\right) \leq 3\sqrt{n}e^{-\gamma^2 E_n(p)/2}.$$

*Proof.* A nice attribute of Poisson sampling is that all the multiplicities $\mu_y(X^n)$ are independent of each other. We will first consider $\mathcal{D}_N$ and relate it to the fixed-sample-size version later.

For simplicity and clarity, we suppress $X^n$ in $\mu_y(X^n)$ and write $\nu_y$ instead of $\mu_y$ when the multiplicity is obtained through Poisson sampling. For any $i \in [n]$, denote $G_i(\{\nu_x\}_x) := \bigvee_{x \in \mathcal{X}} \mathbb{1}_{\nu_x=i}$. As

mentioned previously, instead of analyzing $\mathcal{D}_N$, we consider

$$\tilde{\mathcal{D}}_N = \sum_{i=1}^{n} \bigvee_{x \in \mathcal{X}} \mathbb{1}_{\nu_x = i} = \sum_{i=1}^{n} G_i(\{\nu_x\}_x).$$

Note that for any disjoint $I, J \subseteq [n]$, the functions $\sum_{i \in I} G_i(\{\nu_x\}_x)$ and $\sum_{j \in J} G_j(\{\nu_x\}_x)$ are discordant monotone by each argument, namely, when we increase the value of each $\nu_x$, the increase in the value of one function implies the non-increase of the other. Then, by the results in Lehmann [1966], the values of the two functions, when viewed as random variables, are negatively associated.

Next we show that quantity $\tilde{\mathcal{D}}_N$ satisfies a Chernoff-type bound.

Let $\gamma$ be an arbitrary positive number. Note that $G_i$ is a Bernoulli random variable with parameter

$$q_i := \mathbb{E}\left[G_i(\{\nu_x\}_x)\right].$$

Then for the expected value of $\tilde{\mathcal{D}}_N$, we have

$$E_n(p) := \mathbb{E}\left[\tilde{\mathcal{D}}_N\right] = \mathbb{E}\left[\sum_{i=1}^{n} G_i(\{\nu_x\}_x)\right] = \sum_i q_i.$$

For simplicity, temporally write $Y := \tilde{\mathcal{D}}_N$ and $\mu := E_n(p)$. Then, by Markov's inequality and the monotonicity of function $e^{ty}$ over $t > 0$,

$$\Pr\left(Y \geq (1+\gamma)\mu\right) = \Pr\left(e^{tY} \geq e^{t(1+\gamma)\mu}\right) \leq \frac{\mathbb{E}[e^{tY}]}{e^{t(1+\gamma)\mu}}.$$

It suffices to bound $\mathbb{E}[e^{tY}]$ by a function of other parameters.

$$
\begin{aligned}
\mathbb{E}[e^{tY}] &\stackrel{(a)}{=} \mathbb{E}\left[\exp\left(t\left(\sum_{i=1}^{n} G_i(\{M_x\}_x)\right)\right)\right] \\
&\stackrel{(b)}{=} \mathbb{E}\left[\exp\left(tG_1(\{M_x\}_x)\right) \cdot \exp\left(t\left(\sum_{i=2}^{n} G_i(\{M_x\}_x)\right)\right)\right] \\
&\stackrel{(c)}{\leq} \mathbb{E}\left[\exp\left(tG_1(\{M_x\}_x)\right)\right] \cdot \mathbb{E}\left[\exp\left(t\left(\sum_{i=2}^{n} G_i(\{M_x\}_x)\right)\right)\right] \\
&\stackrel{(d)}{\leq} \prod_{i=1}^{n} \mathbb{E}\left[\exp\left(tG_i(\{M_x\}_x)\right)\right] \stackrel{(e)}{=} \prod_{i=1}^{n}\left(1 + q_i(e^t - 1)\right) \\
&\stackrel{(f)}{\leq} \prod_{i=1}^{n}\left(\exp\left(q_i(e^t - 1)\right)\right) \stackrel{(g)}{=} \exp\left(\sum_{i=1}^{n} q_i(e^t - 1)\right) \\
&\stackrel{(h)}{=} \exp\left((e^t - 1)\mu\right),
\end{aligned}
$$

where $(a)$ follows by the definition of $Y$; $(b)$ follows by $e^{a+b} = e^a \cdot e^b$; $(c)$ follows by the fact that $G_1$ is negatively associated with $\sum_{i=2}^{n} G_i$; $(d)$ follows by an induction argument via negative association; $(e)$ follows by the fact that $G_i$ is a Bernoulli random variable with mean $q_i$; $(f)$ follows by the inequality $1 + x \leq e^x, \forall x \geq 0$; $(g)$ follows by $e^a \cdot e^b = e^{a+b}$; and $(h)$ follows by $\mu = \sum_i q_i$.

Applying standard simplifications, we obtain

$$\Pr\left(Y \geq (1+\gamma)\mu\right) \leq e^{-\min\{\gamma^2, \gamma\}\mu/3}, \ \forall \gamma > 0,$$

and

$$\Pr\left(Y \leq (1-\gamma)\mu\right) \leq e^{-\gamma^2\mu/2}, \ \forall \gamma \in (0,1).$$

The proof will be complete upon noting that: 1) the probability that $N = n$ is at least $1/(3\sqrt{n})$; 2) conditioning on $N = n$ transforms the sampling model to that with a fixed sample size $n$. $\qquad\square$

As a corollary, the value of $\mathcal{D}_n$ is often close to $E_n(p)$.

**Corollary 4.** *Under the same conditions as above and for any $n \in \mathbb{Z}^+$, $p \in \Delta$, with probability at least $1 - 6/\sqrt{n}$,*

$$\frac{1}{2}E_n(p) - 4\log n \le \mathcal{D}_n \le 2E_n(p) + 3\log n.$$

*Proof.* To establish the lower bound, note that if $E_n(p) \ge 3\log n$, setting $\gamma = 1$ in Theorem 9 yields

$$\Pr\left(\mathcal{D}_n \ge 2E_n(p) + 3\log n\right) \le \Pr\left(\mathcal{D}_n \ge 2E_n(p)\right) \le 3\sqrt{n}e^{-E_n(p)/3} \le \frac{3}{\sqrt{n}},$$

else if $E_n(p) < 3\log n$, setting $\gamma = (3\log n)/E_n(p)$ yields

$$\Pr\left(\mathcal{D}_n \ge 2E_n(p) + 3\log n\right) \le \Pr\left(\mathcal{D}_n \ge E_n(p) + 3\log n\right) \le 3\sqrt{n}e^{-(3\log n)/3} = \frac{3}{\sqrt{n}}.$$

As for the upper bound, if $E_n(p) \ge 8\log n$,

$$\Pr\left(\mathcal{D}_n + 4\log n \le \left(1 - \frac{1}{2}\right)E_n(p)\right) \le \Pr\left(\mathcal{D}_n \le \left(1 - \frac{1}{2}\right)E_n(p)\right) \le 3\sqrt{n}e^{-\mu/8} \le \frac{3}{\sqrt{n}},$$

and for any $E_n(p) < 8\log n$,

$$\Pr\left(\mathcal{D}_n + 4\log n \le \left(1 - \frac{1}{2}\right)E_n(p)\right) \le \Pr\left(\mathcal{D}_n < 0\right) = 0 \le \frac{3}{\sqrt{n}}.$$

Combining these tail bounds through the union bound completes the proof. $\qquad\square$

In addition to the above, we establish an Efron-Stein type inequality.

**Theorem 10.** *For any distribution $p$ and $\mathcal{D}_n \sim p$,*

$$\mathrm{Var}(\mathcal{D}_n) \le \mathbb{E}[\mathcal{D}_n].$$

*Proof.* First, note that for any $j, t \in [n]$ and $j \ne t$,

$$\begin{aligned}
C_{j,t} &:= \mathrm{Cov}\left(\mathbb{1}_{\varphi_j(X^n)>0}, \mathbb{1}_{\varphi_t(X^n)>0}\right) \\
&= \Pr\left(\varphi_j(X^n), \varphi_t(X^n) > 0\right) - \Pr\left(\varphi_j(X^n) > 0\right) \cdot \Pr\left(\varphi_t(X^n) > 0\right) \\
&= \left(\Pr\left(\varphi_j(X^n) > 0 | \varphi_t(X^n) > 0\right) - \Pr\left(\varphi_j(X^n) > 0\right)\right) \cdot \Pr\left(\varphi_t(X^n) > 0\right) \\
&= \left(\Pr\left(\varphi_j(X^n) > 0 | \varphi_t(X^n) > 0\right) - \Pr\left(\varphi_j(X^n) > 0 | \varphi_t(X^n) = 0\right)\right) \\
&\quad \times \Pr\left(\varphi_t(X^n) = 0\right) \cdot \Pr\left(\varphi_t(X^n) > 0\right) \\
&\le 0
\end{aligned}$$

Therefore, the variance of the profile dimension $\mathcal{D}_n$ satisfies

$$\begin{aligned}
\mathrm{Var}\left(\mathcal{D}_n\right) &= \mathrm{Var}\left(\sum_{i=1}^n \mathbb{1}_{\varphi_i(X^n)>0}\right) \\
&\le \sum_{i=1}^n \mathrm{Var}\left(\mathbb{1}_{\varphi_i(X^n)>0}\right) + \sum_{j \ne t}\mathrm{Cov}\left(\mathbb{1}_{\varphi_j(X^n)>0}, \mathbb{1}_{\varphi_t(X^n)>0}\right) \\
&\le \sum_{i=1}^n \mathbb{E}\left[\mathbb{1}_{\varphi_i(X^n)>0}\right] + \sum_{j \ne t} C_{j,t} \\
&\le \sum_{i=1}^n \mathbb{E}\left[\mathbb{1}_{\varphi_i(X^n)>0}\right] \\
&= \mathbb{E}\left[\mathcal{D}_n\right]. \qquad\qquad\qquad\qquad\qquad\qquad\qquad\square
\end{aligned}$$

## A.2  Theorem 1: Dimension-Entropy Equivalence

The following theorem shows that for every distribution and sampling parameter $n$, the induced profile entropy and dimension are of the same order, with high probability.

**Theorem 1** (Entropy-dimension equivalence). *For any distribution $p \in \Delta$ and $\mathcal{D}_n \sim p$,*

$$\Pr(\mathcal{D}_n \simeq \mathcal{H}_n(p)) \geq 1 - \frac{1}{\sqrt{n}}.$$

## A.3  Proof of Theorem 1

**Proof outline**  We decompose the proof of the theorem into three steps.

First, we show $\mathcal{H}_n(p) \lesssim \mathcal{D}_n$ with high probability, which is a consequence of Theorem 9 (which shows that $\mathcal{D}_n$ highly concentrates around its expectation) and Shannon's source coding theorem. Second, we introduce a simple quantity $\mathcal{H}_n^{\mathcal{S}}(p)$ that approximates the expectation of $\mathcal{D}_n$ to within logarithmic factors of $n$. Finally, leveraging this approximation guarantee, we establish the other direction of the theorem. This step is more involved due to the aforementioned complications.

**Step 1: Bounding Profile Entropy by Its Dimension**

By the tail bounds (Theorem 9) and trivial lower bound of $1$ on the profile dimension, with probability at least $1 - 1/\sqrt{n}$, the expectation of $\mathcal{D}_n$ satisfies

$$\mathbb{E}[\mathcal{D}_n] \lesssim \mathcal{D}_n.$$

By our result on block profile compression (Section 2.4), storing profile $\Phi^n \sim p$ losslessly takes

$$\mathcal{O}(\log n) \cdot \mathbb{E}[\mathcal{D}_n] + \mathcal{O}\left(\frac{1}{\sqrt{n}}\right) \cdot \log \mathbb{P}(n) \lesssim \mathbb{E}[\mathcal{D}_n]$$

nats space in expectation. By Shannon's source coding theorem, the expected space to losslessly storing a random variable is at least its entropy. Hence, with probability at least $1 - \mathcal{O}(1/\sqrt{n})$,

$$\mathcal{H}_n(p) \lesssim \mathbb{E}[\mathcal{D}_n] \lesssim \mathcal{D}_n.$$

Applying $\mathcal{D}_n \geq 1$ completes the proof.

**Step 2: Simple Approximation Formula for Profile Dimension**

Next, we show that $\mathcal{H}_n(p) \gtrsim \mathcal{D}_n$, with high probability. Note that $\mathcal{D}_n \sim p$ is often close to $E_n(p)$, the expectation of its Poissonized version $\tilde{\mathcal{D}}_N$, with an exponentially small deviation probability. Hence, to approximate $\mathcal{D}_n$, it suffices to accurately compute $E_n(p)$.

By independence and the linearity of expectations,

$$E_n(p) = \mathbb{E}[\tilde{\mathcal{D}}_N] = \sum_{i=1}^n \left(1 - \prod_{x \in \mathcal{X}} \left(1 - e^{-np_x} \frac{(np_x)^i}{i!}\right)\right).$$

The expression is exact but does not relate to $p$ in a simple manner. For an intuitive approximation, we partition the unit interval into a sequence of ranges,

$$I_j := \left((j-1)^2 \frac{\log n}{n}, \, j^2 \frac{\log n}{n}\right], 1 \leq j \leq \sqrt{\frac{n}{\log n}},$$

denote by $p_{I_j}$ the number of probabilities $p_x$ belonging to $I_j$, and relate $E_n(p)$ to an induced shape-reflecting quantity,

$$\mathcal{H}_n^{\mathcal{S}}(p) := \sum_{j \geq 1} \min \left\{p_{I_j}, j \cdot \log n\right\},$$

the sum of the effective number of probabilities lying within each range [Hao and Orlitsky, 2019b]. To compute $\mathcal{H}_n^{\mathcal{S}}(p)$, we simply count the number of probabilities in each $I_j$. Our main result shows that $\mathcal{H}_n^{\mathcal{S}}(p)$ well approximates $E_n(p)$ over the entire $\Delta$, up to logarithmic factors of $n$.

**Theorem 11.** *For any $n \in \mathbb{Z}^+$ and $p \in \Delta$,*

$$\frac{1}{\sqrt{\log n}} \cdot \Omega(\mathcal{H}_n^{\mathcal{S}}(p)) \leq E_n(p) \leq \mathcal{O}(\mathcal{H}_n^{\mathcal{S}}(p)).$$

*Proof.* The fact that $\mathcal{O}(H_n^{\mathcal{S}}(p))$ upperly bounds $\mathbb{E}[\tilde{\mathcal{D}}_N]$ simply follows by the concentration of Poisson variables, and is established in Hao and Orlitsky [2019b]. Below we show that the quantity also serves as a lower bound. By construction, for any given sampling parameter $n$, index $j$, and symbol $x$ with probability $p_x \in I_j$, the corresponding symbol multiplicity $\mu_x \sim \text{Poi}(np_x)$. Hence, we can express the expectation of $\tilde{\mathcal{D}}_N$ as

$$
\begin{aligned}
\mathbb{E}\left[\tilde{\mathcal{D}}_N\right] &= \mathbb{E}\left[\sum_{i=1}^n \bigvee_x \mathbb{1}_{\mu_x = i}\right] \\
&= \sum_{i=1}^n \mathbb{E}\left[1 - \bigwedge_x \mathbb{1}_{\mu_x \neq i}\right] \\
&= \sum_{i=1}^n \left(1 - \mathbb{E}\left[\prod_x \mathbb{1}_{\mu_x \neq i}\right]\right) \\
&= \sum_{i=1}^n \left(1 - \prod_x \mathbb{E}\left[\mathbb{1}_{\mu_x \neq i}\right]\right) \\
&= \sum_{i=1}^n \left(1 - \prod_x \left(1 - e^{-np_x}\frac{(np_x)^i}{i!}\right)\right).
\end{aligned}
$$

This proves the aforementioned formula. Then, for every sufficiently large index $j$ and $i \in S_j := [(j-1)^2, j^2] \log n$, define a sequence of intervals,

$$I_j^i := \frac{i}{n} + [-j, j] \frac{\sqrt{\log n}}{n}.$$

Then for any $i \in S_j$ and $p_x \in I_j^i \cap I_j$, the corresponding Poisson probability satisfies

$$
\begin{aligned}
e^{-np_x}\frac{(np_x)^i}{i!} &= e^{-i}\frac{i^i}{i!} \cdot \left(e^{i-np_x} \cdot \frac{(np_x)^i}{i^i}\right) \\
&= e^{-i}\frac{i^i}{i!} \cdot \left(e^{-(np_x-i)} \cdot \left(1 + \frac{np_x - i}{i}\right)^i\right) \\
&= e^{-i}\frac{i^i}{i!} \cdot \exp\left(-(np_x - i) + i \cdot \log\left(1 + \frac{np_x - i}{i}\right)\right) \\
&\geq \frac{1}{3\sqrt{i}} \cdot \exp\left(-\frac{2i}{3} \cdot \left(\frac{np_x - i}{i}\right)^2\right) \\
&\geq \frac{1}{9\sqrt{i}} \geq \frac{1}{9j\sqrt{\log n}}.
\end{aligned}
$$

Now we analyze the contribution of indices $i \in S_j$ to the expected value of $\tilde{\mathcal{D}}_N$. For clarity, we divide our analysis into two cases: $p_{I_j} \geq j \log n$ and $p_{I_j} < j \log n$.

Consider the collection $\mathcal{P}_j$ of probabilities $p_x \in I_j$, and the collection $\mathcal{I}_j$ of intervals $I_j^i, i \in S_j$. By construction, each probability in $\mathcal{P}_j$ is contained in at least $j\sqrt{\log n}$ many intervals in $\mathcal{I}_j$. Hence the total number of probabilities (repeatedly counted) included in $\mathcal{I}_j$ is at least $p_{I_j} \cdot j\sqrt{\log n}$. Note that the number of intervals in $\mathcal{I}_j$ is less than $2j \log n$. We claim that there exists one (or more) interval $I_j^{i'} \in \mathcal{I}_j$ containing at least $p_{I_j}/(2\sqrt{\log n})$ probabilities. By construction, there are at least $j\sqrt{\log n}/2$ neighboring intervals of $I_j^{i'}$ that contain at least $p_{I_j}/(4\sqrt{\log n})$ probabilities. The

contribution of these these intervals to the expected value of $\tilde{\mathcal{D}}_N$ is at least $j\sqrt{\log n}/2$ times

$$1 - \left(1 - \frac{1}{9j\sqrt{\log n}}\right)^{\frac{p_{I_j}}{4\sqrt{\log n}}} \geq 1 - \exp\left(\frac{p_{I_j}}{4\sqrt{\log n}} \log\left(1 - \frac{1}{9j\sqrt{\log n}}\right)\right)$$

$$\geq 1 - \exp\left(-\frac{p_{I_j}}{40 j \log n}\right)$$

$$\geq \Theta\left(\frac{p_{I_j}}{j \log n}\right),$$

where the last step holds if $p_{I_j} \leq j \log n$. This yields a lower bound of $\Theta(p_{I_j}/\sqrt{\log n})$.

It remains to consider the $p_{I_j} > j \log n$ case. Again, the total number of probabilities included in $\mathcal{I}_j$ is at least $p_{I_j} \cdot j\sqrt{\log n}$. Furthermore, each interval $I_j^i$ contains at most $p_{I_j}$ probabilities and there are less than $2j \log n$ intervals. Therefore, the number of intervals that contain at least $j\sqrt{\log n}/4$ probabilities is at least $j\sqrt{\log n}/2$. Otherwise, the number of probabilities included in $\mathcal{I}_j$ is less than

$$\frac{j\sqrt{\log n}}{4} \cdot 2j \log n + p_{I_j} \cdot \frac{j\sqrt{\log n}}{2} \leq p_{I_j} \cdot j\sqrt{\log n},$$

which leads to a contradiction. Analogously, the contribution of these these intervals to the expected value of $\tilde{\mathcal{D}}_N$ is at least $j\sqrt{\log n}/2$ times

$$1 - \left(1 - \frac{1}{9j\sqrt{\log n}}\right)^{\frac{j\sqrt{\log n}}{4}} \geq 1 - \exp\left(\frac{j\sqrt{\log n}}{4} \log\left(1 - \frac{1}{9j\sqrt{\log n}}\right)\right)$$

$$\geq 1 - \exp\left(-\frac{1}{40}\right)$$

$$= \Theta(1),$$

which yields a lower bound of $\Theta(j\sqrt{\log n})$ on the expected value of $\tilde{\mathcal{D}}_N$.

Consolidating the previous results shows that

$$\mathbb{E}\left[\tilde{\mathcal{D}}_N\right] \geq \frac{1}{\sqrt{\log n}} \cdot \Omega(\sum_{j \geq 1} \min\left\{p_{I_j}, j \cdot \log n\right\}). \qquad \square$$

**Step 3: Bounding Profile Dimension by Its Entropy**

Next, we establish that for any distribution $p \in \Delta$, $\Phi^n \sim p$, with probability at least $1 - 1/\sqrt{n}$,

$$\mathcal{H}_n(p) \gtrsim \mathcal{D}_n.$$

Let $p$ be an arbitrary distribution in $\Delta$. Recall that we partition the interval $(0, 1]$ into a sequence of sub-intervals,

$$I_j := \left((j-1)^2 \frac{\log n}{n}, j^2 \frac{\log n}{n}\right], \quad 1 \leq j \leq \sqrt{\frac{n}{\log n}},$$

and denote by $p_{I_j}$ the number of probabilities $p_x$ in $I_j$.

Our current objective is to bound $H(\Phi^n \sim p)$ from below by a nontrivial multiple of $H_n^{\mathcal{S}}(p)$. For simplicity of derivations, we will adopt the standard Poisson sampling scheme and make the sample size an independent Poisson variable $N \sim \mathrm{Poi}(n)$. For notational simplicity, we will suppress $X^N$ in all the expressions and write the profile as $\varphi := \Phi^N$ by slightly abusing the notation.

Note that the profile can be equivalently expressed as a length-$n$ vector

$$\varphi = (\varphi_1, \ldots, \varphi_n),$$

where $\varphi_i$ denotes the number of symbols appearing exactly $i$ times.

For a sufficiently large absolute constant $c$, decompose $\varphi$ into $c$ parts according to $I_j$ such that the $t$-th part ($t = 1, \ldots, c$) consists of $\varphi_i$'s satisfying $i \in nI_j$ with $j \equiv t \mod c$. Since by definition,

$$H_n^{\mathcal{S}}(p) = \sum_{j \geq 1} \min\{p_{I_j}, j \cdot \log n\},$$

one of the $c$ parts corresponds to a partial sum of at least $H_n^{\mathcal{S}}(p)/c$. Without loss of generality, we assume that it is the second part, i.e.,

$$\sum_{j\equiv 1 \bmod c} \min\{p_{I_j}, j \cdot \log n\} \geq \frac{H_n^{\mathcal{S}}(p)}{c}.$$

Apply standard Poisson tail probability bounds. For example,

**Lemma 2.** *Let $Y$ be a Poisson or binomial random variable with mean value $\lambda$. Then,*

$$\Pr(X \leq \lambda(1-\delta)) \leq \exp\left(-\frac{\delta^2\lambda}{2}\lambda\right), \quad \forall \delta \in [0,1],$$

*and*

$$\Pr(X \geq \lambda(1+\delta)) \leq \exp\left(-\frac{\delta^2\lambda}{2+2\delta/3}\right), \quad \forall \delta \geq 0.$$

For any $j \equiv 1 \bmod c$ and with probability at least $1 - 1/n^4$, one can express the truncated profile $(\varphi_i)_{i\in nI_j}$ over $I_j$ as a function of $\mu_x$ for $x$ satisfying $np_x \in I_{j'}, j' \in (j - c/2, j + c/2)$.

Basically, this says that for every $x$, the number of its appearance is not too far away from the expected value. By the union bound, this is true for all $j \equiv 1 \bmod c$ with probability at least $1 - 1/n^3$, as $j$ can take only $n$ possible values. Denote the last event by $A$.

To proceed, we recall the formula of Hardy and Ramanujan [1918] on the number $\mathbb{P}(n)$ of integer partitions of $n$, which happens to equal the number of length-$n$ profiles:

$$\log \mathbb{P}(n) = 2\pi\sqrt{\frac{n}{6}}(1 + o(1)).$$

Below, we will use a weaker version that works for any $n$:

$$\log \mathbb{P}(n) \leq \sqrt{3n}.$$

Then, conditioning on $A$, the truncated profiles $(\varphi_i)_{i\in nI_j}$ for $j \equiv 1 \bmod c$ are independent. Since conditioning reduces entropy,

$$
\begin{aligned}
H(\varphi) &\geq H((\varphi_i)_{i\in nI_j, j\equiv 1 \bmod c}) \\
&\geq H((\varphi_i)_{i\in nI_j, j\equiv 1 \bmod c}|\mathbb{1}_A) \\
&\geq H((\varphi_i)_{i\in nI_j, j\equiv 1 \bmod c}|\mathbb{1}_A = 1) \cdot \Pr(A) \\
&= \sum_{j\equiv 1 \bmod c} H((\varphi_i)_{i\in nI_j}|\mathbb{1}_A = 1) \cdot \Pr(A) \\
&= \sum_{j\equiv 1 \bmod c} H((\varphi_i)_{i\in nI_j}|\mathbb{1}_A) - \sum_{j\equiv 1 \bmod c} H((\varphi_i)_{i\in nI_j}|\mathbb{1}_A = 0) \cdot (1 - \Pr(A)) \\
&\geq \sum_{j\equiv 1 \bmod c} (H((\varphi_i)_{i\in nI_j}) - H(\mathbb{1}_A)) - \frac{1}{n^3} \sum_{j\equiv 1 \bmod c} H((\varphi_i)_{i\in nI_j}|\mathbb{1}_A = 0) \\
&\geq -nH(\mathbb{1}_A) + \sum_{j\equiv 1 \bmod c} H((\varphi_i)_{i\in nI_j}) - \frac{1}{n^3} \cdot n \cdot \log(\exp(\Theta(\sqrt{n}))) \\
&= -\mathcal{O}\left(\frac{1}{\sqrt{n}}\right) + \sum_{j\equiv 1 \bmod c} H((\varphi_i)_{i\in nI_j}),
\end{aligned}
$$

where the third last step follows by

$$H(X|Y) = H(X) - I(X,Y) = H(X) - H(Y) + H(Y|X) \geq H(X) - H(Y);$$

the second last follows by $H(X) \leq \log k$ for any $X$ with a support size of $k$, and the fact that there are at most $\exp(3\sqrt{m})$ many profiles of length $m$, as we explained above; and the last step follows by the elementary inequality

$$H(\mathrm{Bern}(\theta)) \leq 2(\log 2)\sqrt{\theta(1-\theta)}, \quad \forall \theta \in [0,1].$$

Our new objective is to bound $H((\varphi_i)_{i \in nI_j})$ from below. We will find a sub-interval $I_j^s$ of $I_j$ and bound $H((\varphi_i)_{i \in nI_j^s})$ in the rest of the section, since

$$H((\varphi_i)_{i \in nI_j}) \geq H((\varphi_i)_{i \in nI_j^s}).$$

For all $j \equiv 1 \bmod c$, our lower bound is simply

$$H((\varphi_i)_{i \in nI_j^s}) \geq \Omega \left( \frac{1}{\sqrt{\log n}} \min \left\{ p_{I_j}, j \cdot \log n \right\} \right),$$

which, together with $\sum_{j \equiv 1 \bmod c} \min\{p_{I_j}, j \cdot \log n\} \geq H_n^{\mathcal{S}}(p)/c$, implies that

$$H(\varphi) \geq -\mathcal{O}\left( \frac{1}{\sqrt{n}} \right) + \sum_{j \equiv 1 \bmod c} H((\varphi_i)_{i \in nI_j}) \geq \Omega\left( \frac{1}{\sqrt{\log n}} \right) \cdot T_n.$$

Henceforth, we assume that $j$ is sufficiently large and denote $L_j := j\sqrt{\log n}$.

For any $j$ and every integer $i \in S_j := [(j-1)^2, j^2] \log n$, define a sequence of intervals,

$$I_j^i := \frac{i}{n} + \frac{L_j}{n} [-1, 1].$$

Then for any $i \in S_j$ and $p_x \in I_j^i \cap I_j$, the corresponding Poisson probability satisfies

$$e^{-np_x} \frac{(np_x)^i}{i!} = e^{-i} \frac{i^i}{i!} \cdot \exp\left( -(np_x - i) + i \cdot \log\left( 1 + \frac{np_x - i}{i} \right) \right)$$

$$\geq \frac{1}{3\sqrt{i}} \cdot \exp\left( -\frac{2i}{3} \cdot \left( \frac{np_x - i}{i} \right)^2 \right)$$

$$\geq \frac{1}{9\sqrt{i}} \geq \frac{1}{9L_j}.$$

On the other hand, the following upper bound holds.

$$e^{-np_x} \frac{(np_x)^i}{i!} = e^{-i} \frac{i^i}{i!} \cdot \exp\left( -(np_x - i) + i \cdot \log\left( 1 + \frac{np_x - i}{i} \right) \right)$$

$$\leq e^{-i} \frac{i^i}{i!} \leq \frac{1}{\sqrt{2\pi i}} \leq \frac{1}{2L_j}.$$

In other words, for any $p_x, i/n \in I_j$ that differ by at most $L_j/n$,

$$\Pr(\mathrm{Poi}(np_x) = i) \in \frac{1}{L_j} \left[ \frac{1}{9}, \frac{1}{2} \right].$$

Partition $I_j$ into sub-intervals of equal length $L_j/n$. The partition has a size of at most $2\sqrt{\log n}$. Assign each probability $p_x \in I_j$ a length-$L_j/n$ interval $I_{p_x}$ centered at $p_x$. Then, each interval $I_{p_x}$ covers at least one of the sub-intervals in the partition. Since there are exactly $p_{I_j}$ intervals $I_{p_x}$, one can find a partition sub-interval $I_j^s$ contained in at least $p_{I_j}/(2\sqrt{\log n})$ of them. Denote by $\mathcal{X}_s$ the collection of symbols corresponding to these intervals.

Next, we bound from below the entropy of the truncated profile $(\varphi_i)_{i \in nI_j^s}$ over $nI_j^s$. Denote by $j_s$ the left end point of $nI_j^s$. By the chain rule of entropy for multiple random variables,

$$H((\varphi_i)_{i \in nI_j^s}) = \sum_{i=j_s}^{j_s + L_j - 1} H(\varphi_i | \varphi_{j_s}, \dots, \varphi_{i-1}).$$

Consider a particular term on the right-hand side with $i \in [j_s, j_s + L_j - 1]$. By the conditional independence and fact that conditioning reduces entropy,

$$H(\varphi_i | \varphi_{j_s}, \dots, \varphi_{i-1}) \geq H(\varphi_i | \varphi_{j_s}, \dots, \varphi_{i-1}; \mathbb{1}_{j_s \leq \mu_x \leq i-1}, x \in \mathcal{X})$$

$$= H(\varphi_i | \mathbb{1}_{j_s \leq \mu_x \leq i-1}, x \in \mathcal{X})$$

$$= H(\varphi_i | \mathbb{1}_{j_s \leq \mu_x \leq i-1}, x \in \mathcal{X}_s; \mathbb{1}_{j_s \leq \mu_x \leq i-1}, x \notin \mathcal{X}_s)$$

To characterize the condition, we define a random variable

$$K_i^s := \sum_{x \in \mathcal{X}_s} \mathbb{1}_{j_s \leq \mu_x \leq i-1}.$$

Note that $\mathbb{E}[\mathbb{1}_{j_s \leq \mu_x \leq i-1}] = \sum_{t=j_s}^{i-1} \Pr(\text{Poi}(np_x) = t) \leq (i - j_s)/(2L_j)$, which is at most $1/10$ for $i \leq j_s + L_j/5$. The following lemma transforms this into a high-probability statement.

**Lemma 3.** *Let $Y_i, i \in [1, m]$ be independent indicator random variables. Let $Y := \sum_i Y_i$ denote their sum and $\lambda := \mathbb{E}[Y]$ denote the expected sum. Then for $c > 0$, we have*

$$\Pr(Y \geq \lambda(1 + c)) \leq \exp(-\lambda c^2/(2 + 2c/3)).$$

Below we consider only $i \leq j_s + L_j/5$. Note that $c/(2 + 2c/3)$ is increasing for $c > 0$.

Since $\mathbb{E}[K_i^s] = \sum_{x \in \mathcal{X}_s} \mathbb{E}[\mathbb{1}_{j_s \leq \mu_x \leq i-1}] \leq |\mathcal{X}_s|/10$,

$$\Pr(K_i^s \geq |\mathcal{X}_s|/2) \leq \exp(-36/35) < 1/2.$$

where we set $c = 4$ in the above lemma and assume that $|\mathcal{X}_s| \geq 3$ (assuming only $|\mathcal{X}_s| \geq 1$, the upper bound becomes $3/4$). Recall that

$$
\begin{aligned}
H(\varphi_i|\varphi_{j_s}, \ldots, \varphi_{i-1}) &\geq H(\varphi_i|\mathbb{1}_{j_s \leq \mu_x \leq i-1}, x \in \mathcal{X}_s; \mathbb{1}_{j_s \leq \mu_x \leq i-1}, x \notin \mathcal{X}_s) \\
&= \sum_{(c_x)_{x \in \mathcal{X}} \in \{0,1\}^{\mathcal{X}}} H(\varphi_i|\mathbb{1}_{j_s \leq \mu_x \leq i-1} = c_x, x \in \mathcal{X}_s) \\
&\qquad\qquad \times \Pr(\mathbb{1}_{j_s \leq \mu_x \leq i-1} = c_x, x \in \mathcal{X}_s).
\end{aligned}
$$

Denote by $V_s \subseteq \{0,1\}^{\mathcal{X}}$ the collection of $(c_x)_{x \in \mathcal{X}}$ satisfying $\sum_{x \in \mathcal{X}_s} c_x < |\mathcal{X}_s|/2$. The above derivation shows that

$$\sum_{(c_x)_{x \in \mathcal{X}} \in V_s} \Pr(\mathbb{1}_{j_s \leq \mu_x \leq i-1} = c_x, x \in \mathcal{X}_s) \geq \frac{1}{2}.$$

By independence, for any $(c_x)_{x \in \mathcal{X}} \in V_s$, we have

$$
\begin{aligned}
(\varphi_i|\mathbb{1}_{j_s \leq \mu_x \leq i-1} = c_x, x \in \mathcal{X}_s) &= \sum_{x \in \mathcal{X}:c_x=0} (\mathbb{1}_{\mu_x=i}|\mathbb{1}_{j_s \leq \mu_x \leq i-1} = 0) \\
&= \sum_{x \in \mathcal{X}_s:c_x=0} (\mathbb{1}_{\mu_x=i}|\mathbb{1}_{j_s \leq \mu_x \leq i-1} = 0) \\
&\qquad\qquad + \sum_{x \notin \mathcal{X}_s:c_x=0} (\mathbb{1}_{\mu_x=i}|\mathbb{1}_{j_s \leq \mu_x \leq i-1} = 0).
\end{aligned}
$$

For any $x \in \mathcal{X}_s$ with $c_x = 0$, the corresponding indicator variable satisfies

$$
\begin{aligned}
\mathbb{E}[\mathbb{1}_{\mu_x=i}|\mathbb{1}_{j_s \leq \mu_x \leq i-1} = 0] &= \frac{\Pr(\mathbb{1}_{\mu_x=i} \text{ and } \mu_x \notin [j_s, i-1])}{\Pr(\mu_x \notin [j_s, i-1])} \\
&= \frac{\Pr(\mathbb{1}_{\mu_x=i})}{1 - \Pr(\mu_x \in [j_s, i-1])} \\
&= \frac{\frac{1}{L_j}\left[\frac{1}{9}, \frac{1}{2}\right]}{1 - \left[0, \frac{L_j}{5}\right] \cdot \frac{1}{L_j}\left[\frac{1}{9}, \frac{1}{2}\right]} \\
&= \frac{1}{L_j}\left[\frac{1}{9}, \frac{5}{9}\right].
\end{aligned}
$$

On the other hand, for any $x \notin \mathcal{X}_s$,

$$e^{-np_x}\frac{(np_x)^i}{i!} \leq e^{-i}\frac{i^i}{i!} \leq \frac{1}{\sqrt{2\pi i}} \leq \frac{1}{2L_j}.$$

Therefore, the corresponding indicator variable satisfies

$$\mathbb{E}[\mathbb{1}_{\mu_x=i}|\mathbb{1}_{j_s\leq\mu_x\leq i-1}=0] = \frac{\Pr(\mathbb{1}_{\mu_x=i})}{1-\Pr(\mu_x\in[j_s,i-1])} \leq \frac{\frac{1}{L_j}\left[0,\frac{1}{2}\right]}{1-\left[0,\frac{L_j}{5}\right]\cdot\frac{1}{L_j}\left[0,\frac{1}{2}\right]} \leq \frac{5}{9}\cdot\frac{1}{L_j}.$$

To summarize, we have shown that $(\varphi_i|\mathbb{1}_{j_s\leq\mu_x\leq i-1}=c_x, x\in\mathcal{X}_s)$ is the sum of $|\mathcal{X}|$ independent Bernoulli random variables. Among these Bernoulli variables, at least $|\mathcal{X}_s|/2\geq p_{I_j}/(2\sqrt{\log n})$ have a bias of $\frac{1}{L_j}\left[\frac{1}{9},\frac{5}{9}\right]$, while others have a bias of at most $\frac{5}{9}\cdot\frac{1}{L_j}$.

The following lemma, recently established by Hillion et al. [2019], shows the relation among the entropy values of sums of independent Bernoulli random variables with different bias parameters.

**Lemma 4.** *Let* $X_t, Y_t, t\in[m]$ *be independent indicator random variables. Denote by $X$ and $Y$ the sums of $X_t$'s and $Y_t$'s, respectively. If $\mathbb{E}[X_t]\leq\mathbb{E}[Y_t]\leq 1/2, \forall t\in[m]$,*

$$H(\sum_t X_t) \leq H(\sum_t Y_t).$$

This lemma, together with the previous results, shows that

$$H(\varphi_i|\mathbb{1}_{j_s\leq\mu_x\leq i-1}=c_x, x\in\mathcal{X}_s) \geq H\left(\mathrm{bin}(p_{I_j}/(2\sqrt{\log n}), 1/(9L_j))\right).$$

The next lemma further bounds the entropy of a binomial random variable.

**Lemma 5.** *For any $m>1$ and $q\in[1/m, 1-1/m]$,*

$$H(\mathrm{bin}(m,q)) \geq \frac{1}{2}\log\left((2\pi)^{1-(1-q)^m-q^m}mq(1-q)\right) - \frac{1}{12m}.$$

*Proof.* By definition, the left-hand side satisfies

$$\begin{aligned}
H(\mathrm{bin}(m,q)) &= -\sum_{t=0}^m \binom{m}{t}q^t(1-q)^{m-t}\log\left(\binom{m}{t}q^t(1-q)^{m-t}\right)\\
&= -\sum_{t=0}^m \binom{m}{t}q^t(1-q)^{m-t}(t\log q + (m-t)\log(1-q)\\
&\qquad\qquad\qquad\qquad + \log m! - \log t! - \log(m-t)!)\\
&= mH(\mathrm{Bern}(q)) - \log m! + \sum_{t=0}^m \binom{m}{t}q^t(1-q)^{m-t}(\log t! + \log(m-t)!).
\end{aligned}$$

By Stirling's formula, for any $t\geq 1$,

$$\log t! \geq \left(t+\frac{1}{2}\right)\log t + \frac{1}{2}\log(2\pi) - t.$$

Substituting the right-hand side into the above equation yields

$$\begin{aligned}
S_m(q) := \sum_{t=0}^m \binom{m}{t}q^t(1-q)^{m-t}\log t! &\geq \frac{1}{2}(1-(1-q)^m)\log(2\pi) - mq\\
&\quad + \sum_{t=1}^m \binom{m}{t}q^t(1-q)^{m-t}\left(t+\frac{1}{2}\right)\log t.
\end{aligned}$$

Let $g(x) := 0$ for $x\in[0,1)$ and $g(x) := (x+1/2)\log x$ for $x\geq 1$. Simple calculus shows that the function is concave. Applying the concavity of $g$ to the last sum yields

$$\sum_{t=1}^m \binom{m}{t}q^t(1-q)^{m-t}\left(t+\frac{1}{2}\right)\log t \geq g\left(\sum_{t=0}^m\binom{m}{t}q^t(1-q)^{m-t}\cdot t\right) = \left(mq+\frac{1}{2}\right)\log(mq),$$

where the last step follows by the fact that $mq \geq 1$. A similar inequality holds for the weighted sum of $\log(m-t)!$. Consolidating these inequalities, we obtain

$$
S_m(q) + S_m(1-q) \geq \left(mq + \frac{1}{2}\right)\log(mq) + \left(m(1-q) + \frac{1}{2}\right)\log(m(1-q))
$$

$$
+ \frac{1}{2}(1 - (1-q)^m)\log(2\pi) - mq + \frac{1}{2}(1 - q^m)\log(2\pi) - m(1-q)
$$

$$
= (m+1)\log m - mH(\mathrm{Bern}(q)) + \frac{1}{2}\log(q(1-q))
$$

$$
+ \frac{1}{2}(2 - (1-q)^m - q^m)\log(2\pi) - m.
$$

On the other hand, for the $\log m!$ term,

$$
\log m! \leq \left(m + \frac{1}{2}\right)\log m + \frac{1}{2}\log(2\pi) - m + \frac{1}{12m}.
$$

Substituting the previous term bounds into the $H(\mathrm{bin}(m, q))$ expression yields

$$
H(\mathrm{bin}(m, q)) = mH(\mathrm{Bern}(q)) - \log m! + S_m(q) + S_m(1-q)
$$

$$
\geq \frac{1}{2}\log\left((2\pi)^{1-(1-q)^m - q^m} mq(1-q)\right) - \frac{1}{12m}. \qquad \square
$$

Before continuing, we remark that the bound in the above lemma has the right dependence on $mq(1-q)$ in the sense that if we fix $q$ and increase $m$, the lower bound converges to $\frac{1}{2}\log(\Theta(mq(1-q)))$. Another point to mention is that the above bound covers $q \in [1/m, 1 - 1/m]$, while Lemma 6 appearing later in this section covers $q \notin [1/m, 1 - 1/m]$. Note that the dependence on $mq(1-q)$ changes from logarithmic to linear, showing an "elbow effect" around $1/m$.

Assume that $p_{I_j}/(2\sqrt{\log n}) \geq 9L_j$, then for any $(c_x)_{x \in \mathcal{X}} \in V_s$,

$$
H(\varphi_i | \mathbb{1}_{j_s \leq \mu_x \leq i-1} = c_x, x \in \mathcal{X}_s) \geq H(\mathrm{bin}(p_{I_j}/(2\sqrt{\log n}), 1/(9L_j))) \geq \frac{1}{2}.
$$

Consolidating this with the previous results yields that

$$
H(\varphi_i | \varphi_{j_s}, \ldots, \varphi_{i-1}) \geq \sum_{(c_x)_{x \in \mathcal{X}} \in V_s} \frac{1}{2} \cdot \Pr(\mathbb{1}_{j_s \leq \mu_x \leq i-1} = c_x, x \in \mathcal{X}_s) \geq \frac{1}{2} \cdot \frac{1}{2} = \frac{1}{4},
$$

where we utilize $p_{I_j}/(2\sqrt{\log n}) \geq 9L_j \geq 9$ and $(1-q)^m + q^m < 1/e$ for $\forall m \geq 3, q \in [1/m, 1/2]$. We can then bound the quantity of interest as follows.

$$
H((\varphi_i)_{i \in nI_j^s}) = \sum_{i=j_s}^{j_s + L_j - 1} H(\varphi_i | \varphi_{j_s}, \ldots, \varphi_{i-1})
$$

$$
\geq \sum_{i=j_s}^{j_s + L_j/5} H(\varphi_i | \varphi_{j_s}, \ldots, \varphi_{i-1})
$$

$$
\geq \frac{L_j}{5} \cdot \frac{1}{4} = \frac{L_j}{20}
$$

$$
= \frac{1}{20\sqrt{\log n}} \min\left\{p_{I_j}, j \cdot \log n\right\}.
$$

On the other hand, if $9L_j \geq p_{I_j}/(2\sqrt{\log n}) \gg 1$, we can further "compress" the truncated profile $(\varphi_i)_{i \in nI_j^s}$ over $nI_j^s$ to reduce the effective value of $L_j$. Specifically, for any integer $t < L_j$, we define the $t$-compressed version of $(\varphi_i)_{i \in nI_j^s}$ as

$$
(\varphi_i)_{i \in nI_j^s}^t := \left(\sum_{i=j_s+(\ell-1)t}^{j_s+\ell t - 1} \varphi_i\right)_{\ell \in [L_j/t]}.
$$

Note that for each $t$, the length of $(\varphi_i)_{i\in nI_j^s}^t$ is $L_j^t := L_j/t$. For each entry in the compressed version, we can again express the entry as the sum of independent indicator random variables. Specifically,

$$\sum_{i=j_s+(\ell-1)t}^{j_s+\ell t-1} \varphi_i = \sum_{x\in\mathcal{X}} \mathbb{1}_{\mu_x\in[j_s+(\ell-1)t,j_s+\ell t-1]}.$$

Furthermore, for any $x\in\mathcal{X}_s$, the expectation of each indicator variable satisfies

$$\mathbb{E}[\mathbb{1}_{\mu_x\in[j_s+(\ell-1)t,j_s+\ell t-1]}] = \sum_{i=j_s+(\ell-1)t}^{j_s+\ell t-1} e^{-np_x}\frac{(np_x)^i}{i!}$$
$$= \frac{t}{L_j}\left[\frac{1}{9},\frac{1}{2}\right] = \frac{1}{L_j^t}\left[\frac{1}{9},\frac{1}{2}\right].$$

Similarly, for any $x\in\mathcal{X}$, we have $\mathbb{E}[\mathbb{1}_{\mu_x\in[j_s+(\ell-1)t,j_s+\ell t-1]}] \le 1/(2L_j^t)$.

Now, choose $t$ large enough so that $18L_j^t \ge p_{I_j}/(2\sqrt{\log n}) \ge 9L_j^t$. Following the reasoning in the previous case shows that

$$H((\varphi_i)_{i\in nI_j^s}) \ge H((\varphi_i)_{i\in nI_j^s}^t) \ge \Omega\left(\frac{1}{\sqrt{\log n}}\min\left\{p_{I_j}, j\cdot\log n\right\}\right).$$

It remains to consider the case of $\mathcal{O}(\sqrt{\log n}) \ge p_{I_j} \ge 1$, for which we adopt our previous analysis.

Again, partition $I_j$ into sub-intervals of equal length $L_j/n$. Then, assign each probability $p_x \in I_j$ a length-$L_j/n$ interval $I_{p_x}$ centered at $p_x$. By construction, each interval $I_{p_x}$ covers at least one of the sub-intervals in the partition. Redefine any of these covered sub-intervals as $I_j^s$. Denote by $\mathcal{X}_s$ the collection of symbols corresponding to the covering intervals.

Note that $\mathcal{O}(\sqrt{\log n}) \ge p_{I_j} \ge |\mathcal{X}_s| \ge 1$. For any $i\in[j_s, j_s+L_j/5]$, the previous analysis shows that

$$H(\varphi_i|\varphi_{j_s},\ldots,\varphi_{i-1}) \ge H(\mathrm{bin}(|\mathcal{X}_s|, 1/(9L_j)))\cdot(1-3/4).$$

We bound the right-hand side with the following lemma.

**Lemma 6.** *For any $m \ge 1$, and $q \le \min\{1/2, 1/m\}$ or $q \ge \max\{1/2, 1-1/m\}$,*

$$H(\mathrm{bin}(m,q)) \ge \frac{m}{4}\min\{q, 1-q\} \ge \frac{1}{4}mq(1-q).$$

*Proof.* By symmetry, we need to consider only the case of $q\in[0, 1/m]$.

$$H(\mathrm{bin}(m,q)) \ge H(\mathbb{1}_{\mathrm{bin}(m,q)\ge 1})$$
$$= H(((1-q)^m, 1-(1-q)^m))$$
$$\ge -(1-q)^m(m\log(1-q))$$
$$\ge -\frac{m}{4}\log(1-q)$$
$$\ge \frac{m}{4}\cdot q. \qquad\square$$

Consolidating the lemma and the chain rule of entropy yields,

$$H((\varphi_i)_{i\in nI_j^s}) = \sum_{i=j_s}^{j_s+L_j-1} H(\varphi_i|\varphi_{j_s},\ldots,\varphi_{i-1})$$
$$\ge \sum_{i=j_s}^{j_s+L_j/5} H(\varphi_i|\varphi_{j_s},\ldots,\varphi_{i-1})$$
$$\ge \frac{L_j}{5}\cdot\frac{|\mathcal{X}_s|}{4\cdot 9\cdot L_j}\cdot\left(1-\frac{3}{4}\right) = \frac{|\mathcal{X}_s|}{720}$$
$$= \Omega\left(\frac{1}{\sqrt{\log n}}\min\left\{p_{I_j}, j\cdot\log n\right\}\right).$$

Alternatively, we can use the fact that adding independent random variables does not decrease entropy, i.e., $H(Y + Z) \geq H(Y)$ for any independent variables $Y$ and $Z$. Note that

$$(\varphi_i)_{i \in nI_j^s}^t = \sum_{x \in \mathcal{X}} (\mathbb{1}_{\mu_x=i})_{i \in I_j^s}.$$

Let $y$ be an arbitrary symbol that belongs to $\mathcal{X}_s$. Then,

$$H((\varphi_i)_{i \in nI_j^s}) \geq H((\varphi_i)_{i \in nI_j^s}^t) \geq H((\mathbb{1}_{\mu_y=i})_{i \in I_j^s}) \geq H((\mathbb{1}_{\mu_y=j_s}, \mathbb{1}_{\mu_y=j_s+1})).$$

By the previous derivations, both $\Pr(\mu_y = j_s)$ and $\Pr(\mu_y = j_s + 1)$ belong to $\frac{1}{L_j}[1/9, 1/2]$. Hence,

$$H((\varphi_i)_{i \in nI_j^s}) \geq H\left(\mathrm{Bern}\left(\frac{2}{11}\right)\right) \geq \frac{2}{5} = \Omega\left(\frac{1}{\sqrt{\log n}} \min\left\{p_{I_j}, j \cdot \log n\right\}\right).$$

Note that this argument does not apply to other cases, since

$$H((\mathbb{1}_{\mu_y=i})_{i \in I_j^s}) = \mathcal{O}(\log L_j) = \mathcal{O}(\log n),$$

while $\min\left\{p_{I_j}, j \cdot \log n\right\}$ can be as large as $\tilde{\Theta}(n^{1/3})$ in general.

The proof is complete upon noting that indices with $j = \mathcal{O}(1)$ corresponds to a total contribution of at most $\mathcal{O}(1)$ to $H_n^{\mathcal{S}}(p)$ and $H_n^{\mathcal{S}}(p) = \tilde{\Theta}(\mathbb{E}[\mathcal{D}(\varphi)]) = \tilde{\Theta}(D(\varphi))$, with probability at least $1 - \mathcal{O}(1/\sqrt{n})$.

**Summary**   The simple expression shows that $\mathcal{H}_n^{\mathcal{S}}(p)$ characterizes the variability of ranges that the actual probabilities spread over. As Theorem 11 shows, $\mathcal{H}_n^{\mathcal{S}}(p)$ closely approximates $E_n(p)$, the value around which $\mathcal{D}_n \sim p$ concentrates (Theorem 9) and $\mathcal{H}_n(p)$ lies (Thoerem 1). Henceforth, we use $\mathcal{H}_n^{\mathcal{S}}(p)$ as a proxy for both $\mathcal{H}_n(p)$ and $\mathcal{D}_n$, and study its attributes and values.

Let $p \in \Delta$ be an arbitrary discrete distribution. Recall that in Section A, we partition the unit interval into a sequence of ranges,

$$I_j := \left((j-1)^2 \frac{\log n}{n}, \; j^2 \frac{\log n}{n}\right], 1 \leq j \leq \sqrt{\frac{n}{\log n}},$$

denote by $p_{I_j}$ the number of probabilities $p_x$ belonging to $I_j$, and relate $E_n(p)$ to an induced shape-reflecting quantity,

$$\mathcal{H}_n^{\mathcal{S}}(p) := \sum_{j \geq 1} \min\left\{p_{I_j}, j \cdot \log n\right\},$$

the sum of the effective number of probabilities lying within each range.

The simple expression of $\mathcal{H}_n^{\mathcal{S}}(p)$ shows that it characterizes the variability of ranges the actual probabilities spread over. As Theorem 11 shows, $\mathcal{H}_n^{\mathcal{S}}(p)$ closely approximates $E_n(p)$, the value around which $\mathcal{D}_n \sim p$ concentrates (Theorem 9) and $\mathcal{H}_n(p)$ lies (Thoerem 1). In this section, we use $\mathcal{H}_n^{\mathcal{S}}(p)$ as a proxy for both $\mathcal{H}_n(p)$ and $\mathcal{D}_n$, and study its attributes and values.

To further our understanding of profile entropy and dimension, in the next two sections, we investigate the analytical attributes of $\mathcal{H}_n^{\mathcal{S}}(p)$ concerning monotonicity and Lipschitzness.

### A.4 Extension: Profile Entropy Estimation via Monotonicity

Among the many attributes that $\mathcal{H}_n^{\mathcal{S}}(p)$ possesses, monotonicity is perhaps most intuitive. One may expect a larger value of $\mathcal{H}_n^{\mathcal{S}}(p)$ as the sample size $n$ increases, since additional observations reveal more information on the variability of probabilities. Below we confirm this intuition.

**Theorem 12.** *For any $n \geq m \gg 1$ and $p \in \Delta$,*
$$\mathcal{H}_n^{\mathcal{S}}(p) \geq H_m^{\mathcal{S}}(p).$$

The above result that lowerly bounds $\mathcal{H}_n^{\mathcal{S}}(p)$ with $H_m^{\mathcal{S}}(p)$ for $m \leq n$. Besides this, a more desirable result is to upperly bound $\mathcal{H}_n^{\mathcal{S}}(p)$ with some function of $H_m^{\mathcal{S}}(p)$. Such a result will enable us to draw a sample of size $m \leq n$, obtain an estimate of $H_m^{\mathcal{S}}(p)$ from $\mathcal{D}_m$ (by the entropy-dimension equivalence), and use it to bound the value of $\mathcal{H}_n^{\mathcal{S}}(p)$ for a much larger sample size $n$.

With such an estimate, we can perform numerous tasks such as *predicting* the performance of PML when more observations are available, or the space needed for storing the profile of a longer sample sequence. These applications are closely related to the recent works on *learnability estimation* by Kong and Valiant [2018], Kong et al. [2019], namely, one wish to know how many (additional) observations are required for a learning algorithm to achieve a certain level of performance.

The next theorem provides a simple and tight upper bound on $\mathcal{H}_n^{\mathcal{S}}(p)$ in terms of $H_m^{\mathcal{S}}(p)$.

**Theorem 13.** *For any $n \geq m \gg 1$ and $p \in \Delta$,*
$$\mathcal{H}_n^{\mathcal{S}}(p) \leq \sqrt{\frac{n \log n}{m \log m}} \cdot H_m^{\mathcal{S}}(p).$$

**Estimation**   Before continuing to the proof, we present some direct implications.

1. If for $m = \Omega(n^{0.01})$, we have $H_m^{\mathcal{S}}(p) \ll \sqrt{m}$, then $H_n^{\mathcal{S}}(p) \ll \sqrt{n}$.

2. For any two integers $m \leq n$ and distribution $p$,
$$\frac{H_m^{\mathcal{S}}(p)}{\sqrt{m \log m}} \geq \frac{H_n^{\mathcal{S}}(p)}{\sqrt{n \log n}}.$$
   In other words, the sequence $A_m := H_m^{\mathcal{S}}(p)/\sqrt{m \log m}$, $m \leq n$, is monotonically decreasing and converges to $A_n$. As we increase the value of $m$, $(\sqrt{n \log n} \cdot A_m)$, which can be viewed as our estimate of $H_n^{\mathcal{S}}(p)$, is getting more and more accurate. For the purpose of adaptive estimation, if $n = 2^t$, we can choose $m = 2^0, 2^1, \ldots, 2^t$.

*Proof.* Below we prove both the lower and upper bounds. For clarity, denote by $p(m,j)$ the value of $p_{I_j}$ corresponding to $H_m^{\mathcal{S}}(p)$, and $p(n,j)$ the value of $p_{I_j}$ corresponding to $H_n^{\mathcal{S}}(p)$. Furthermore, denote $r := \sqrt{(n/m)((\log m)/\log n)}$, which is treated as an integer. Then, by the definition of $H_{\cdot}^{\mathcal{S}}$,

$$rH_m^{\mathcal{S}}(p) = r \sum_{j \geq 1} \min\{p(m,j), j \cdot \log m\}$$

$$= \sum_{j \geq 1} \min\left\{r \cdot \sum_{i=rj-r+1}^{rj} p(n,i),\ rj \cdot \log m\right\}$$

$$\geq \sum_{j \geq 1} \sum_{t=0}^{r-1} \min\left\{\sum_{i=rj-r+1}^{rj} p(n,i),\ (rj-t) \cdot \log m\right\}$$

$$\geq \sum_{j \geq 1} \sum_{t=0}^{r-1} \min\{p(n, rj-t),\ (rj-t) \cdot \log m\}$$

$$= \sum_{i \geq 1} \min\{p(n,i),\ i \cdot \log m\}$$

$$\geq \frac{\log m}{\log n} \cdot H_n^{\mathcal{S}}(p).$$

The lower-bound part basically follows by reversing the above inequalities.

$$H_n^{\mathcal{S}}(p) = \sum_{i \geq 1} \min\{p(n,i),\ i \cdot \log n\}$$

$$= \sum_{j \geq 1} \sum_{t=0}^{r-1} \min\{p(n,rj-t),\ (rj-t) \cdot \log n\}$$

$$\geq \sum_{j \geq 1} \sum_{t=0}^{r-1} \min\{p(n,rj-t),\ (rj-r+1) \cdot \log n\}$$

$$\geq \sum_{j \geq 1} \min\left\{\sum_{t=0}^{r-1} p(n,rj-t),\ (rj-r+1) \cdot \log n\right\}$$

$$= \sum_{j \geq 1} \min\{p(m,j),(rj-r+1) \cdot \log m\}$$

$$\geq H_m^{\mathcal{S}}(p).$$

This completes the proof of the theorem. $\qquad\square$

### A.5 Extension: Lipschitzness of Profile Entropy

Note that we can view $\mathcal{H}_n^{\mathcal{S}}(p)$ as a distribution property. In this section, we establish the Lipschitzness of $\mathcal{H}_n^{\mathcal{S}}(p)$ under a weighted Hamming distance and the $\ell_1$ distance between distributions. Precisely, given two distributions $p, q \in \Delta$, the vanilla *Hamming distance* is denoted by

$$h(p,q) := \sum_x \mathbb{1}_{p_x \neq q_x}.$$

This may not be suitable for the purpose of statistical inference since the two distributions could differ at many symbols, while these symbols account for only a negligible total probability and has little effects on most induced statistics. To address this, we propose a *weighted Hamming distance*

$$h_{\mathcal{W}}(p,q) := \sum_{x \in \mathcal{X}} \max\{p_x, q_x\} \cdot \mathbb{1}_{p_x \neq q_x}.$$

The next result measures the Lipschitzness of $H_n^{\mathcal{S}}$ under $h_{\mathcal{W}}$.

**Theorem 14.** *For any integer $n$, and distributions $p$ and $q$, if $h_{\mathcal{W}}(p,q) \leq \varepsilon$ for some $\varepsilon \geq 1/n$,*

$$\left|\mathcal{H}_n^{\mathcal{S}}(p) - H_n^{\mathcal{S}}(q)\right| \leq \tilde{\mathcal{O}}(\sqrt{\varepsilon n}).$$

*Proof.* Recall that the quantity of interest is

$$\mathcal{H}_n^{\mathcal{S}}(p) := \sum_{j \geq 1} \min\{p_{I_j}, j \cdot \log n\}.$$

Given the bound of $h_{\mathcal{W}}(p,q) \leq \varepsilon$, we denote by $\mathcal{Y}$ the collection of symbols $x$ at which $p_x \neq q_x$. By definition, we have both $\sum_{x \in \mathcal{Y}} p_x \leq \varepsilon$ and $\sum_{x \in \mathcal{Y}} q_x \leq \varepsilon$. Below, we show that these symbols modify the value of $\mathcal{H}_n^{\mathcal{S}}(p)$ by at most $\tilde{\mathcal{O}}(\sqrt{\varepsilon n})$. By symmetry, the same claim also holds for the distribution $q$. Combining the two claims yields the desired result.

First, we consider $x \in \mathcal{Y}$ satisfying $p_x = 0$ or $p_x \in I_1 = (0, (\log n)/n]$. Such a symbol either does not contribute the value of $\mathcal{H}_n^{\mathcal{S}}(p)$, or affects only the value of the first term $\min\{p_{I_1}, \log n\}$, which is at most $\log n$. Hence the claim holds for this case.

Next, consider symbols $x \in \mathcal{Y}$ satisfying $p_x \in I_j = ((j-1)^2 \frac{\log n}{n}, j^2 \frac{\log n}{n}]$ for some $j \geq 2$ and denote the collection of them by $\mathcal{Z} \subseteq \mathcal{Y}$. By the above assumption, we have $\sum_{x \in \mathcal{Z}} p_x \leq \varepsilon$. To maximize their impact on $\mathcal{H}_n^{\mathcal{S}}(p)$ under this constraint, we should set their values to be

$$p_j := (j-1)^2 \frac{\log n}{n}, \ j = 2, \ldots J,$$

for some $J$ to be determined, where each $p_j$ repeats exactly $j \log n$ times. Then, the symbols in $\mathcal{Z}$ contributes at most $\sum_{j=2}^{J} j \log n = (\log n)(J-1)(J+2)/2$ to $\mathcal{H}_n^{\mathcal{S}}(p)$, and the above constraint on the total probability mass bounds transforms to

$$\varepsilon \geq \sum_{x \in \mathcal{Z}} p_x \geq \sum_{j=2}^{J} (j \log n) \cdot (j-1)^2 \frac{\log n}{n} \geq \frac{(\log n)^2}{12n} J(J^2-1)(-2+3J).$$

Therefore in this case, the contribution is again $\tilde{\mathcal{O}}(\sqrt{\varepsilon n})$, which completes the proof. $\qquad\square$

Replacing $\max\{p_x, q_x\}$ with $|p_x - q_x|$ induces a common similarity measure, the $\ell_1$ distance. The next theorem is an analog to Theorem 14 under this classical distance.

**Theorem 15.** *For any integer $n$, and distributions $p$ and $q$, if $\ell_1(p, q) \leq \varepsilon$ for some $\varepsilon \geq 0$,*

$$\left| \mathcal{H}_n^{\mathcal{S}}(p) - c H_n^{\mathcal{S}}(q) \right| \leq \mathcal{O}((\varepsilon n)^{2/3}),$$

*where $c$ is a constant in $[1/3, 3]$. Note that the inequality is significant iff $\varepsilon \leq \tilde{\Theta}(1/n^{1/4})$, since the value of $\mathcal{H}_n^{\mathcal{S}}(p)$ is at most $\mathcal{O}(\sqrt{n \log n})$ for all $p$.*

By symmetry, it suffices to prove that under the conditions in Theorem 15,

$$H_n^{\mathcal{S}}(p) \leq 3 H_n^{\mathcal{S}}(q) + \mathcal{O}((\varepsilon n)^{2/3}).$$

*Proof.* Consider the optimization problem of modifying $p$ by at most $\varepsilon$ and maximizing the increase in $H_n^{\mathcal{S}}(p)$. For each $j$ and each probability $p_x \in j$, denote by $p'_x$ the modified value. Depending on the location of $p'_x$, there are three types of possible modifications, as illustrated below.

- For the first type, we still have $p'_x \in I_j$. This does not change the value of $p_{I_j}$ and hence does not increase $H_n^{\mathcal{S}}(p)$.

- For the second type, we have $p'_x \in I_{j-1}$ or $p'_x \in I_{j+1}$. If $p_{I_j} \leq j \cdot \log n$, this will decrease the value of $\min\{p_{I_j}, j \cdot \log n\}$ by 1 and increase the value of $\min\{p_{I_{j-1}}, (j-1) \cdot \log n\}$ or $\min\{p_{I_{j+1}}, (j+1) \cdot \log n\}$ by at most one. Hence in this case, the value of $H_n^{\mathcal{S}}(p)$ can only decrease. If $p_{I_j} > j \cdot \log n$, then $\min\{p_{I_j}, j \cdot \log n\} = j \cdot \log n$. For a particular $j$, all such modifications can increase the value of $H_n^{\mathcal{S}}(p)$ by at most $(j-1) \log n + (j+1) \log n = 2j \log n$, which is twice the value of $\min\{p_{I_j}, j \cdot \log n\}$. Hence, all such modifications, when combined, increase the value of $H_n^{\mathcal{S}}(p)$ by at most $2 H_n^{\mathcal{S}}(p)$.

- For the third type, we have $p'_x \in I_i$ and $|i - j| \geq 2$. If $i < j$, we require a probability mass of at least $((j-1)^2 \log n - i^2 \log n)/n \geq (i \log n)/n$, where $j \geq 3$. If $i > j$, we require a probability mass of at least $((i-1)^2 \log n - j^2 \log n)/n \geq (i \log n)/n$. The number of such modifications that could lead to an increase in the value of $H_n^{\mathcal{S}}(p)$ is at most $i \log n$. For each $i$, let $c_i$ denote the number of such modifications that will lead to an increase of $H_n^{\mathcal{S}}(p)$. Then, the total increase is $\sum_i c_i$, each $c_i$ is at most $i \log n$, and the total required probability mass required is at least $\sum_i c_i \cdot (i \log n)/n \leq \varepsilon$.

  Let $\{c_i\}$ be the optimal solution that maximizes $\sum_i c_i$. Assume that there are two indices $i < j$ satisfying $c_i < i \log n$ and $c_j > 0$. Then, if we replace $c_i$ and $c_j$ by $c_i + 1$ and $c_j - 1$, respectively, $\sum_i c_i$ will not change and $\sum_i c_i \cdot (i \log n)/n$ will decrease. Hence, we can assume that there exists $i'$ satisfying $c_i = i \log n, \forall i < i'$ and $c_i = 0, \forall i > i'$. In addition, assuming $\varepsilon n \geq \log n$ implies that $i' \geq 2$. Hence, we have $\sum_i c_i \leq (\log n) i'(i'+1)/2$ and

$$\sum_i c_i \leq 3.5 \cdot \left( \frac{n\varepsilon}{\sqrt{\log n}} \right)^{2/3}. \qquad\square$$

# B   Competitive-Optimal Property Inference

## B.1   Theorem 3: Sufficiency of Profiles

Numerous practical applications call for inferring *property values* of an unknown distribution from its samples, such as entropy for graphical modeling [Koller and Friedman, 2009], Rényi entropy

for sequential decoding [Arikan, 1996], and support size for species richness estimation [Magurran, 2013]. Therefore, *property inference* has attracted considerable attention over the past few decades.

**Property inference**    Formally, a *distribution property* over some collection $P \subseteq \Delta$ is a functional $f : P \to \mathbb{R}$ that associates with each distribution a real value. Given a sample $X^n$ from an unknown distribution $p \in P$, the problem of interest is to infer the value of $f(p)$. For this purpose, we employ another functional $\hat{f} : \mathcal{X}^* \to \mathbb{R}$, an *estimator* mapping every sample to a real value. We measure the statistical efficiency of $\hat{f}$ in approximating $f$ over P by its *absolute error* $|\hat{f}(X^n) - f(p)|$.

Given $X^n \sim p \in P$, the *minimal absolute error rate*, or simply *error*, that $\hat{f}$ achieves with probability at least $9/10$ is $r_n(p, \hat{f}) := \min\{r : \Pr(|\hat{f}(X^n) - f(p)| \leq r) \geq 9/10\}$, where the dependence on $f$ is *implicit*. While $p$ is often unknown, the *worst-case error* of an estimator $\hat{f}$ over all distributions in P is $r_n(P, \hat{f}) := \max_{p \in P} r_n(p, \hat{f})$, and the lowest worst-case error for P, achieved by the optimal estimator, is the *minimax error* $r_n(P) := \min_{\hat{f}'} r_n(P, \hat{f}')$.

**Symmetric properties**    An important class of properties is the collection of symmetric ones, which encompasses numerous well-known distribution characteristics, such as Shannon entropy, Rényi entropy, support size, and $\ell_1$ distance to the uniform distribution. Symmetry connects the estimation of such property to the sample profile, a sufficient statistic for the task in hand. The general principle of maximum likelihood then provides an intuitive estimator, *profile maximum likelihood (PML)* [Orlitsky et al., 2004], that maximizes the probability of observing the profile.

An estimator is *profile-based* if its values depends on only the profile. The theorem below shows that profile-based estimators are sufficient for inferring symmetric properties.

**Theorem 3** (Sufficiency of profiles). *For any symmetric property $f$ and set $P \subseteq \Delta$, and estimator $\hat{f}$, we can construct an explicit estimator $\hat{F}$ over length-$n$ profiles satisfying*

$$r_n(p, \hat{f}) = r_n(P, \hat{F} \circ \varphi),$$

*where both estimators can have independent randomness.*

*Proof.* First we show that given estimator $\hat{f}$, there is an estimator $\hat{f}_s$ which is symmetric, i.e., invariant with respect to domain-symbol permutations, and achieves the same guarantee. To see this, consider a random permutation $\tilde{\sigma}$ chosen uniformly randomly from the collection of permutations over the underlying alphabet. Let $\hat{f}_s := \hat{f} \circ \tilde{\sigma}$. Then for any $p \in \mathcal{P}$,

$$\Pr_{X^n \sim p}\left(\left|\hat{f}_s(X^n) - f(p)\right| > \varepsilon\right) \overset{(a)}{=} \Pr_{X^n \sim p}\left(\left|\hat{f} \circ \tilde{\sigma}(X^n) - f(p)\right| > \varepsilon\right)$$

$$\overset{(b)}{=} \sum_{\sigma} \Pr_{X^n \sim p}\left(\left|\hat{f} \circ \sigma(X^n) - f(p)\right| > \varepsilon \,\Big|\, \tilde{\sigma} = \sigma\right) \cdot \Pr\left(\tilde{\sigma} = \sigma\right)$$

$$\overset{(c)}{=} \sum_{\sigma} \Pr_{X^n \sim p}\left(\left|\hat{f} \circ \sigma(X^n) - f(p)\right| > \varepsilon\right) \cdot \Pr\left(\tilde{\sigma} = \sigma\right)$$

$$\overset{(d)}{=} \sum_{\sigma} \Pr_{X^n \sim \sigma(p)}\left(\left|\hat{f}(X^n) - f(\sigma(p))\right| > \varepsilon\right) \cdot \Pr\left(\tilde{\sigma} = \sigma\right)$$

$$\overset{(e)}{<} \sum_{\sigma} \delta \cdot \Pr\left(\tilde{\sigma} = \sigma\right)$$

$$\overset{(f)}{=} \delta,$$

where $(a)$ follows by the definition of $\hat{f}_s$; $(b)$ follows by the law of total probability; $(c)$ follows by the independence between $\tilde{\sigma}$ and $X^n$; $(d)$ follows by the symmetry of $f$ and the equivalence of applying $\sigma$ to $X^n$ and to $p$; $(e)$ follows by the fact that $\sigma(p) \in \mathcal{P}$ and the guarantee satisfied by the estimator $\hat{f}$; and $(f)$ follows by the law of total probability.

Before we proceed further, we introduce the following definitions. For any sequence $x^n$, the *sketch* of a symbol $x$ in $x^n$ is the set of indices $i \in [n]$ for which $x_i = x$. The *type* of a sequence $x^n$ is the set $\tau(x^n)$ of sketches of symbols appearing in $x^n$.

Since $\hat{f}_s$ is symmetric, there exists a mapping $\hat{f}_\tau$ over types satisfying $\hat{f}_s = \hat{f}_\tau \circ \tau$. Due to the i.i.d. assumption on the sample generation process, given the profile of a sample sequence, all the different types corresponding to this profile are equally likely. Let $\Lambda$ be a mapping that recovers this relation, i.e., $\Lambda$ maps each profile uniformly randomly to a type having this profile.

Then, for any $p \in \mathcal{P}$ and $X^n \sim p$,

$$\hat{f}_s(X^n) = \hat{f}_\tau \circ \tau(X^n) = \hat{f}_\tau \circ \Lambda \circ \varphi(X^n).$$

Consequently, the mapping $\hat{F} := \hat{f}_\tau \circ \Lambda$ is a profile-based estimator that satisfies

$$\Pr_{X^n \sim p} \left( \left| \hat{F}(\varphi(X^n)) - f(p) \right| > \varepsilon \right) = \Pr_{X^n \sim p} \left( \left| \hat{f}_s(X^n) - f(p) \right| > \varepsilon \right) < \delta, \ \forall p \in \mathcal{P}. \qquad \square$$

## B.2  Theorem 4: Competitiveness of PML

Naturally and generally, we study symmetric property inference over a distribution collection $\mathrm{P} \subseteq \Delta$ that is also *symmetric*, i.e., if $p \in \mathrm{P}$, then P as well contains all the symbol-permuted versions of $p$. For every sample $x^n \in \mathcal{X}^n$ and symmetric P, the *PML estimator* over P maps $x^n$ to a distribution

$$\mathcal{P}_\varphi(x^n) := \arg\max_{p \in \mathrm{P}} \Pr_{X^n \sim p} \left( \varphi(X^n) = \varphi(x^n) \right).$$

Given a sample $X^n \sim p \in \mathrm{P}$ and a symmetric property $p$, the PML plug-in estimator uses $f \circ \mathcal{P}(X^n)$ to estimate $f(p)$. Recent researches [Acharya et al., 2017, Hao and Orlitsky, 2019a] show that for an extensive family of symmetric properties, including the previously mentioned four, the PML plug-in estimator *universally* achieves minimax error in the large-alphabet regime, up to constant factors.

The next result shows that the PML estimator is adaptive to the simplicity of underlying distributions in inferring all symmetric properties, over any symmetric P. Specifically, the theorem states that the $n$-sample PML plug-in essentially performs as well as the optimal $n/\mathcal{H}_n(p)$-sample estimator, which approaches the performance of the optimal $n$-sample estimator if $p$ has a small $\mathcal{H}_n(p)$. Furthermore, for any property and estimator, there is a symmetric set $\mathrm{P}'$ for which this $1/\mathcal{H}_n(p)$ ratio is *optimal*.

**Theorem 4** (Competitiveness of PML). *For any symmetric property $f$ and set $\mathrm{P} \subseteq \Delta$, and every distribution $p \in \mathrm{P}$, the PML plug-in estimator satisfies*

$$r_n(p, f \circ \mathcal{P}_\varphi) \le 2r_{n_p}(\mathrm{P}),$$

*where $n_p := \simeq n/\mathcal{H}_n(p)$. On the other hand, for any estimator $\hat{f}$ and symmetric property $f$, there exists a symmetric set $\mathrm{P}' \subseteq \Delta$ such that for some $p \in \mathrm{P}'$,*

$$r_n(p, \hat{f}) \ge 2r_{n_p}(\mathrm{P}').$$

## B.3  Prior Work and Discussions

**Results**   Recent years have shown interests in determining the limits of inferring symmetric distribution properties. Building upon worst-case analysis, the major contribution of these works is showing that for several specific properties, one can design more involved estimators whose worst-case performance is better than the empirical-distribution plug-in estimators (*empirical estimators*), over $\Delta_\mathcal{X}$ for some *finite* alphabet $\mathcal{X}$. Note that $\Delta_\mathcal{X}$ is a special symmetric distribution collection.

For example, the empirical estimator for Shannon entropy has a worst-case error rate of $\Theta(|\mathcal{X}|/n)$, whereas the minimax error rate is $\Theta(|\mathcal{X}|/(n \log n))$ [Valiant and Valiant, 2011, 2013, Jiao et al., 2015, Wu and Yang, 2016, Acharya et al., 2017, Hao and Orlitsky, 2019a,c, 2020]. Similar results also hold for support size and $\ell_1$ distance to the uniform distribution over $\mathcal{X}$ (See [Valiant and Valiant, 2011, 2013, Acharya et al., 2017, Jiao et al., 2018, Wu et al., 2019, Hao and Orlitsky, 2019a,c, 2020]). One observation is that all these properties are in the form of $\sum_x f_r(p_x)$, where $f_r$ is a relative smooth real function (for support size, one needs a lower bound like $1/|\mathcal{X}|$ on the positive probabilities, which effectively smoothes the function).

It is apparent that most symmetric properties are not in the $\sum_x f_r(p_x)$ form. A simple example is Rényi entropy, for which the learning error rates exhibit a significantly different behavior. Specifically, for a power parameter $\alpha > 1, \alpha \in \mathbb{N}$, the minimax error of inferring Rényi entropy varies according to $|\mathcal{X}|$ and $n$ as follows [Acharya et al., 2016].

If $n \lesssim |\mathcal{X}|^{1-1/\alpha}$ (sample-sparse regime), then $r_n(\Delta_{\mathcal{X}}) \gtrsim \max_p f(p)$ (consistent estimation is impossible); if $n \gtrsim |\mathcal{X}|^{1+1/\alpha}$ (large-sample regime), then $r_n(\Delta_{\mathcal{X}}) \simeq (|\mathcal{X}|^{1-1/\alpha}/n)^{1/2}$, which is *achieved* by the empirical estimator (trivial regime); if $|\mathcal{X}|^{1-1/\alpha} \lesssim n \lesssim |\mathcal{X}|^{1+1/\alpha}$, then the empirical estimator has an order $\max\{|\mathcal{X}|/n, 1\}$ worst-case error, whereas the minimax error is $(|\mathcal{X}|^{1-1/\alpha}/n)^{1/2}$ (potentially much lower than that of empirical).

The recent work of Hao and Orlitsky [2019a] significantly extends our understanding of symmetric property estimation by showing that the PML estimator is sample optimal for all $\sum_x f_r(p_x)$ properties that are approximately *Lipschitz*, and is as good as the best known estimators for Rényi entropy of power $\alpha > 3/4$. The paper also presents resulting on other tasks such as testing.

Given the special structures, even the combination of all the properties mentioned above corresponds to only an extremely small subclass of symmetric properties. The general landscape for how the worst-case error rate behaves when we consider either the empirical or the minimax estimator is far from understood, even for just $\Delta_{\mathcal{X}}$. In fact, even for Rényi entropy, a simple and widely studied property, the minimax rates are not fully characterized – the lower and upper bounds in Acharya et al. [2016] for non-integer powers do not match in all parameter regimes. Ideally, there should be a set of formulas such that once the explicit form of $f$ is available, the respective error rates can be computed, and more importantly, an explicit algorithm can be derived.

Our result pushes forward the general understanding of symmetric property estimation. It leverages the method of PML to derive competitive learning guarantees for all symmetric properties and distribution collections. The theorem even adapts itself to individual distributions, leading to numerous nontrivial estimation results without introducing sophisticated analysis or additional algorithms.

**Methods**   As the task involves two components, the property and distribution (probability multiset), the design of statistical methods also advances in two veins.

The first vein concerns constructing a universal plug-in estimator for all *symmetric properties*. A symmetric property is invariant under symbol permutations, hence it suffices to obtain an accurate estimate of the probability multiset.

One method is PML, the approach that our theorem adopts. Recently, following the papers by Das [2012], Acharya et al. [2017], the work of Hao and Orlitsky [2019a] shows that for any symmetric property that is in the form of $\sum_x f_r(p_x)$ and appropriately Lipschitz, both the profile maximum likelihood [Orlitsky et al., 2004] and its near-linear-time computable variant in Charikar et al. [2019b] achieve the optimal sample complexity up to small constant factors.

Another method is moment matching via linear programming (LP). In typical works using LP, such as Valiant and Valiant [2011, 2013, 2016], Han et al. [2018], one first estimates the (lower-order) moments of the underlying distributions (e.g., $\sum_x p_x^i$ for $i \leq \log n$), which are also symmetric properties, and then finds a distribution through an LP method (up to domain-symbol permutations), whose lower order moments match with the estimates. These methods are known to achieve the minimax error rates over $\Delta_{\mathcal{X}}$ for only a few specific properties, such as entropy, support size (also assume a $1/|\mathcal{X}|$ lower bound on the positive probabilities), and $\ell_1$-distance to the uniform distribution.

The second vein of methods addresses the bias of empirical estimators and (often partially) replaces the given property by a bias-corrected polynomial, for which we can efficiently construct a near-unbiased estimator. There are mainly three different types of constructions for the bias-corrected polynomial: using classical minimax approximation [Jiao et al., 2015, 2018, Wu and Yang, 2016, Wu et al., 2019, Hao and Orlitsky, 2019c], applying smoothing techniques to the coefficients of the unbiased estimator [Orlitsky et al., 2016, Hao et al., 2018, Hao and Li, 2020], and computing the derivative of the (property's) Bernstein polynomial and employing the integral of its minimax approximation [Hao and Orlitsky, 2020].

Early works in this direction address specific properties, such as entropy [Jiao et al., 2015, Wu and Yang, 2016], support size [Wu et al., 2019], support coverage [Orlitsky et al., 2016], and $\ell_1$-distance to the uniform distribution [Jiao et al., 2018], and determine their respective minimax error rates. Recent works consider broader families of properties [Hao et al., 2018, Hao and Orlitsky, 2019c, 2020, Hao and Li, 2020], in particular those in the $\sum_x f_r(p_x)$ form and appropriately Lipschitz. Besides these results, some state-of-the-art Rényi entropy estimators [Acharya et al., 2016] also use polynomial approximation. Excluding properties in these special forms, it is unknown whether these techniques/methods work for the large amount of symmetric properties in general, even just over $\Delta_{\mathcal{X}}$.

**Outline** The rest of Appendix B presents the proof of the our result on PML. For clarity, we divide the full proof into three parts: a) the sufficiency of profiles for estimating symmetric properties (already established above); b) the standard "median trick" often used to boost the confidence of learning algorithms; c) the PML method and its competitiveness to the min-max estimators. As one may expect, the proof utilizes several previously established results.

## B.4 Proof of Theorem 4

**Proof outline** We begin with a proof sketch on the high level. While our theorem states only a constant-error-probability result for the vanilla PML, the guarantee holds for approximations of PML and any general error probability bound $\delta$, and this outline corresponds to the general setting.

1. For simplicity, let $k$ denote the (expected or high-probability) dimension of a length-$n$ profile from an unknown $p \in \Delta$, and refer to the actual random quantity $\mathcal{D}_n \sim p$ as "dimension".

2. Let's say $p \in \mathcal{P}$ (which is symmetric), and we have an $m$-sample estimator over $\mathcal{P}$ with an $(\varepsilon, \delta)$ guarantee, i.e., for every distribution in $\mathcal{P}$, the estimator learns its property value up to an $\varepsilon$ error, with probability at least $1 - \delta$. In addition, we assume that $m \ll n$ with the ratio $r := n/m$ to be determined.

3. Now, assume that $r$ has been properly chosen, and we could utilize at most $r$ copies of the $m$-sample estimator to construct an $n$-sample $(\varepsilon, \delta \cdot \exp(-2k))$ estimator (the existence of $r$ follows by the standard "median trick"). Furthermore, by the sufficiency of profiles (Theorem 3), there is a profile-based estimator that achieves the same guaranty.

4. Divide all length-$n$ profiles into two groups: one group with dimension at most of order $k$ (hiding logarithmic factors), the other with dimension much larger than $k$.

5. By the concentration of sample profile dimensions (e.g., Theorem 9), the profile of an arbitrary sample from $p$ belongs to the first group with high probability (say at least $1 - 1/n$), we can safely ignore the second group.

6.1 Pick a profile from "the first group", if its probability is $\gg \delta \cdot \exp(-k)$, the approximate PML (APML) will have a probability of $\gg \delta \cdot \exp(-2k)$. Here, the definition of APML is based on profile probabilities – for every length-$n$ sample, its profile probability under the true distribution and the APML estimate should differ by a factor of at most $\exp(k)$ (more generally, a fixed factor of at least 1, which covers the vanilla PML). This definition is analogous to those in Acharya et al. [2017] and Charikar et al. [2019a,b].

6.2 So, the *profile-based estimator* must work properly on both distributions, the original and the APML. Triangle inequality then relates the property values of these distributions (by eliminating the estimator's value) and yields a $2\varepsilon$ estimation guarantee for the APML.

7.1 On the other hand, if the profile we picked has a probability at most $\delta \cdot \exp(-k)$, then the APML may fail, i.e., not produce a reasonable estimate.

7.2 However, there are at most (ignore logarithmic factors in the exponent) $\exp(k)$ such profiles, hence by the union bound, the total probability of failing is at most $\delta + 1/n$.

8. Finally, we tune parameter $r$, which becomes something like $k$, up to logarithmic factors. Utilizing our entropy-dimension equivalence (Theorem 1) completes the proof.

**Median Trick** The following argument is standard method for boosting the confidence of learning algorithms, commonly known as the *median trick*.

**Lemma 7** (Median trick). *Let $\alpha, \beta \in (0,1)$ be real parameters satisfying $1/10 \geq \alpha > \beta$. For an accuracy $\varepsilon > 0$ and a distribution set $\mathcal{P} \subseteq \Delta$, if there exists an estimator $\hat{f}_A$ such that*

$$\Pr_{X^n \sim p} \left( \left| \hat{f}_A(X^n) - f(p) \right| > \varepsilon \right) < \alpha, \ \forall p \in \mathcal{P},$$

*we can construct another estimator $\hat{f}_B$ that takes a sample of size $m := \left\lceil \frac{4n}{\log \frac{1}{2\alpha}} \log \frac{1}{\beta} \right\rceil$ and achieves*

$$\Pr_{Y^m \sim p} \left( \left| \hat{f}_B(Y^m) - f(p) \right| > \varepsilon \right) < \beta, \ \forall p \in \mathcal{P}.$$

*Proof.* Given $t \in \mathbb{N}$ i.i.d. copies of $\hat{f}_A(X^n)$, the probability that less than half of them satisfy the inequality in the parentheses is at least

$$\Pr\left(\sum_{i=1}^{t} \mathbb{1}_{A_i} < \frac{t}{2} \text{ for } A_i\text{'s satisfying } \Pr(A_i) < \alpha\right) \geq \Pr\left(\text{bin}(t, \alpha) < \frac{t}{2}\right).$$

By the law of total probability, the right-hand side equals to

$$1 - \Pr\left(\text{bin}(t, \alpha) \geq \frac{t}{2}\right) \geq 1 - \exp\left(\left(\left(\frac{1}{2\alpha} - 1\right) - \frac{1}{2\alpha} \log \frac{1}{2\alpha}\right) \cdot \alpha t\right)$$

$$\geq 1 - \exp\left(-\frac{t}{4} \log \frac{1}{2\alpha}\right),$$

where the first step follows by the Chernoff bound of binomial random variables, and the second step follows by $\alpha \leq 1/10$ and the inequality $c - 1 - \frac{c}{2} \log c > 0, \forall c \geq 5$.

Set $t := \left\lceil \frac{4}{\log \frac{1}{2\alpha}} \log \frac{1}{\beta} \right\rceil$, the right-hand side is at least $1 - \beta$.

Therefore, given a sample of size $m = t \cdot n$, we can partition it into $t$ sub-samples of equal size, apply the estimator $\hat{f}_A$ to each subsample, and define the median of the corresponding estimates as $\hat{f}_B$.

By the previous reasoning, this estimator satisfies

$$\Pr_{Y^m \sim p}\left(\left|\hat{f}_B(Y^m) - f(p)\right| > \varepsilon\right) < \beta, \; \forall p \in \mathcal{P}. \qquad \square$$

*Proof of the theorem.* For any tolerance $\delta \in (0, 1)$ and distribution $p \in \Delta$, define the $(\delta, n)$-*typical cardinality of profiles with respect to* $p$ as the smallest cardinality $C_{\delta,n}(p)$ of a set of length-$n$ profiles such that the probability of observing a sample from $p$ with a profile in this set is at least $1 - \delta$. The following lemma provides a tight characterization of $C_{\delta,n}(p)$ in terms of the dimension of $\Phi^n \sim p$.

**Lemma 8.** *For any $p \in \Delta$ and $\Phi^n \sim p$, with probability at least $1 - 6/\sqrt{n}$,*

$$C_{\frac{6}{\sqrt{n}}, n}(p) \leq n^{8(\mathcal{D}_n + 20 \log n)}.$$

The proof of the lemma follows by recursively applying Theorem 9. Specifically, let $d := 2E_n(p) + 3 \log n$, which is at least $\mathcal{D}_n \sim p$, with probability at least $1 - 6/\sqrt{n}$. Then,

$$C_{\frac{6}{\sqrt{n}}, n}(p) \leq \binom{n}{d}\binom{n+d-1}{d-1} \leq n^{2d-1} \leq n^{2(2E_n(p)+3\log n)} \leq n^{8\mathcal{D}(\Phi^n)+20\log n},$$

where the last inequality holds with with probability at least $1 - 6/\sqrt{n}$.

Now, let $f$ be a symmetric property over $\mathcal{P}$. For simplicity, we will establish the theorem for the vanilla PML, since as our *proof outline* shows, the proof for any approximate PML (APML) is essentially the same. In addition, for a sequence $x^n$ with profile $\phi := \varphi(x^n)$, we write $\mathcal{P}_\phi$ for the PML estimate $\mathcal{P}_\varphi(x^n)$. According to Theorem 3, for any parameters $\varepsilon > 0$ and $\delta \in (0, 1)$, if there exists an estimator $\hat{f}$ such that

$$\Pr_{X^n \sim p}\left(\left|\hat{f}(X^n) - f(p)\right| > \varepsilon\right) < \delta, \; \forall p \in \mathcal{P},$$

there is an estimator $\hat{f}_\varphi$ over profiles satisfying

$$\Pr_{X^n \sim p}\left(\left|\hat{f}_\varphi(\varphi(X^n)) - f(p)\right| > \varepsilon\right) < \delta, \; \forall p \in \mathcal{P}.$$

For an arbitrary length-$n$ profile $\phi$ that satisfies $\Pr_{\Phi^n \sim p}(\Phi^n = \phi) \geq 2\delta$, these error bounds yield $\Pr(|\hat{f}_\varphi(\phi) - f(p)| > \varepsilon) < \frac{1}{2}$, and since $\Pr_{\Phi^n \sim \mathcal{P}_\phi}(\Phi^n = \phi) \geq \Pr_{\Phi^n \sim p}(\Phi^n = \phi) \geq 2\delta$ by the definition of PML (as we take the distribution that maximizes the probability),

$$\Pr\left(\left|\hat{f}_\varphi(\phi) - f(\mathcal{P}_\phi)\right| > \varepsilon\right) < \frac{1}{2}.$$

By the union bound and triangle inequality,

$$\Pr\left(|f(p) - f(\mathcal{P}_\phi)| > 2\,\varepsilon\right) < 1 \iff |f(p) - f(\mathcal{P}_\phi)| \le 2\,\varepsilon \text{ surely.}$$

Furthermore, by Lemma 8, with probability at least $1 - 6/\sqrt{n}$, the total probability of length-$n$ profiles $\phi$ satisfying $\Pr_{\Phi^n \sim p}(\Phi^n = \phi) < 2\delta$ is at most

$$2\delta \cdot C_{\frac{6}{\sqrt{n}}, n}(p) + \frac{6}{\sqrt{n}} \le 2\delta \cdot n^{8\mathcal{D}_n + 20 \log n} + \frac{6}{\sqrt{n}},$$

which basically upperly bounds the probability that $|f(p) - f(\mathcal{P}_{\Phi^n})| > 2\,\varepsilon$. Next we will assume that there exists an estimator $\hat{f}$ satisfying $\Pr_{X^m \sim p}(|\hat{f}(X^m) - f(p)| > \varepsilon) < \delta$, $\forall p \in \mathcal{P}$. By Lemma 7, if $\delta \le 1/10$, we can construct another estimator $\hat{f}'$ that takes a sample of size $n = \frac{4m}{\log \frac{1}{2\delta}} \log \frac{1}{\delta'}$ ($n$ is assumed to be an integer here) and achieves a higher-confidence guarantee

$$\Pr_{X^n \sim p}\left(\left|\hat{f}'(X^n) - f(p)\right| > \varepsilon\right) < \delta', \ \forall p \in \mathcal{P}.$$

Then by the above reasoning, with probability at least $1 - 6/\sqrt{n}$,

$$\Pr_{\Phi^n \sim p}(|f(p) - f(\mathcal{P}_{\Phi^n})| > 2\,\varepsilon) \le 2\delta' \cdot n^{8\mathcal{D}_n + 20 \log n} + \frac{6}{\sqrt{n}}$$

$$= 2 \exp\left(-\frac{n}{4m} \log \frac{1}{2\delta} + (8\mathcal{D}_n + 20 \log n) \log n\right) + \frac{6}{\sqrt{n}}.$$

For the first term on the right hand side to vanish as quickly as $1/\sqrt{n}$, it suffices to have

$$\frac{n}{4m} \log \frac{1}{2\delta} \ge 20 \cdot \mathcal{D}_n \log n \text{ and } \frac{n}{4m} \log \frac{1}{2\delta} \ge 40 \cdot \log^2 n.$$

Simplifying the expressions and applying the union bound yield that $|f(p) - f(\mathcal{P}_{\Phi^n})| \le 2\,\varepsilon$ with probability at least $1 - 1/\sqrt{n}$, given both

$$\frac{n}{\mathcal{D}_n} \gtrsim \frac{m}{\log \frac{1}{\delta}} \text{ and } n \ge 8m. \qquad \square$$

## B.5 Experiments

Prior works such as Hao and Orlitsky [2019a], Pavlichin et al. [2019] have experimentally demonstrated the efficiency of PML on estimating several classical properties, including the Shannon and Rényi entropy, support size, and $\ell_1$ distance to the uniform distribution. Our result further extends and establishes the efficiency of PML for numerous symmetric properties that are under-explored. Given the broadness of this property class, the potential applications are countless.

Consider a variant of Shannon entropy, $f(p) := \sum_x p_x \log^2 p_x$, that mildly puts more emphasis on small probabilities. As the property is relatively new and non-Lipschitz, prior works and approaches do not easily yield a satisfiable learning guarantee. Our result hence comes into play, because $f$ is symmetric, which suffices for Theorem 4 to take effect. Below, we will estimate this property by an $n$-sample PML plug-in, and compare its performance to two estimators: the $n$-sample empirical estimator that evaluates the entropy of the empirical distribution, serving as a baseline, and the $10n$-sample empirical estimator whose sample size is larger than others by *an order of magnitude*.

We considered six natural distributions: uniform, Zipf(1/2), Zipf(2), Dirichlet(1)-drawn-, Dirichlet(2)-drawn-, and geometric, all having support size $k = 5,000$. The plots are presented in Figure 1, with both vertical and horizontal axes showing in *log-scale* (base 10). The sample size $n$ ranges from $10^3$ to $10^5$, and every data point represents the average absolute error over 20 independent simulations.

Specifically, the geometric distribution has a success probability of $(k-1)/k$; the Zipf(1/2) and Zipf(2) distributions have probability $p_i \propto i^{-1/2}$ and $p_i \propto i^{-2}$ for $i \ge 1$, both being truncated at $k$ and re-normalized; drawing a distribution from the Dirichlet(1) prior is equivalent to drawing one uniformly from the $k$-dimensional standard simplex.

As the experiments demonstrate, the PML plug-in estimator significantly improves over the empirical estimator (note that the axes are in log-scale) and is as good as an estimator having access to samples larger by order of magnitudes. There are multiple PML implementations and we have used the one by Hao and Orlitsky [2019a] (Section 4 of that paper presents a list of PML computation algorithms). Code is included in the supplementary material. For instructions on how to use the code, please refer to the inline comments and Section 4.1 in the supplementary material of Hao and Orlitsky [2019a].

Figure 1: Inferring property $f$ via the PML plug-in. For clarity, both the horizontal axis (sample size) and the vertical axis (average absolute error) are in the log-10 scale.

## C   Competitive Estimation of Distributions and Their Entropy

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

## C.2    Proof of Theorem 2

*Proof.*  The upper bound follows by the main result of Hao and Orlitsky [2019b] and Theorem 1 asserting the entropy-dimension equivalence. To establish the lower bound, denote $s := (H/\log n)^{1/2}$, $I := \{s, s+1, \ldots, 2s\}$, and $P := \cup_{i \in I} P_i := \cup_{i \in I} U_i/n$ where

$$U := \bigcup_{i \in I} U_i := \bigcup_{i \in I} \{i^2 \log^2 n, i^2 \log^2 n + 1, \ldots, i^2 \log^2 n + i \log n\},$$

where $H \lesssim \sqrt{n/\log n}$ for the total to be at most $n$. Let $A \cdot \{B\}$ denote the length-$A$ constant sequence of value $B$. Let $C$ be the set of distributions in the form of

$$p := L \cdot \left\{ \frac{1}{n^2} \right\} \bigcup \left( \bigcup_i (i \log n) \cdot \{q_i \text{ or } q_i' : nq_i = i^2 \log^2 n, nq_i' = i^2 \log^2 n + i \log n \} \right).$$

where the probability values are sorted according to the ordering they appear above, $L$ is a proper variable that makes the probabilities sum to 1, and the range of support of distribution $p$ is irrelevant for our purpose and hence unspecified. Equip a uniform prior over $C$ (equivalently, construct a random distribution). We have several claims in order:

- For any $i \in I$ and $\mu \in U_i$, by the construction and independence,

$$\Pr(\varphi_\mu = 1 | q_i \text{ is chosen}) \approx (i \log n) \cdot \left( \Pr(\mathrm{Poi}(nq_i) = \mu) \cdot (\Pr(\mathrm{Poi}(nq_i) \ne \mu))^{i \log n - 1} \right)$$

$$\approx (i \log n) \cdot \left( \frac{1}{\sqrt{nq_i}} \cdot \left( 1 - \frac{1}{\sqrt{nq_i}} \right)^{i \log n - 1} \right)$$

$$\ge \Omega(1).$$

  Similarly, we have $\Pr(\varphi_\mu = 1 | q_i' \text{ is chosen}) \ge \Omega(1)$. Hence,
$$\Pr(\varphi_\mu = 1) \ge \Omega(1).$$

- For any $i \in I$ and $\mu \in U_i$, by Bayes' rule,

$$\Pr(q_i \text{ is chosen}|\varphi_\mu = 1) = \frac{\Pr(\varphi_\mu = 1|q_i \text{ is chosen}) \cdot 0.5}{\Pr(\varphi_\mu = 1)} \geq \Omega(1).$$

  Similarly, we have $\Pr(q_i' \text{ is chosen}|\varphi_\mu = 1) \geq \Omega(1)$.

- For any $i \in I$ and $\mu \in U_i$, the value of $M_\mu$, the total probability of symbols appearing $\mu$ times, is $q_i$ if $\varphi_\mu = 1$ and $q_i$ is chosen; and is $q_i'$ if $\varphi_\mu = 1$ and $q_i$ is chosen. Any estimator $E_\mu$ will incur an expected absolute error of $\Omega(i(\log n)/n)$ in estimating $M_\mu$ given $\varphi_\mu = 1$.

- Note that for any $\alpha \in [0,1]$ and $x, y > 0$,

$$\alpha(y - z)^2 + (1 - \alpha)(z - x)^2 \geq \alpha(1 - \alpha)(x - y)^2.$$

- Therefore, the expected squared Hellinger distance $\mathbb{H}^2(\cdot, \cdot)$ of any estimator $E_\mu$ in estimating $(M_\mu)_{\mu \geq 0}$ satisfies, by the linearity of expectation,

$$\frac{1}{2} \sum_{\mu \geq 0} \mathbb{E}\left(\sqrt{E_\mu} - \sqrt{M_\mu}\right)^2 \geq \frac{1}{2} \sum_{i \in I} \sum_{\mu \in U_i} \mathbb{E}\left[\left(\sqrt{E_\mu} - \sqrt{M_\mu}\right)^2 \Big| \varphi_\mu = 1\right] \Pr(\varphi_\mu = 1)$$

$$= \frac{1}{2} \sum_{i \in I} \sum_{\mu \in U_i} \mathbb{E}\left[\left(\frac{E_\mu - M_\mu}{\sqrt{E_\mu} + \sqrt{M_\mu}}\right)^2 \Big| \varphi_\mu = 1\right] \Pr(\varphi_\mu = 1)$$

$$\geq \sum_{i \in I} (i \log n) \cdot \Omega\left(\frac{(i \log n)/n}{\sqrt{i^2(\log^2 n)/n}}\right)^2$$

$$\geq s \cdot \Omega\left(\frac{s \log n}{n}\right)$$

$$= \Omega\left(\frac{H}{n}\right).$$

- Consequently, by the inequality $\mathrm{D}(P \,\|\, Q) \geq 2\mathbb{H}^2(P, Q)$,

$$\mathbb{E}\left[\mathrm{D}(E \,\|\, M)\right] \geq \mathbb{E}\left[2\mathbb{H}^2(E, M)\right] \geq \Omega\left(\frac{H}{n}\right).$$

- Finally, combining Theorem 1, 9 and 11 yields that, with high probability,

$$\mathcal{H}_n(p) \simeq \mathcal{D}_n \simeq E_n(p) \simeq \mathcal{H}_n^S(p) = \sum_{j \geq 1} \min\left\{p_{I_j}, j \cdot \log n\right\},$$

  which, by our definition, is at most $\mathcal{O}(\log n + s(s \log n)) = \mathcal{O}(\log n + H)$. $\qquad \square$

### C.3 Extension: Competitive Entropy Estimation

Recall that a distribution estimator is *natural* if it assigns the same probability to symbols of equal multiplicity, and a property estimator is *plug-in* if it first finds an estimate of the distribution and then evaluates the property at this estimate. As an off-the-shelf method, the plug-in approach is widely used in estimating distribution properties.

As we mentioned in Appendix B.3, to estimate a symmetric property, an accurate estimate of the probability multiset of the underlying distribution suffices. Intuitively, it should be easier in terms of statistical efficiency to recover just the probability mutiset than to learn the entire distribution. For example, over distribution collection $\Delta_\mathcal{X}$, the PML plug-in estimator is minimax optimal for learning entropy, while the empirical distribution, being minimax optimal for distribution estimation, is suboptimal as a plug-in entropy estimator.

However, the analysis and computation (though efficient) of such multiset-based estimation methods are often involved [Valiant and Valiant, 2011, 2013, 2016, Han et al., 2018, Charikar et al., 2019b,

Acharya et al., 2017, Hao and Orlitsky, 2019a]. For this reason, plug-in estimators that first estimate the true distribution are still popular in practice, and often, the distribution component is natural.

For example, several notable and widely used entropy estimators are *natural plug-in*, including the empirical estimator that simply uses the empirical distribution, James-Stein shrinkage [Hausser and Strimmer, 2009] that shrinks the distribution estimate towards uniform, and Dirichlet-smoothed [Schürmann and Grassberger, 1996] that imposes a Dirichlet prior over $\Delta_\mathcal{X}$.

The logic behind these estimators is simple – if two distributions (e.g., the true distribution and our estimate) are close, the same is expected for their entropy values. The next theorem shows that for *every* distribution and among all plug-in entropy estimators, the distribution estimator in Hao and Orlitsky [2019b] is as good as the one that performs best in estimating the actual distribution.

Denote by $\mathcal{N}$ the collection of all natural estimators, and write $|H(p) - H(q)|$ as $\ell_H(p, q)$.

**Theorem 13** (Competitive entropy estimation). *For any distribution $p$, sample $X^n \sim p$, and the respective best natural estimator $\hat{p}_{X^n}^\mathcal{N} := \arg\min_{\hat{p} \in \mathcal{N}} \mathrm{D}(p \Vert \hat{p}_{X^n})$, with probability at least $1 - 1/n$,*

$$\ell_H(p, \hat{p}_{X^n}^\star) - \ell_H(p, \hat{p}_{X^n}^\mathcal{N}) \leq \tilde{\mathcal{O}}\left(\sqrt{\frac{\mathcal{H}_n(p)}{n}}\right).$$

*Proof.* Given any natural estimator and a sample $X^n \sim p$, we denote by $q$ the distribution estimate. The entropy of $q$ differs from the true entropy by

$$
\begin{aligned}
H(q) - H(p) &= -\sum_x q_x \log q_x + \sum_x p_x \log p_x \\
&= \sum_x p_x \log p_x - \sum_x p_x \log q_x + \sum_x p_x \log q_x - \sum_x q_x \log q_x \\
&= \sum_x p_x \log \frac{p_x}{q_x} + \sum_x (p_x - q_x) \log q_x \\
&= \mathrm{D}(p \Vert q) + \sum_x (p_x - q_x) \log q_x.
\end{aligned}
$$

Denote by $P_\mu(X^n)$ and $Q_\mu(X^n)$ the total probability that distributions $p$ and $q$ assign to symbols with multiplicity $\mu$. Since $q$ is induced by a natural estimator, we also write $q_\mu(X^n)$ for the probability that $q$ assigns to *each* symbol with multiplicity $\mu$ in $X^n$. Recall that prevalence $\varphi_\mu(X^n)$ denotes the number of symbols with multiplicity $\mu$ in $X^n$. Therefore, $Q_\mu(X^n) = \varphi_\mu(X^n) \cdot q_\mu(X^n)$.

Henceforth, whenever it is clear from the context, we suppress $X^n$ in related expressions. Then, the second term on the right-hand side satisfies

$$
\begin{aligned}
\sum_x (p_x - q_x) \log q_x &= \sum_x \left(\sum_\mu \mathbb{1}_{\mu_x = \mu} \cdot p_x - \sum_\mu \mathbb{1}_{\mu_x = \mu} \cdot q_\mu\right) \log\left(\sum_\mu \mathbb{1}_{N_x = \mu} \cdot q_\mu\right) \\
&= \sum_x \sum_\mu \mathbb{1}_{\mu_x = \mu} \cdot (p_x - q_\mu) \log q_\mu \\
&= \sum_\mu \left(\sum_x \mathbb{1}_{\mu_x = \mu} \cdot p_x - \sum_x \mathbb{1}_{\mu_x = \mu} \cdot q_\mu\right) \log q_\mu \\
&= \sum_\mu (P_\mu - Q_\mu) \log q_\mu.
\end{aligned}
$$

Let $q_{\min}$ be the smallest nonzero probability of $q$. By the triangle inequality and Pinsker's inequality,

$$
\begin{aligned}
\left|\sum_\mu (P_\mu - Q_\mu) \log q_\mu\right| &\leq \sum_\mu |(P_\mu - Q_\mu) \log q_\mu| \\
&\leq |\log q_{\min}| \sum_\mu |P_\mu - Q_\mu| \\
&\leq |\log q_{\min}| \sqrt{2\mathrm{D}(P \Vert Q)}.
\end{aligned}
$$

For simplicity, suppress the subscript $X^n$ from all estimators, e.g., write $\hat{p}^{\mathcal{N}} := \hat{p}^{\mathcal{N}}_{X^n}$. Now we show that if a symbol $x$ has multiplicity $\mu$, the estimator $\hat{p}^{\mathcal{N}}$ will assign a probability mass of $P_\mu/\varphi_\mu$. In other words, $\hat{P}^{\mathcal{N}}_\mu = P_\mu$ since $p^{\mathcal{N}} \in \mathcal{N}$. Indeed, the corresponding KL-divergence values differ by

$$\sum_x p_x \log \frac{p_x}{q_x} - \sum_x \sum_\mu \mathbb{1}_{\mu_x = \mu} \cdot p_x \log \frac{p_x}{P_\mu/\varphi_\mu} = \sum_x p_x \log \frac{1}{q_x} - \sum_x \sum_\mu \mathbb{1}_{\mu_x = \mu} \cdot p_x \log \frac{\varphi_\mu}{P_\mu}$$

$$= \sum_x \sum_\mu \mathbb{1}_{\mu_x = \mu} \cdot p_x \log \frac{P_\mu}{\varphi_\mu q_\mu}$$

$$= \sum_\mu P_\mu \log \frac{P_\mu}{Q_\mu} = \mathrm{D}(P \| Q) \geq 0.$$

Then, the above equalities yield that,

$$H(\hat{p}^{\mathcal{N}}) - H(p) = \mathrm{D}(p \| \hat{p}^{\mathcal{N}}) + \sum_\mu \left( P_\mu - \hat{P}^{\mathcal{N}}_\mu \right) \log p^{\mathcal{N}}_\mu = \mathrm{D}(p \| \hat{p}^{\mathcal{N}}).$$

Next consider the other estimator $\hat{p}^\star$, which is also natural. Let $\mathcal{D}_n = \mathcal{D}_n$ be the profile dimension of $X^n$. By the results in Hao and Orlitsky [2019b], estimator $\hat{p}^\star$ achieves a $\mathcal{D}_n/n$ excess loss, i.e.,

$$\mathrm{D}(p \| \hat{p}^\star_{X^n}) - \min_{\hat{p} \in \mathcal{N}} \mathrm{D}(p \| \hat{p}_{X^n}) = \mathrm{D}(P \| \hat{P}^\star) \leq \tilde{\mathcal{O}}\left( \frac{\mathcal{D}_n}{n} \right),$$

for every $p$ and $X^n \sim p$, with probability at least $1 - \mathcal{O}(1/n)$. In addition, by its construction, the minimum probability of $\hat{p}_{X^n}$ is at least $1/n^4$. Therefore, with probability at least $1 - \mathcal{O}(1/n)$,

$$\left| \sum_x (p_x - \hat{p}^\star_x) \log \hat{p}^\star_x \right| = \left| \sum_\mu \left( P_\mu - \hat{P}^\star_\mu \right) \log \hat{p}^\star_\mu \right| \leq |\log \hat{p}^\star_{\min}| \cdot \sqrt{2\mathrm{D}(P \| \hat{P}^\star)} \leq \tilde{\mathcal{O}}\left( \sqrt{\frac{\mathcal{D}_n}{n}} \right).$$

Finally, the triangle inequality combines the above results and yields

$$\ell_H(p, \hat{p}^\star) - \ell_H(p, \hat{p}^{\mathcal{N}}) = |H(p) - H(\hat{p}^\star)| - |H(p) - H(\hat{p}^{\mathcal{N}})|$$

$$= \left| \mathrm{D}(p \| \hat{p}^\star_x) + \sum_x (p_x - \hat{p}^\star_x) \log \hat{p}^\star_x \right| - \left| \min_{\hat{p} \in \mathcal{N}} \mathrm{D}(p \| \hat{p}) \right|$$

$$\leq \left| \mathrm{D}(p \| \hat{p}^\star_x) - \min_{\hat{p} \in \mathcal{N}} \mathrm{D}(p \| \hat{p}) \right| + \left| \sum_x (p_x - \hat{p}^\star_x) \log \hat{p}^\star_x \right|$$

$$= \mathrm{D}(P \| \hat{P}^\star_\mu) + \tilde{\mathcal{O}}\left( \sqrt{\frac{\mathcal{D}_n}{n}} \right)$$

$$\leq \tilde{\mathcal{O}}\left( \sqrt{\frac{\mathcal{D}_n}{n}} \right).$$

This together with Theorem 1 completes the proof. □

## C.4 Experiments

The experiments in Hao and Orlitsky [2019b] have demonstrated the efficiency of $\hat{p}^\star$, showing that the estimator frequently and uniformly outperforms an improved version of the well-known Good-Turing estimation scheme [Orlitsky and Suresh, 2015], for numerous distributions and parameter settings. Our results confirmed the optimality of estimator $p^\star$ from a theoretical point of view, and moves forward considerably our understanding of how well one can approach the performance of a genie having the full knowledge of the true distribution, but restricted to be natural as all human beings.

In the following, we do not repeat the experiments in Orlitsky and Suresh [2015] (see Section 2 of its supplementary), and instead, investigate a novel and highly related task – employing $\hat{p}^\star$ as a plug-in estimator for Shannon entropy. By Theorem 13 and its proof, we already see that the resulting plug-in estimator $H \circ \hat{p}^\star$ is as good as any plug-in estimator with a natural distribution component, and how

well it performs, to a certain extent, depends on how well it approximates the true distribution under the KL divergence. But is this plug-in estimator still competitive when compared to estimators having observed samples of much larger sizes, or to the state-of-the-art estimators that are designed just for entropy estimation? The following experiments answered this question in the affirmative.

Below we demonstrate the efficiency of $\hat{p}^\star$ when used as a plug-in entropy estimator. We will compare its performance with a size-$n$ sample to three estimators: the $n$-sample *empirical* estimator that evaluates the entropy of the empirical distribution, the $n \log n$-sample empirical estimator that has access to much more information, and a state-of-the-art entropy estimator in Wu and Yang [2016] based on minimax polynomial approximations (which we refer to as WY). Shown by the experiments in Wu and Yang [2016], under numerous settings, the WY estimator frequently outperformed several classical estimators and other minimax estimators such as Valiant and Valiant [2011, 2013], Jiao et al. [2015]. Hence, we maintain simplicity and do not compare our approach to the latter ones.

We considered six natural distributions: uniform, two-steps-, Zipf(1/2), binomial, geometric, and Dirichlet(1)-drawn-, all having support size $k = 5,000$. The plots are presented in Figure 2, with both vertical and horizontal axes showing in *log-scale* (base 10). The sample size $n$ ranges from $10^3$ to $10^5$, and every data point represents the average absolute error over 20 independent simulations. We refer to the plug-in estimator using $\hat{p}^\star$ as *HO*.

Specifically, 10% probability values of the two-steps distribution $\propto 9/k$, and the rest $\propto 1/k$; the binomial and geometric distributions have success probabilities of $10/k$ and $(k-1)/k$, respectively; the Zipf(1/2) distribution has probability $p_i \propto i^{-1/2}$ for $i \geq 1$, and is truncated at $k$ and re-normalized.

We see that the performance of the WY estimator and our plug-in approach are essentially the same. In particular, for Dirichlet(1)-drawn-, WY is better, but for binomial, WY is worse; for all other cases, the two error curves basically follow the same trend and lie in the same region. This is somewhat surprising since intuitively, $\hat{p}^\star$ is a distribution estimator and its design has no consideration about entropy estimation, while WY is geared towards this task. On the other hand, the performance of the induced plug-in estimator should be both efficient and competitive, as guaranteed by Theorem 13.

Figure 2: Competitive entropy estimation. For clarity, both the horizontal axis (sample size) and the vertical axis (average absolute error) are in the log-10 scale.

# D  Optimal Characterization for Structured Families

Following the previous discussions, we will derive nearly tight bounds on $\mathcal{H}_n(p)$ for three important structured families – log-concave, power-law, and histogram. These bounds clearly demonstrate the power of profile entropy in charactering natural shape constraints.

For the subsections below, we adopt the convention of specifying structured distributions over $\mathcal{X} = \mathbb{Z}$.

## D.1  Theorem 6: Log-Concave Family

The log-concave family encompasses a broad range of discrete distributions, such as Poisson, hyper-Poisson, Poisson binomial, binomial, negative binomial, and geometric, and hyper-geometric, with broad applications to statistics [Saumard and Wellner, 2014], computer science [Lovász and Vempala, 2007], economics [An, 1997], and geometry [Stanley, 1989].

Formally, a distribution $p \in \Delta_{\mathbb{Z}}$ is *log-concave* if $p$ has a contiguous support and $p_x^2 \geq p_{x-1} \cdot p_{x+1}$ for all $x \in \mathbb{Z}$. The next result bounds the profile entropy of this family, and is *tight* up to logarithmic factors. For simplicity, henceforth we write $a \wedge b$ for $\min\{a, b\}$ (and $\vee$ for max), and slightly abuse the notation and write $a \simeq b$ for $a+1 = \tilde{\Theta}(b+1)$, which does not change the nature of the results.

**Theorem 6.** *Let $\mathcal{L}_\sigma \subseteq \Delta_{\mathbb{Z}}$ denote the collection of log-concave distributions with variance $\sigma^2$. Then,*

$$\max_{p \in L_\sigma} \mathcal{H}_n(p) \simeq \sigma \wedge \frac{n}{\sigma}.$$

*In particular, if we discretize a Gaussian variable $X \sim \mathcal{N}(\mu, \sigma^2)$ by rounding it to the nearest integer, the distribution of the resulting variable achieves the maximum, up to logarithmic factors. Moreover, such a discretization procedure preserves log-concavity for any continuous distribution over $\mathbb{R}$.*

A similar bound holds for $t$-mixtures of log-concave distributions. More concretely,

**Theorem 14.** *For any $t$-mixture $p \in \Delta_{\mathbb{Z}}$ of log-concave distributions with variances $\sigma_i^2, 1 \leq i \leq t$,*

$$\mathcal{H}_n(p) \lesssim \left(\sum_i \sigma_i\right) \wedge \max_i \left\{\frac{n}{\sigma_i}\right\},$$

*where the right-hand side is assumed to be at least $t$ since otherwise $\mathcal{H}_n(p) \lesssim t$, and in practice, $t$ is often a small quantity, e.g. a constant.*

## D.2  Proof of Theorem 6 and 14

We start by showing the $\mathcal{H}_n(p) \gtrsim \sigma \wedge n/\sigma$ lower bound. A requirement is that $p$ must be a discrete log-concave distribution. We show that one can take $p$ as a discretized Gaussian $\mathcal{N}(\mu, \sigma^2)$. In addition, the discretization procedure works for any continuous distribution and preserves log-concavity and essentially also the variance. We will start by introducing the discretization procedure.

*Proof.* Log-concavity is a generic structure exhibited by numerous classical distributions, both those discrete (introduced above) and continuous ones, such as Gaussian, exponential, uniform, logistic, and Laplace distributions. Below, we present a discretization procedure that preserves distribution shapes such as monotonicity, modality, and log-concavity. Applying this procedure to a Gaussian distribution $\mathcal{N}(\mu, \sigma^2)$ yields the lower bound in Theorem 6.

Let $X$ be a continuous random variable with density function $f(x)$. For any $x \in \mathbb{R}$, denote by $\lceil x \rfloor$ the closest integer $z$ such that $x \in (z - 1/2, z + 1/2]$. The distribution of $\lceil X \rfloor$ is over $\mathbb{Z}$ and satisfies

$$p(z) := \int_{z-\frac{1}{2}}^{z+\frac{1}{2}} f(x)dx, \ \forall z \in \mathbb{Z}.$$

We refer to the random variable $\lceil X \rfloor$ as the discretized version of $X$.

**Shape Preservation**  By the definition of $\lceil x \rfloor$, one can readily verify that the above procedure preserves several important shape characteristics of distributions, such as monotonicity, modality, and $k$-modality (possibly yields a smaller $k$). The following theorem further covers log-concavity.

**Lemma 9.** *For any continuous random variable $X$ over $\mathbb{R}$ with a log-concave density $f$, the distribution $p \in \Delta_{\mathbb{Z}}$ associated with $\lceil X \rfloor$ is also log-concave.*

To show this, we need the following basic lemma about concave functions.

**Lemma 10.** *If $f$ is a real concave distribution, for any real numbers $x_1, x_2, y_1$, and $y_2$ satisfying $x_1 \leq x_2$, $y_1 \leq y_2$, $x_1 < y_1$, and $x_2 < y_2$,*

$$\frac{f(y_1) - f(x_1)}{y_1 - x_1} \geq \frac{f(y_2) - f(x_2)}{y_2 - x_2}.$$

By the above lemma, for any $x, y \in \mathbb{R}$ such that $|x - y| \leq 1$, and any function $f$ that is log-concave,

$$\log f(x + 1) - \log f(x) \leq \log f(y) - \log f(y - 1) \iff f(x + 1)f(y - 1) \leq f(x)f(y).$$

*Proof of Lemma 9.* By definition, distribution $p$ is log-concave if $p$ has a consecutive support and $p(z)^2 \geq p(z+1)p(z-1), \forall z$. The first condition holds for $\lceil X \rfloor$ since $X$ is has a continuous support on $\mathbb{R}$, and $p(z)$ is positive as long as $f(x) > 0$ for a non-empty sub-interval of $(z - 1/2, z + 1/2]$.

Below we show that $p$ also satisfies the second condition. Specifically, for any $z \in \mathbb{Z}$,

$$
\begin{aligned}
p(z - 1)p(z + 1) &= \left( \int_{z-\frac{3}{2}}^{z-\frac{1}{2}} f(x)dx \right) \left( \int_{z+\frac{1}{2}}^{z+\frac{3}{2}} f(x)dx \right) \\
&= \left( \int_{z-\frac{1}{2}}^{z+\frac{1}{2}} f(x - 1)dx \right) \left( \int_{z-\frac{1}{2}}^{z+\frac{1}{2}} f(x + 1)dx \right) \\
&= \int_{z-\frac{1}{2}}^{z+\frac{1}{2}} \int_{z-\frac{1}{2}}^{z+\frac{1}{2}} f(x - 1)f(y + 1)dxdy \\
&\leq \int_{z-\frac{1}{2}}^{z+\frac{1}{2}} \int_{z-\frac{1}{2}}^{z+\frac{1}{2}} f(x)f(y)dxdy \\
&= \left( \int_{z-\frac{1}{2}}^{z+\frac{1}{2}} f(x)dx \right)^2 \\
&= p(z)^2,
\end{aligned}
$$

where the inequality follows by Lemma 10 and its implication. □

**Moment preservation**    Denote by $p$ the distribution of $\lceil X \rfloor$ for $X \sim f$. Let $\mu$ and $\sigma^2$ be the mean and variance of density $f$, given that they exist. The theorem below shows that distribution $p$ has, within small additive absolute constants, a mean of $\mu$ and variance of $\Theta(\sigma^2)$.

**Lemma 11.** *Under the aforementioned conditions, the mean of $\lceil X \rfloor$ satisfies*

$$\mathbb{E}\lceil X \rfloor = \mu \pm \frac{1}{2},$$

*and the variance of $\lceil X \rfloor$ satisfies*

$$(\sigma - 1)^2 \leq \mathbb{E}(\lceil X \rfloor - \mathbb{E}\lceil X \rfloor)^2 \leq (\sigma + 1)^2.$$

*Proof of Lemma 11.* First consider the mean value of $\lceil X \rfloor$ for $X \sim f$. We have

$$\mathbb{E}\lceil X \rfloor = \mathbb{E}[\lceil X \rfloor - X] + \mathbb{E}[X] = \mu \pm \frac{1}{2}.$$

Consider the variance of $\lceil X \rfloor$ and apply inequality $(a+b)^2 \leq a^2(1+1/t) + b^2(1+t), \forall t > 0$.

$$
\begin{aligned}
\mathbb{E}(\lceil X \rfloor - \mathbb{E}\lceil X \rfloor)^2 &= \int_{-\infty}^{\infty} (\lceil x \rfloor - \mathbb{E}\lceil X \rfloor)^2 \cdot f(x)dx \\
&= \int_{-\infty}^{\infty} (\lceil x \rfloor - x + (x - \mathbb{E}X) + \mathbb{E}X - \mathbb{E}\lceil X \rfloor)^2 \cdot f(x)dx \\
&\leq \int_{-\infty}^{\infty} \left( (\lceil x \rfloor - x + \mathbb{E}X - \mathbb{E}\lceil X \rfloor)^2 \left(1 + \frac{1}{t}\right) + (x - \mathbb{E}X)^2 (1 + t) \right) f(x)dx \\
&\leq \int_{-\infty}^{\infty} \left( \left(1 + \frac{1}{t}\right) + (x - \mathbb{E}X)^2 (1 + t) \right) f(x)dx \\
&= 1 + \frac{1}{t} + t\sigma^2 + \sigma^2 \\
&= (\sigma + 1)^2.
\end{aligned}
$$

By a different inequality, $(a+b)^2 \geq a^2(1 - 1/t) + b^2(1 - t), \forall t > 0$, we also have

$$
\mathbb{E}(\lceil X \rfloor - \mathbb{E}\lceil X \rfloor)^2 \geq (\sigma - 1)^2. \qquad \square
$$

By the above lemma, for almost any $\sigma \geq 1$, we can construct a discrete log-concave distribution of variance $\sigma^2$ if there is a continuous one with roughly the same variance.

Next, letting $p_G$ denote the distribution of $\lceil X \rfloor$ for $X \sim \mathcal{N}(\mu, \sigma^2)$, we lower bound $\mathcal{H}_n^{\mathcal{S}}(p_G)$ (effectively, the profile entropy $\mathcal{H}_n(p_G)$). By definition, this discretized Gaussian, which we write as $\lceil \mathcal{N} \rfloor (\mu, \sigma^2)$, has a distribution in the form of

$$
p_G(z) := \frac{1}{\sqrt{2\pi}\sigma} \int_{z-\frac{1}{2}}^{z+\frac{1}{2}} \exp\left( -\frac{(x - \mu)^2}{2\sigma^2} \right) dx, \ \forall z \in \mathbb{Z}.
$$

Through the subsequent analysis, we show that

**Lemma 12.** *Under the aforementioned conditions,*

$$
H_n^{\mathcal{S}}(p_G) \geq \Omega\left( \frac{1}{\log n} \right) \left( \sigma \wedge \frac{n}{\sigma} \right).
$$

The lower bound in Theorem 6 follows by these inequalities.

*Proof.* At it is clear from the context, we write $p$ instead of $p_G$. Recall that

$$
H_n^{\mathcal{S}}(p) = \sum_{j \geq 1} \min\left\{ p_{I_j}, j \cdot \log n \right\},
$$

where $p_{I_j}$ denotes the number of probabilities belonging to $I_j = ((j-1)^2, j^2] \cdot (\log n)/n$. Computing the quantity for part of the distribution can only reduce the value of $H_n^{\mathcal{S}}(p)$. Hence, we will focus on symbols in the $(\mu + 1, \infty) \cap \mathbb{Z}$ range, over which the probability mass function $p(z)$ is monotone.

We will further assume that $n/\log n \gg \sigma \gg \log n$, since otherwise the right-hand side of the inequality reduces to $\mathcal{O}(1)$, and the result follows by $H_n^{\mathcal{S}}(p) \geq 1$ for all $n$ and $p$. In addition, we focus on $j \gg 1$ in the following argument, as the contribution to from $j = \mathcal{O}(1)$ is relatively small.

Given these assumptions, we have

$$
\begin{aligned}
p(z) \in I_j &\iff \frac{1}{\sqrt{2\pi}\sigma} \exp\left( -\frac{(z \pm 1/2 - \mu)^2}{2\sigma^2} \right) \in \left( (j-1)^2 \frac{\log n}{n}, j^2 \frac{\log n}{n} \right] \\
&\iff z \pm 1/2 - \mu \in \sqrt{2}\sigma \left[ \sqrt{c(\sigma, n) - 2\log j}, \sqrt{c(\sigma, n) - 2\log(j-1)} \right),
\end{aligned}
$$

where $c(\sigma, n) := \log\left( n/(\sqrt{2\pi}\sigma \log n) \right)$ and the interval is well-defined iff

$$
c(\sigma, n) \geq 2\log j \iff \frac{n}{\sqrt{2\pi}\sigma \log n} \geq j^2 \iff \sqrt{\frac{n}{\sqrt{2\pi}\sigma \log n}} \geq j \impliedby \sqrt{\frac{n}{\sigma \log n}} \geq 2j.
$$

For clarity, we divide our analysis into two cases: $\sqrt{n} \geq \sigma \gg \log n$ and $n/\log n \gg \sigma > \sqrt{n}$.

For the first case and $j \leq \sqrt{\sigma/\log n}/2 \leq \sqrt{n/(\sigma \log n)}/2$, the length $L_j$ of the above interval, which equals to $p_{I_j}$ up to an additive slack of 2, satisfies

$$
\begin{aligned}
\frac{L_j}{\sqrt{2}\sigma} &= \sqrt{c(\sigma,n) - 2\log(j-1)} - \sqrt{c(\sigma,n) - 2\log j} \\
&= \frac{2\log(j/(j-1))}{(c(\sigma,n) - 2\log(j-1)) + (c(\sigma,n) - 2\log j)} \\
&= \frac{\log(j/(j-1))}{\log\left(n/(\sqrt{2\pi}j(j-1)\sigma \log n)\right)} \\
&= \Omega\left(\frac{1}{\log n}\log\left(1 + \frac{1}{j-1}\right)\right) \\
&= \Omega\left(\frac{1}{j\log n}\right).
\end{aligned}
$$

Therefore, we have $L_j = \Omega(\sigma/(j\log n))$. Since $\sigma \gg \log n$ ensures $L_j \geq 3$ and $j \leq \sqrt{\sigma/\log n}/2$ is equivalent to $\sigma \geq 4j^2 \log n$, the lower bound on $L_j$ transforms into $p_{I_j} \geq \Omega(j)$. Hence in this case, $H_n^{\mathcal{S}}(p)$ admits the following bound

$$
H_n^{\mathcal{S}}(p) = \sum_{j \geq 1} \min\left\{p_{I_j}, j \cdot \log n\right\} \geq \sum_{j=\mathcal{O}(1)}^{\sqrt{\sigma/\log n}/2} \Omega(j) = \Omega\left(\frac{\sigma}{\log n}\right).
$$

In the $n/\log n \gg \sigma > \sqrt{n}$ case, we have $\sqrt{\sigma/\log n} > \sqrt{n/(\sigma \log n)}$. Repeating the previous reasoning for $j \leq \sqrt{n/(\sigma \log n)}/2$, we again obtain $L_j = \Omega\left(\sigma/(j\log n)\right)$ and $p_{I_j} \geq \Omega(j)$.

Therefore,

$$
H_n^{\mathcal{S}}(p) = \sum_{j \geq 1} \min\left\{p_{I_j}, j \cdot \log n\right\} \geq \sum_{j=\mathcal{O}(1)}^{\sqrt{n/(\sigma \log n)}/2} \Omega(j) = \Omega\left(\frac{n}{\sigma \log n}\right).
$$

Finally, note that in the first case, $\min\{\sigma, n/\sigma\} = \sigma$, and in the second, $\min\{\sigma, n/\sigma\} = n/\sigma$.

Consolidating these results yields the desired lower bound

$$
\mathcal{O}(\log n) \cdot H_n^{\mathcal{S}}(p) \geq \sigma \wedge \frac{n}{\sigma}. \qquad \square
$$

Next we proceed to the upper bound.

For any sample $X^n \sim p$, the profile dimension $\mathcal{D}(X^n)$ is at most the number of distinct symbols in the sample. It is well known that the tail probability of a log-concave distribution decays exponentially fast. Hence, the effective support size of $p$ with respect to $X^n$ is $\tilde{\mathcal{O}}(\sigma + 1)$, beyond which the tail probabilities can be as small as $1/n^3$ (the asymptotic notation hides logarithmic factors of $n$). Given this, even we sample from $p$ for $n$ times, the probability that we get only $\tilde{\mathcal{O}}(\sigma + 1)$ distinct symbols is at least $(1 - 1/n^3)^n \geq 1 - 1/n$. Therefore, we have $\mathcal{H}_n(p) \simeq \mathcal{D}(X^n) \lesssim \sigma + 1$.

Now, we extend this argument to a $t$-mixture of log-concave distributions with variances $\sigma_i^2, i \in [t]$. For a length-$n$ sample from this a distribution, the number of sample points from each mixture component is is at most $n$. Hence, with high probability, the number of distinct symbols in a length-$n$ sample is at most $\sum \sigma_i + t$, up to logarithmic factors of $n$.

For the other part of the upper bound, we can assume that $\sigma \geq \sqrt{n}$ (otherwise we need to consider only the above case) and $n$ is larger than some absolute constant. Then by a concentration inequality in Diakonikolas et al. [2016], the maximum probability $p_{\max}$ of $p$ belongs to $[1/(8\sigma), 1/\sigma]$. Hence, the last index $J$ for which $p_{I_J} \neq 0$ satisfies

$$
(J-1)^2 \frac{\log n}{n} \leq \frac{1}{\sigma} \iff J \leq \sqrt{\frac{n}{\sigma \log n}} + 1.
$$

Therefore, we have

$$\mathcal{H}_n^{\mathcal{S}}(p) = \sum_{j \geq 1} \min\left\{p_{I_j}, j \cdot \log n\right\} \leq \log n + \sum_{j=1}^{\sqrt{n/(\sigma \log n)}+1} j \cdot \log n \leq \mathcal{O}(\log n)\left(1 + \frac{n}{\sigma}\right).$$

Our upper bound is uniformly better than the $\min\{\sigma, (n^2/\sigma)^{1/3}\}$ bound in Hao and Orlitsky [2019b], which is derived for $\mathcal{D}_n \sim p$. More importantly, we actually provide a complete characterization of the profile entropy value that is optimal up to logarithmic factors.

Next, we extend the $n/\sigma$ bound to the mixture model. Write the mixture distribution as $p := \sum_i w_i \cdot p_i$, with $w_i$'s being the mixing weights and $p_i$'s being log-concave distributions with variances $\sigma_i^2$, respectively for $1 \leq i \leq t$. It is clear that $p_{\max}$ in this case is at most the maximum probability of some $p_i$, which at most $\max_i 1/\sigma_i$. The rest of the proof is the same as above. $\square$

### D.3 Theorem 7: Power-Law Family

**Power-law**   Power-law is a ubiquitous structure appearing in many situations of scientific interest, ranging from natural phenomena such as the initial mass function of stars [Kroupa, 2001], species and genera [Humphries et al., 2010], rainfall [Machado and Rossow, 1993], population dynamics [Taylor, 1961], and brain surface electric potential [Miller et al., 2009], to human-made circumstances such as the word frequencies in a text [Baayen, 2002], income rankings [Drăgulescu and Yakovenko, 2001], company sizes [Axtell, 2001], and internet topology [Faloutsos et al., 1999].

Formally, a discrete distribution $p \in \Delta_{\mathbb{Z}}$ is a *power-law with power* $\alpha \geq 0$ if $p$ has a support of $[k] := \{1, \ldots, k\}$ for some $k \in \mathbb{Z}^+ \cup \{\infty\}$ and $p_x \propto x^{-\alpha}$ for all $x \in [k]$. Note that if $\alpha \in [0, 1]$, the distribution is well-defined for only finite $k$. The next result fully characterizes the profile entropy of power-laws over the entire ranges of $\alpha, n$, and $k$.

**Theorem 7.** *Let $p \in \Delta_{[k]}$ be a power-law distribution with power $\alpha$. Then,*

$$\mathcal{H}_n(p) \simeq \begin{cases} k & \text{if } \alpha > \frac{k^{1+\alpha}}{n} \vee 1 \text{ or } 1 \geq \alpha > \frac{k^2}{n}, \\ n^{\frac{1}{\alpha+1}} & \text{if } \frac{k^{1+\alpha}}{n} \geq \alpha > 1, \\ \left(\frac{n}{k^{1-\alpha}}\right)^{\frac{1}{1+\alpha}} & \text{if } \frac{k^2}{n} \wedge 1 \geq \alpha > \frac{k^{1-\alpha}}{n}, \\ \frac{n}{k^{1-\alpha}} - \frac{n}{k} & \text{if } \frac{k^{1-\alpha}}{n} \wedge 1 \geq \alpha \text{ and } \alpha \geq 2\log_k\left(7\sqrt{\frac{k}{n}}+1\right), \\ k \wedge \sqrt{\frac{n}{k^{1-\alpha}}} & \text{if } \frac{k^{1-\alpha}}{n} \wedge 1 \geq \alpha \text{ and } 2\log_k\left(7\sqrt{\frac{k}{n}}+1\right) > \alpha. \end{cases}$$

*In particular, as $\alpha \to 0$, the bound degenerates to $k \wedge \sqrt{\frac{n}{k}}$, which is at most $n^{\frac{1}{3}}$.*

Since a power-law sample profile is completely specified by $\alpha$, $k$, and $n$, the above theorem directly applies to model parameter estimation. Specifically, we first compute $\mathcal{D}_n \sim p$, which is a simple function of the symbol counts. By Theorem 1, we can then use it to approximate $\mathcal{H}_n(p)$. Finally, we utilize the characterization theorem and find the parameter relations (testing might be necessary).

The theorem fully characterizes the profile entropy of power-laws and is significantly better than the basic $\{k, \sqrt{n \log n}\}$ bound for both $k \gg \sqrt{n}$ and $k \ll \sqrt{n}$. We can see how different parameter interplay with each other and leverage these relations in applications such as parameter estimation. In comparison, a result in Hao and Orlitsky [2019b], when combined with our entropy-dimension equivalence theorem, yields only an $n^{1/(1+\alpha)}$ upper bound (and no lower bounds nor the right dependence on $k$), which is clearly suboptimal and provides no improvement over $\sqrt{n \log n}$ for $\alpha < 1$.

### D.4 Proof of Theorem 7

*Proof.* For the ease of exposition, write the probability of symbol $i$ assigned by distribution $p$ as $p_i := c_\alpha^{-1} \cdot i^{-\alpha}$, where $c_\alpha$ is a normalizing constant that implicitly depends on $k$. Note that

$$\frac{k^{1-\alpha}}{1-\alpha} + \frac{\alpha}{1-\alpha} \geq 1 + \int_1^k x^{-\alpha} dx \geq c_\alpha = \sum_{i=1}^k i^{-\alpha} \geq \int_1^{k+1} x^{-\alpha} dx = \frac{(k+1)^{1-\alpha}}{1-\alpha} - \frac{1}{1-\alpha}.$$

Basic calculus shows that, up to logarithmic factors, we can approximate the normalizing constant as

$$c_\alpha = \sum_{i=1}^{k} \frac{1}{i^\alpha} \simeq k^{1-\alpha} \vee 1,$$

Recall that the quantity of interest is essentially

$$H_n^{\mathcal{S}}(p) = \sum_{j \geq 1} \min \left\{ p_{I_j}, j \cdot \log n \right\}.$$

It will be convenient to denote $c := c(\alpha, k, n) := (c_\alpha \log n)/n \simeq (k^{1-\alpha} \vee 1)/n$. First, consider $p_{I_j}$ for a sufficiently large $j$ (i.e., $j \gg 1$) and note that

$$p_i \in I_j \iff \frac{1}{c_\alpha i^\alpha} \in \left( (j-1)^2 \frac{\log n}{n}, j^2 \frac{\log n}{n} \right]$$

$$\iff i \in I_j' := \left[ \left( j^2 c \right)^{-\frac{1}{\alpha}}, \left( (j-1)^2 c \right)^{-\frac{1}{\alpha}} \right).$$

Observe that the length $L_j$ of interval $I_j'$, which differs from the value of $p_{I_j}$ by at most 2, is proportional to $(j-1)^{-2/\alpha} - j^{-2/\alpha}$, and hence is a decreasing function of $j$. Furthermore, each term $\min\{p_{I_j}, j \cdot \log n\} \approx \min\{L_j, j \cdot \log n\}$ is basically the minimum between this decreasing function and $j \log n$, an increasing function of $j$. This naturally calls for determining the value of $j$ at which the two functions are equal. Concretely,

$$\left( (j-1)^2 c \right)^{-\frac{1}{\alpha}} - \left( j^2 c \right)^{-\frac{1}{\alpha}} = j \log n \implies j \simeq J := \left( \frac{1}{\alpha^\alpha c} \right)^{\frac{1}{2+2\alpha}},$$

where $J$ implicitly depends on $\alpha$ and $n$. In addition, since probability $p_i$ vanishes if $i \notin [1, k]$, we need to consider only $\sqrt{1/(ck^\alpha)} + 1 \leq j \leq \sqrt{1/c}$.

We can decompose the summation $H_n^{\mathcal{S}}(p)$ into two parts. The first part consists of indices $j \leq J$,

$$H_{n,1}^{\mathcal{S}}(p) := \sum_{j=\sqrt{1/(ck^\alpha)}+1}^{J \wedge \sqrt{1/c}} \min \left\{ p_{I_j}, j \cdot \log n \right\} \simeq \sum_{j=\sqrt{1/(ck^\alpha)}+1}^{J \wedge \sqrt{1/c}} j.$$

Correspondingly, the second part consists of indices $j \geq J$. For these indices $j$, we have $L_j \leq j \cdot \log n$. Recall that $I_j'$ specifies the range of $i$ satisfying $p_i \in I_j$. Then the second part satisfies

$$H_{n,2}^{\mathcal{S}}(p) := \sum_{j=J \vee (\sqrt{1/(ck^\alpha)}+1)}^{\sqrt{1/c}} \min \left\{ p_{I_j}, j \cdot \log n \right\} \simeq \sum_{j=J \vee (\sqrt{1/(ck^\alpha)}+1)}^{\sqrt{1/c}} L_j,$$

where the inequality follows by the fact that the intervals $I_j'$ are consecutive. In addition, note that the left end point of $I_j'$ equals $(J^2 c)^{-\frac{1}{\alpha}} = (\alpha/c)^{\frac{1}{1+\alpha}}$.

The rest of the proof follows by dividing the analysis into several cases according to whether $\alpha > 1$ and the relative magnitude of $J$, $\sqrt{1/c}$, and $(\sqrt{1/(ck^\alpha)} + 1)$.

For a concrete example, if $\alpha > 1$, then our approximation of $c_\alpha$ becomes $c_\alpha \simeq 1$, hence $c \simeq 1/n$, and it is also clear that $J = 1/(\alpha^\alpha c)^{\frac{1}{2\alpha+2}} \leq \sqrt{1/c}$. Therefore,

$$H_{n,1}^{\mathcal{S}}(p) \simeq \sum_{j=\sqrt{1/(ck^\alpha)}+1}^{J} j.$$

Now, consider the relation between $J$ and $\sqrt{1/(ck^\alpha)}$. By the continuity of profile entropy, we can treat $c$ as $1/n$. If $\alpha \geq k^{1+\alpha}/n$, then $J \leq \sqrt{1/(ck^\alpha)}$ and our upper bound for $H_{n,1}^{\mathcal{S}}(p)$ vanishes. The quantity of interest hence becomes $H_{n,1}^{\mathcal{S}}(p)$, which equals to

$$H_n^{\mathcal{S}}(p) = H_{n,2}^{\mathcal{S}}(p) \simeq \sum_{j=\sqrt{1/(ck^\alpha)}+1}^{\sqrt{1/c}} L_j = k.$$

On the other hand, if $\alpha < k^{1+\alpha}/n$, then $J \geq \sqrt{1/(ck^\alpha)} + 1$ and $H_{n,1}^{\mathcal{S}}(p)$ satisfies

$$H_{n,1}^{\mathcal{S}}(p) \simeq \sum_{j=\sqrt{1/(ck^\alpha)}+1}^{J} j \leq J^2 \simeq \left(\frac{n}{\alpha^\alpha}\right)^{\frac{1}{\alpha+1}}.$$

Our approximation of $H_{n,2}^{\mathcal{S}}(p)$ reduces to

$$H_{n,2}^{\mathcal{S}}(p) \simeq \sum_{j=J}^{\sqrt{1/c}} L_j \approx (J^2 c)^{-\frac{1}{\alpha}} = \left(\frac{\alpha}{c}\right)^{\frac{1}{\alpha}} \simeq (\alpha n)^{\frac{1}{\alpha+1}} \simeq n^{\frac{1}{\alpha+1}}.$$

Consolidating these bounds and noting $\alpha^{\frac{1}{\alpha+1}} \in (1, 2)$ yield that $H_n^{\mathcal{S}}(p) \simeq n^{\frac{1}{\alpha+1}}$. The expressions for $\alpha < 1$ can be derived in the similar manner. $\qquad\square$

### D.5 Theorem 8: Histogram Family

**Histogram**  While histogram is among the most widely studied representations, histogram distributions' importance also rises with the rapid growth of data sizes in modern scientific applications. For example, *subsampling*, a generic strategy to handle large datasets, naturally induces a histogram distribution over different categories of the data. This induced distribution often summarizes vital data statistics, leveraging which yields efficient and flexible inference procedures.

Formally, a discrete distribution $p \in \Delta_{\mathbb{Z}}$ is a *t-histogram* if we can partition its support into at most $t$ pieces such that $p$ takes the same probability value over each piece. The theorem below provides near-optimal bounds on the profile entropy of the $t$-histogram distributions.

**Theorem 8.** *Denote by $\mathcal{I}_t \subseteq \Delta_{\mathbb{Z}}$ the collection of $t$-histogram distributions. Then,*

$$\max_{p \in \mathcal{I}_t} \mathcal{H}_n(p) \simeq (nt^2)^{\frac{1}{3}} \wedge \sqrt{n}.$$

In practical settings, the value of $t$ is often poly-logarithmic in $n$, and the bound reduces to $\tilde{\mathcal{O}}(n^{1/3})$. For the particular case of $t = 1$, distribution $p$ is uniform over some unknown contiguous support. This result overlaps with Theorem 7 with $\alpha = 0$, yielding the following bound.

**Corollary 5.** *For any uniform distribution $p$ with support size $k$, we have $\mathcal{H}_n(p) \simeq k \wedge \sqrt{\frac{n}{k}}$.*

Next we consider mixtures of histogram distributions.

**Theorem 9.** *Let $T$ be the positive integer sequence $\{t_i\}_{i=1}^{s}$. Denote by $S_T$ the sum $\sum_i t_i$, and by $\mathcal{I}_T$ the $s$-mixture of $t$-histograms with parameters specified by $T$. Then,*

$$\max_{p \in \mathcal{I}_T} \mathcal{H}_n(p) \simeq (nS_T^2)^{\frac{1}{3}} \wedge \sqrt{n}.$$

*Proof.* The proof follows by Theorem 8, which holds for any $t$, and the fact that $\mathcal{I}_T$ coincides with the collection of all $S_T$-histogram distributions. $\qquad\square$

### D.6 Proof of Theorem 8

*Proof.* First we establish the lower bound. Recall that the quantity of interest is essentially

$$H_n^{\mathcal{S}}(p) = \sum_{j \geq 1} \min\left\{p_{I_j}, j \cdot \log n\right\}.$$

Our construction depends on the value of $t$ as follows. Let $A \cdot \{B\}$ denote the length-$A$ constant sequence with value $B$. If $t = 1$, distribution $p$ has the following form

$$p := \tilde{\Theta}(n^{1/3}) \cdot \{p_0 \in I_{n^{1/3}}\},$$

where $p_0$ is a properly chosen probability in $I_{n^{1/3}}$ so that $p$ is well-defined, and the range of support of distribution $p$ is irrelevant for our purpose and hence unspecified. If $2 \leq t < n^{1/4}/(2\sqrt{\log n})$, then for some parameter $s \geq 0$ to be determined, the distribution $p$ has the following form

$$p := L \cdot \left\{\frac{1}{n^2}\right\} \cup \left(\bigcup_{j=s+1}^{s+t-1}\left((j \log n) \cdot \left\{j^2 \frac{\log n}{n}\right\}\right)\right),$$

where the probability values are sorted according to the ordering they appear above, and $L$ is a properly chosen to make the probabilities sum to 1. For the distribution to be well-defined, we require

$$\sum_{j=s+1}^{s+t-1} (j \log n) \cdot \left(j^2 \frac{\log n}{n}\right) \leq 1 \impliedby t(s+t)^3 \leq \frac{n}{\log^2 n} \impliedby s \leq \left(\frac{n}{t \log^2 n}\right)^{1/3} - t.$$

Note that the last inequality is valid if $t < n^{1/4}/(2\sqrt{\log n})$. Let $s$ be the maximum integer satisfying the above inequality. Then, $H_n^{\mathcal{S}}(p)$ admits the lower bound

$$H_n^{\mathcal{S}}(p) \geq \sum_{j=s+1}^{s+t-1} (j \log n) \geq \frac{(2s+t)(t-1)}{2} \log n \geq \frac{1}{4}\left(\frac{n}{t \log^2 n}\right)^{1/3} t \log n = \Omega((nt^2 \log n)^{1/3}).$$

Finally, if $t \geq n_0 := n^{1/4}/(2\sqrt{\log n})$, distribution $p$ has the following form

$$p := (t - n_0 + 1) \cdot \{p_0\} \bigcup \left(\bigcup_{j=1}^{n_0-1} \left((j \log n) \cdot \left\{j^2 \frac{\log n}{n}\right\}\right)\right),$$

where $p_0$ is a properly chosen to make the probabilities sum to 1. According to the previous reasoning, distribution $p$ is well-defined and quantity $H_n^{\mathcal{S}}(p)$ satisfies

$$H_n^{\mathcal{S}}(p) \geq \sum_{j=1}^{n_0-1} (j \log n) \geq \frac{n_0(n_0-1)}{2} \log n \geq \Omega(\sqrt{n}).$$

Consolidating these results yields the desired lower bound.

Regarding the upper bound, the work of Hao and Orlitsky [2019b] studies the profile dimension for distributions $p \in \mathcal{I}_t$ and shows that

$$\mathbb{E}[\mathcal{D}_n] \lesssim (nt^2)^{\frac{1}{3}} \wedge \sqrt{n}.$$

Consolidating this inequality with Theorem 1 (dimension-entropy equivalence) and Corollary 4 (dimension concentration) yields the desired upper bound. $\qquad\square$

# E   Extensions

## E.1   Multi-Dimensional Profiles

As we elaborate below, the notion of profile generalizes to the multi-sequence setting.

Let $\mathcal{X}$ be a finite or countably infinite alphabet. For every vector $\vec{n} := (n_1, \ldots, n_d) \in \mathbb{N}^d$ and tuple $x^{\vec{n}} := (x_1^{n_1}, \ldots, x_d^{n_d})$ of sequences in $\mathcal{X}^*$, the *multiplicity* $\mu_y(x^{\vec{n}})$ of a symbol $y \in \mathcal{X}$ is the vector of its frequencies in the tuple of sequences. The *profile* of $x^{\vec{n}}$ is the multiset $\varphi(x^{\vec{n}})$ of multiplicities of the observed symbols [Acharya et al., 2010, Das, 2012, Charikar et al., 2019b], and its *dimension* is the number $\mathcal{D}(x^{\vec{n}})$ of distinct elements in the multiset. Drawing independent samples from each distribution in $\vec{p} := (p_1, \ldots, p_d) \in \Delta^d$, the *profile entropy* is the entropy of the joint-sample profile.

Many of the previous results potentially generalize to this multi-dimensional setting. For example, the $\sqrt{2n}$ bound on $\mathcal{D}(x^{\vec{n}})$ in the 1-dimensional case becomes

**Theorem 20.** *For any $\mathcal{X}$, $\vec{n}$, and $x^{\vec{n}} \in \mathcal{X}^{\vec{n}}$, there exists $r > 0$ such that*

$$\sum_i n_i \geq \frac{(r+1)(r+2)}{d+1}\binom{d+r+1}{d-1} \quad and \quad \binom{d+r}{d} - 1 \geq \mathcal{D}(x^{\vec{n}}).$$

Note that this recovers the $\sqrt{2n}$ bound for $d = 1$.

*Proof.* For simplicity, we suppress $x^{\vec{n}}$ in $\mathcal{D}(x^{\vec{n}})$. Let $\Delta_d$ denote the standard $d$-dimensional simplex. As each multiplicity corresponds to a vector in $\mathbb{N}^d$, in the ideal case, the profile that has the maximum

dimension $\mathcal{D}$ corresponds to the integer points in the scaled simplex $(r \cdot \Delta_d)$, for some properly chosen parameter $r > 0$. For a valid choice of $r$, we have

$$\sum_i n_i \geq \sum_{t=0}^{r+1} \binom{t+d-1}{d-1} \cdot t = \frac{(r+1)(r+2)}{d+1} \binom{d+r+1}{d-1}$$

and

$$\mathcal{D} \leq \sum_{t=1}^{r} \binom{t+d-1}{t} = \binom{d+r}{d} - 1.$$

Consolidating these two inequalities yields the desired result. $\qquad\square$

## E.2  Discrete Multivariate Gaussian Mixtures

Let $\Sigma$ be a $d \times d$ symmetric matrix with eigenvalues $\sigma_d^2 \geq \ldots \geq \sigma_d^2 \geq 1$ and $\mu$ be a $d$-dimensional integer vector. The *discrete $d$-dimensional Gaussian* induced by $(\mu, \Sigma)$ is specified by its *probability mass function*

$$p(x) := \frac{1}{C} \exp\left(-\frac{1}{2}(x-\mu)^T \Sigma^{-1}(x-\mu)\right), \forall x \in \mathbb{Z}^d.$$

where $C_\Sigma := C(n, d, \Sigma) > 0$ is a normalizing constant. In this section, we show that for $d \geq 9$,

$$\mathcal{H}_n(p) \lesssim \frac{n}{C} \wedge C\left(\gamma_d \exp\left(6d\frac{\sigma_d^2}{\sigma_1^2}\right)\left(\frac{2\log n}{d}\right)^{d/2}\right),$$

where $\gamma_d$ is a constant that appears in Lemma 14 and depends only on $d$. The bound resembles that in Theorem 6 for log-concave distributions. For $d = 1$ with $\Sigma = \sigma^2$, the normalizing factor is $C_\Sigma = \sqrt{2\pi}\sigma$, and the right-hand side reduces to $\tilde{\mathcal{O}}(\sigma \wedge n/\sigma)$ in Theorem 6.

Let us denote the multiplicative factor in the parentheses by $F_\Sigma := F(n, d, \Sigma)$. Just like Theorem 6 generalizes to 14, the above result generalizes to also mixtures of discrete $d$-dimensional Gaussians.

**Theorem 21.** *For a $t$-mixture $p \in \Delta_{\mathbb{Z}^d}$ of discrete $d$-dimensional Gaussians with covariance matrices $\Sigma_i$, where $1 \leq i \leq t$, its profile entropy satisfies*

$$\mathcal{H}_n(p) \lesssim \left(\sum_i C_i F_{\Sigma_i}\right) \wedge \max_i \left\{\frac{n}{C_i}\right\},$$

*where the right-hand side is assumed to be at least $t$ since otherwise $\mathcal{H}_n(p) \lesssim t$, and in practice, $t$ is often a small quantity, e.g. a constant.*

*Proof.* Below we establish Theorem 21 for $t = 1$. The proof of the general case follows by the subsequent reasoning and the arguments in Appendix D.2.

**Lower bound on $C$**  First, we bound $C_\Sigma$ from below in terms of the eigenvalues and other parameters. By symmetry, we can decompose the covariance matrix $\Sigma$ as

$$\Sigma = V\Lambda V^T,$$

where $\Lambda$ is a diagonal matrix with $\Lambda_{ii} = \sigma_i^2$, and $V$ is an orthonormal matrix whose $i$-th column is the eigenvector $v_i$ associated with $\sigma_i^2$.

Next, partition the real space $\mathbb{R}^d$ into unit cubes whose vertices belong to $\mathbb{Z}^d$. For any two vectors $\tilde{a}, \tilde{b} \in \mathbb{R}^d$ that belong to the same unit cube, we will bound the ratio between $p(\tilde{a})$ and $p(\tilde{b})$. Denote $a := \tilde{a} - \mu$ and $b := \tilde{b} - \mu$, and express $a$ and $b$ as linear combinations of eigenvectors,

$$a := \sum_{i=1}^{d} x_i \cdot v_i \text{ and } b := \sum_{i=1}^{d} y_i \cdot v_i.$$

The log-ratio between the induced probabilities satisfies

$$-2\log\frac{p(\tilde{a})}{p(\tilde{b})} = a^T\Sigma^{-1}a - b^T\Sigma^{-1}b$$

$$= (a+b)^T\Sigma^{-1}(a-b)$$

$$= \left(\sum_i (x_i+y_i)\cdot v_i^T\right)V\Lambda^{-1}V^T\left(\sum_i(x_i-y_i)\cdot v_i\right)$$

$$= \left(\sum_i (x_i+y_i)\cdot e_i^T\right)\Lambda^{-1}\left(\sum_i(x_i-y_i)\cdot e_i\right)$$

$$= \sum_i \sigma_i^{-2}(x_i^2 - y_i^2).$$

Since by construction, $\tilde{a} - \tilde{b} = a - b$ and $\tilde{a}, \tilde{b}$ belong to the same unit cube, hence $\sum_i(x_i - y_i)^2 = \|a-b\|_2^2 = \sum_i(\tilde{a}_i - \tilde{b}_i)^2 \le d$. Consequently, we bound the absolute value of the ratio by

$$2\left|\log\frac{p(\tilde{a})}{p(\tilde{b})}\right| = \left|\sum_i \sigma_i^{-2}(x_i^2 - y_i^2)\right|$$

$$\le \sum_i \sigma_i^{-2}\left|x_i^2 - (x_i - (x_i - y_i))^2\right|$$

$$\le 2\sum_i \sigma_i^{-2}\left(x_i^2 + (x_i - y_i)^2\right)$$

$$\le 2\sigma_1^{-2}\left(\sum_i x_i^2 + d\right)$$

$$= 2\sigma_1^{-2}\left(\|\tilde{a} - \mu\|_2^2 + d\right).$$

Now, consider the hyper-ellipse $E$ associated with

$$(x-\mu)^T\Sigma^{-1}(x-\mu) \le d.$$

For any $x \in E$, simple algebra shows that $\|x - \mu\|_2^2 \le d\sigma_d^2$. Hence by the previous discussion, for any unit cube $U$ with vertices in $\mathbb{Z}^d$, there exists a vertex $v_U$ (of $U$) such that for any $x \in U \cap E$,

$$\left|\log\frac{p(x)}{p(v_U)}\right| \le \sigma_1^{-2}\left(\|x-\mu\|_2^2 + d\right) \le \sigma_1^{-2}\left(d\sigma_d^2 + d\right) \le 2d\left(\frac{\sigma_d}{\sigma_1}\right)^2.$$

Note that $x \in E$ is equivalent to $p(x) \ge \exp(-d/2)/C$. Then, the probability mass over $E$ is at least

$$\int_{x\in E} p(x)dx \ge \int_{x\in E}\frac{\exp(-d/2)}{C} = \frac{\exp(-d/2)}{C}\cdot\text{Vol}(E) = \frac{\exp(-d/2)}{C}\cdot\frac{(\pi d)^{d/2}}{\Gamma(d/2+1)}\prod_{i=1}^d \sigma_i.$$

On the other hand, this probability mass is at most

$$\int_{x\in E} p(x)dx = \sum_U \int_x p(x)\cdot\mathbb{1}_{x\in E\cap U}dx \le \sum_U p(v_U)\cdot\exp\left(2d\left(\frac{\sigma_d}{\sigma_1}\right)^2\right) \le \exp\left(3d\left(\frac{\sigma_d}{\sigma_1}\right)^2\right).$$

Consolidating the lower and upper bounds and multiplying both sides by $C$ yield

$$C \geq \exp\left(-3d\left(\frac{\sigma_d}{\sigma_1}\right)^2\right)\exp\left(-\frac{d}{2}\right) \cdot \frac{(\pi d)^{d/2}}{\Gamma(d/2+1)}\prod_{i=1}^{d}\sigma_i$$

$$\implies C \geq \exp\left(-3d\left(\frac{\sigma_d}{\sigma_1}\right)^2\right) \cdot \frac{(\pi d/e)^{d/2}}{\sqrt{e\pi(d/2)}(d/(2e))^{d/2}}\prod_{i=1}^{d}\sigma_i$$

$$\implies C \geq \exp\left(-3d\left(\frac{\sigma_d}{\sigma_1}\right)^2\right) \cdot \frac{(2\pi)^{d/2}}{\sqrt{e\pi(d/2)}}\prod_{i=1}^{d}\sigma_i$$

$$\implies C \geq \exp\left(-3d\left(\frac{\sigma_d}{\sigma_1}\right)^2\right)\prod_{i=1}^{d}\sigma_i.$$

where the first step follows by the lemma below.

**Lemma 13.** *For any integer or semi-integer $x \geq 1/2$,*

$$\sqrt{2\pi x}\left(\frac{x}{e}\right)^x \leq \Gamma(x+1) \leq \sqrt{e\pi x}\left(\frac{x}{e}\right)^x.$$

**Upper bound**  We proceed to bound $\mathcal{H}_n^{\mathcal{S}}(p) = \sum_{j\geq 1}\min\{p_{I_j}, j \cdot \log n\}$.

Below we assume that $C < n/\log n$, since otherwise $p(x) \leq (\log n)/n, \forall x$, yielding an $\mathcal{O}(\log n)$ upper bound on $\mathcal{H}_n^{\mathcal{S}}(p)$. Then, by definition, the last index $j$ for which $p_{I_j} > 0$ satisfies

$$(j-1)^2\frac{\log n}{n} \leq \frac{1}{C} \implies j \leq 1 + \sqrt{\frac{1}{C}\frac{n}{\log n}} \leq 2\sqrt{\frac{1}{C}\frac{n}{\log n}}.$$

Denote by $J$ the quantity on the right-hand side. Then,

$$\sum_{j\geq 1}\min\{p_{I_j}, j \cdot \log n\} \leq \sum_{j=1}^{J}j\log n \leq J^2\log n \leq \frac{4n}{C}.$$

Furthermore, by a reasoning similar to the above, the collection of points $x \in \mathbb{Z}^d$ satisfying $p(x) \leq 1/(Cn) = p(\mu)/n \leq 1/n$ contributes at most $\mathcal{O}(\log n)$ to $\mathcal{H}_n^{\mathcal{S}}(p)$. Hence we need to analyze only points $x$ satisfying $p(x) > 1/(Cn)$. Equivalently, those in

$$E^\star := \left\{x \in \mathbb{Z}^d : (x-\mu)^T \Sigma^{-1}(x-\mu) \leq 2\log n\right\}.$$

Clearly, these points contribute at most $|E^\star|$ to the sum. Noting that $E^\star$ is a discrete hyper-ellipse, we can bound its cardinality by the following lemma in Bentkus and Götze [1997].

**Lemma 14.** *Let $\mu \in \mathbb{R}^d$ be a mean vector, and $\Sigma \in \mathbb{R}^{d\times d}$ be a real covariance matrix with nonzero eigenvalues $\sigma_1^2 \leq \ldots \sigma_d^2$. For any $d \geq 9$ and $t \geq \sigma_d^2$, the discrete ellipsoid*

$$E(t) := \left\{x \in \mathbb{Z}^d : (x-\mu)^T \Sigma^{-1}(x-\mu) \leq t\right\}$$

*admits the following inequality on its cardinality,*

$$|E(t)| \leq \left(1 + \frac{\gamma_d}{t}\frac{1}{\sigma_d^2}\left(\frac{\sigma_d}{\sigma_1}\right)^{2d+4}\right)\frac{(\pi t)^{d/2}}{\Gamma(d/2+1)}\prod_{i=1}^{d}\sigma_i,$$

*where $\gamma_d > 1$ is a constant that depends only on $d$.*

Applying the above lemma to bound $|E^\star|$ (where $t = 2\log n$) and combining the result with our lower bound on $C$ yield

$$
\begin{aligned}
|E(2\log n)| &\leq \left(1 + \frac{\gamma_d}{2\log n}\frac{1}{\sigma_d^2}\left(\frac{\sigma_d}{\sigma_1}\right)^{2d+4}\right)\frac{(2\pi\log n)^{d/2}}{\Gamma(d/2+1)}\exp\left(3d\left(\frac{\sigma_d}{\sigma_1}\right)^2\right)C \\
&\leq \left(1 + \frac{\gamma_d}{2\log n}\frac{1}{\sigma_d^2}\left(\frac{\sigma_d}{\sigma_1}\right)^{2d+4}\right)\frac{1}{\sqrt{\pi d}}\left(4e\pi\frac{\log n}{d}\right)^{d/2}e^{3d(\sigma_d/\sigma_1)^2}C \\
&\leq \left(1 + \frac{\gamma_d}{2\log n}\left(\frac{\sigma_d}{\sigma_1}\right)^{3d}\right)\left(\frac{2\log n}{d}\right)^{d/2}e^{5d(\sigma_d/\sigma_1)^2}C \\
&\leq \gamma_d\left(\frac{\sigma_d}{\sigma_1}\right)^{3d}\left(\frac{2\log n}{d}\right)^{d/2}e^{5d(\sigma_d/\sigma_1)^2}C \\
&\leq \gamma_d\left(\frac{2\log n}{d}\right)^{d/2}e^{6d(\sigma_d/\sigma_1)^2}C,
\end{aligned}
$$

where the second step follows by Lemma 13.

To summarize, we have established the desired bound

$$
\mathcal{H}_n^{\mathcal{S}}(p) \leq \mathcal{O}(\log n)\left(1 + \min\left\{\frac{n}{C}, \gamma_d(\alpha_\Sigma \cdot \beta_{d,n})^d \cdot C\right\}\right). \qquad \square
$$