[Reviews · NeurIPS 2020]

Review 1

Summary and Contributions: This paper introduces "profile entropy" for discrete distributions, and it shows its applications in the tasks of estimation, inference, and compression. More precisely, the speed of estimating a distribution relative to the best natural estimator is determined, the rate of inferring all symmetric properties relative to the best estimator over any label-invariant distribution collection is calculated, and compression of sample profiles are studied for which profile entropy plays the role of fundamental limit.

Strengths: - The paper has some merit by having contributions in studying the applications of profile entropy in the tasks of estimation, inference and compression. The topic of the paper is of interest to the NeurIPS community. - The results of the paper seem correct to me, except for some confusions which I point out below.

Weaknesses: - I found the paper fairly hard to read and follow. Especially, at times it is not clear whether a stated result is a known result or is a contribution of this paper. The paper does not seem to have much coherence and is a collection of relatively different results with no particular storyline. - In line 225, where exactly is the block compression algorithm proposed? Similarly in line 22, it should be pointed out where exactly the adaptive learning algorithm is. - Merely due to the fact that the profile of a sequence is the type of its type, it seems premature to claim that the paper connects profile entropy with the method of types in lines 333-338.

Correctness: The results seem correct to me, except that the statement of Theorem 1 and its proof in the appendix can be made more precise. For any given fixed n, I can compute the right hand side of the inequality, but then what does (D_n ~ H_n) mean for fixed n?

Clarity: The paper is fairly hard to read and sometimes confusing for realizing which of the results are contributions of this paper and which are known results. At times it seems that the paper is written with the style of a survey paper. Many \paragraph are used, some of which are just literature review. For example, the first paragraphs in section 2.4 which are applications of sample profiles in prior works, can be compressed and moved to the introduction.

Relation to Prior Work: The paper has compared its contributions with known results all in the "Main results" Section. The paper will benefit by putting a separate related result section.

Reproducibility: Yes

Additional Feedback: - I would like the author(s) to explain how they are going to improve the clarity of the paper and their own contributions in their response. - It will be good if an example can be provided on sample profile and entropy in Section 1 after their definitions. Also, why is small phi used in line 34 but big phi^n in line 39? - Line 143: has been --> has ----- Post Author Response ----- Since the author(s) have explained how they are going to improve the clarity of their paper in their response, I slightly increase my score.


Review 2

Summary and Contributions: This paper characterizes the profile entropy, which is the entropy of a set that measures the frequencies of symbols. This measure act as a fundamental bound that encompasses several statistical problems such as estimation, inference, and compression. In concrete, the profile entropy measures the discrepancy between the estimated distribution of a sample concerning the natural assignment. On the other hand, estimators of symmetric properties based on the profile entropy universally achieve minimax error. Finally, the profile entropy is a measure of complexity for block compression problems.

Strengths: The work is technically robust, and the contribution is clear in the sense that encompasses their results in three different fields. In particular, I like very much the following ingredients : + The profile entropy seems a novelty contribution that can be very useful in the information theory field and serves as a fundamental bound in a variety of statistical problems; + The proofs of the results are correct and consistent, and connections with universal coding are appealing and interesting.

Weaknesses: Even though the profile entropy seems an interesting tool that connects estimation, inference, and compression, there is no insights about its benefits and implications for practical problems. For instance, + How complex is the profile entropy to compute it? Specially, when dealing with high-dimensional alphabets. + Are there some examples where you can see the PML estimator? + Can we see how asymptotically close the profile is with its dimension? + Besides, the connection of the proofs with the method of types could be developed with more depth. The method of types concentrates the probability in a set, and we use it as a way of characterizing the behaviour of long sequences based on the properties of the type of a given sequence. In that sense, what is the gain of using the profile entropy instead of using the empirical distribution?

Correctness: Yes, the claims are robust even though it lacks empirical analysis. The contributions are well ordered and presented. In each area, the authors connected their work with state-of-the-art results from different references.

Clarity: The paper is well written and the results are well stated.

Relation to Prior Work: Most of the results are clearly connected with previous works and contributions. However, the authors could compare their results more naturally with some of the classical information-theoretic contributions in this field (e.g., universal compression of sequences).

Reproducibility: Yes

Additional Feedback: I suggest the authors to better address connections with universal source coding and related problems already mentioned in comments about "the weaknesses". In general, when comparing with other state-of-the-art, it is important in my opinion to show how (technically) problems are connected. I believe this can add much more value to the present work.


Review 3

Summary and Contributions: The paper proposes profile entropy H_phi(p), a function of any discrete distribution p, as a fundamental measure of how effective one can infer its properties from its samples. The main contributions are the following: 1. For any distribution p, its profile entropy characterizes how well one can estimate the distribution in KL divergence compared to any natural estimator. 2. For any discrete distribution and any symmetric property of a distribution, the paper shows that the plug-in estimator with profile maximum likelihood estimation for the distribution achieves the performance of the best estimator with sample size n/H_phi(p). 3. H_phi(p) characterizes how well one can compress the profile of a distribution losslessly. The paper also proposes algorithms to achieve the compression rate.

Strengths: 1. For instance-based distribution estimation, [Hao and Orlitsky 2019b] shows that there exists an algorithm that achieves a competitive risk depending on the expectation of the profile dimension (which is equal to profile entropy up to log factors). This work complements the result by proving a matching lower bound over all distribution with bounded profile entropy. Taking the maximum over the profile entropy for any distribution p, the bound recovers previous bound in the min-max setting. For several natural distribution classes, the bound strictly improves over min-max bounds over all distributions. 2. There have been several attempts on trying to construct a universal estimator that performs well for any property estimation tasks. However, they also impose some constriaints on the format of the properties (e.g. Lipshictz, additive and etc.). The result in the paper shows that the PML + plug-in estimator can be used as a universal estimator for any symmetric distributions with a loss of at most 1/H_phi(p) factor of the samples compared to the optimal estimators. 3. The concept of profile entropy can be of independent interest for other applications beyond those considered in the work.

Weaknesses: The result about competitiveness of PML estimator doesn't recover the previous results on establishing the optimality of PML for several property estimation tasks. So it would be intersting to see when does the optimality of the bound holds.

Correctness: I didn't check all the proofs in the appendix. But the results seem correct to me.

Clarity: The writing of the paper is good overall.

Relation to Prior Work: The paper addresses comparison with previous results nicely overall. One thing to improve is that it is better to indicate explicitly that the upper bound part of Theorem 2 is achieved by a scheme proposed in previous work [Hao and Orlitsky 2019b].

Reproducibility: Yes

Additional Feedback: Minor comments: The lower bound statement of Theorem 2 is a bit hard to understand. The explanation in the following paragraph does a good job but it somehow complicates things. One suggestion is to state it as min_{\hat{p}} \max_{p : H_phi(p) \le H} r_n(p; \hat{p}) \ge H/n. Line 887 in supplementary: "us to " -> "up to"

[Author Response · NeurIPS 2020]

We want to thank all the reviewers for their insightful and valuable comments. All the typos have been corrected. The
rest of the rebuttal addresses specific comments and questions about our results and writing style.

**Reviewer 1:** [*Storyline*] Thanks for commenting on the paper's writing style. Our paper proposes a very novel quantity
and aims to derive a thorough theory for it. Hence, the writing style differs from those that solve a particular problem.

[*Compression and Adaptive Algorithms*] We stated the block compression algorithm just one line below the content
you referred to – in line 227, "we can compress $\varphi(x^n)$ into the set $\mathcal{C}(\varphi(x^n))$ of corresponding multiplicity-prevalence
pairs." The sentence in line 22, in its context, refers to the generic action of designing an adaptive algorithm for the
problem at hand. We have replaced "we design" by "one can design" to eliminate ambiguity.

[*Method of Types*] The word "connects" means "establishing a link" instead of "precisely characterizing the relation."
We agree that the relation between the profiles and the method of types is worth studying. On the other hand, the current
paper already contains 8+ theorems (a few more in the supplementary), 40+ pages proof, and multiple program modules.
Hence, we presented this connection as a future research direction (extension) instead of a major result in line 333-338.

[*Old and New Results*] Since the paper addresses three problems and presents eight theorems, we have, as stated in
line 60, "relegate(d) detailed reviews on related work . . . to the supplementary material." As a result, we have assumed
that the readers are somewhat familiar with the related work. If not, we expect the readers to look at the supplementary
material, which is a good source for background knowledge. An example is property estimation, for which Appendix B
devotes 1.5 pages in B.3 to discuss the prior results and methods. Thanks for your suggestion on presenting such
background materials in the main paper. We plan to add shorter versions of the prior-work reviews in the supplementary
to the beginning of each of the five subsections of Section 2. This should be relatively easy because most theorems are
either entirely new (e.g., thms. 1, 3, 5, and cors. 1, 2) or already equipped with the essential references (e.g., thms. 2, 4).

[*Clarity*] Thanks for suggesting compressing the first part of Section 2.4 and including it in the introduction. We will
modify the paper accordingly, and also reduce the usage of the \paragraph{} command.

[*Example and Notation*] We will add an example for the sample profile and its entropy in Section 1. For the notation,
$\varphi_\mu$ is a function, while $\Phi^n$ is a random multiset, and we followed the convention of capitalizing the random quantities;
$\mathcal{D}_n \simeq \mathcal{H}_n$ means that for any $n$, the two quantities are of the same magnitude, up to a logarithmic factor of $n$ (line 44).

**Reviewer 2:** [*Implications for Practical Problems*] Thanks for the encouraging comments and for suggesting providing
insights on practical applications. The problems we studied in this paper – distribution estimation, property inference,
and profile compression, are quite fundamental. They have wide applications to numerous disciplines, as outlined in
line 79-81, 140-143, and 192-210. Hence, we did not further emphasize the practical implications.

[*Profile Entropy Computation*] For high-dimensional alphabets, even computing a single profile probability often takes
exponential time, and there can be $\exp(\Theta(\sqrt{n}))$ such probabilities. Hence, computing the profile entropy $\mathcal{H}_n$ can be
computationally expensive. On the other hand, Theorem 1 shows that $\mathcal{D}_n$, a quantity computable in near-linear time,
nicely approximates $\mathcal{H}_n$. Appendix A.4 further illustrates how to estimate $\mathcal{H}_n$ with $m \ll n$ observations.

[*Seeing the PML Estimator*] Thanks for your interest in PML. We do not quite understand the meaning of "see the
PML" here. In case you refer to computing the PML precisely, the paper "algebraic computation of pattern maximum
likelihood (ISIT 2011)" provides an exponential-time algorithm with concrete examples. On the other hand, researchers
often employ algorithms that approximate PML, such as the polynomial-time algorithm mentioned in line 183.

[*Asymptotic Closeness between $\mathcal{D}_n$ and $\mathcal{H}_n$*] As both quantities highly depend on the underlying distribution and can
be as large as $\Theta(\sqrt{n})$, currently we do not see the possibility of characterizing the general asymptotic relation.

[*Connection to the Method of Types*] Please see the third point in our response to Reviewer 1 – [*Method of Types*].

[*Empirical Analysis*] Thanks for this point. Please note that we do have 2 sets of experiments in Appendices B.5 and C.4
with complete code accompanied. We did not test the compression algorithm as it was proven to be instance-optimal.

[*Connection to Universal Source Coding*] Thanks for suggesting. Sure, we'll add a section to illustrate this connection.

**Reviewer 3:** Thanks for the insightful suggestions and valuable comments.

[Optimality of PML] We agree that determining the scope of PML's optimality is an important problem. Though
Theorem 1 does not recover some of the prior results on the PML, it applies to all symmetric properties and distribution
collections, hence covering a much broader class of property inference problems.

[Upper Bound in Theorem 2] To be absolutely clear, we have included the statement in line 113 as part of the theorem.

[Lower Bound Statement of Theorem 2] Excellent suggestion on how to present the lower bound. The suggested form is
mathematically equivalent to the original but is more compact. We have modified the theorem's statement accordingly.

[Meta-Review · NeurIPS 2020]

This paper proposes a concept of "profile entropy" for measuring complexity of discrete distributions. They show this concept can be useful in understanding properties of estimation of discrete distributions and compression of sample profiles. The reviewers are in agreement that the work is novel, useful, and mostly well-executed. The reviewers have suggested a number of changes to the presentation format, which the authors seem to have already promised to address in their author feedback. With these changes, the paper should be ready for publication.